# Leveraging Regional Mesh Refinement to Simulate Future Climate Projections for California Using the Simplified Convection Permitting E3SM Atmosphere Model Version 0

Jishi Zhang[1], Peter Bogenschutz[1], Qi Tang[1], Philip Cameron-smith[1], and Chengzhu Zhang[1]

[1]Lawrence Livermore National Laboratory

**Correspondence:** Jishi Zhang (zhang73@llnl.gov)

**Abstract.** The spatial heterogeneity related to complex topography in California demands high-resolution (<5 km) modeling, but global convection-permitting climate models are computationally too expensive to run multi-decadal simulations. We developed a 3.25 km California climate modeling framework by leveraging regional mesh refinement (CARRM) using the U.S. Department of Energy's (DOE) global Simple Cloud Resolution E3SM Atmospheric Model (SCREAM) version 0. Four 5-year time periods (2015-2020, 2029-2034, 2044-2049, 2094-2099) were simulated by nudging CARRM outside California to 1° coupled simulation of E3SMv1 under the SSP5-8.5 future scenario. The 3.25 km grid spacing adds considerable value to the prediction of the California climate changes, including more realistic high temperatures in the Central Valley, much improved spatial distributions of precipitation and snowpack in the Sierra Nevada and coastal stratocumulus. Under the SSP5-8.5 scenario, CARRM simulation predicts widespread warming of 6-10 °C over most of California, a 38% increase in statewide average 30-day winter-spring precipitation, a near complete loss of the alpine snowpack, and a sharp reduction in shortwave cloud radiative forcing associated with marine stratocumulus by the end of the 21st century. We note a climatological wet precipitation bias for the CARRM and discuss possible reasons. We conclude that SCREAM-RRM is a technically feasible and scientifically valid tool for climate simulations in regions of interest, providing an excellent bridge to global convection-permitting simulations.

## 1 Introduction

California is a topographically diverse state known for the rugged Sierra Nevada mountain range, the expansive Central Valley, and its scenic and complex coastline. California has a unique and diverse combination of Mediterranean, mountain, and desert climates each of which includes their own microclimates due to fine-scale heterogeneity caused by complex topography, coastline, and elevation differences. Due to the seasonal persistence of high pressure ridges, California is under the influence of large-scale subsidence that typically results in dry summer months (Karnauskas and Ummenhofer, 2014). The high pressure ridge in combination with marine fog results in a relatively cool summer climate along the coast (Pilié et al., 1979; Samelson et al., 2021), while low-lying inland valleys and desert areas are subjected to a much hotter summer climate. Atmospheric rivers (ARs) are responsible for the majority of California's precipitation (Huang and Swain, 2022) and are characterized as narrow, concentrated moisture surges from the central Pacific Ocean often during wintertime (Ralph et al., 2006; Leung and

Qian, 2009; Dettinger et al., 2011; Chen et al., 2018; Swain et al., 2018; Huang et al., 2020; Rhoades et al., 2021; Huang and Swain, 2022). California's precipitation patterns are highly intermittent, with snowpack acting as a key natural store of that precipitation during the wet winter season and spring snowmelt runoff producing large freshwater supply primarily from the Sierra Nevada (Bales et al., 2011; Hanak et al., 2017). Snowpack is also important to California's energy component, with hydroelectric power providing 56% of the western U.S. energy supply and up to 21% of California's diverse energy portfolio (Stewart, 1996; Bartos and Chester, 2015; Solaun and Cerdá, 2019).

With climate change, California is likely to experience significantly warmer temperatures, less snowpack, a shorter snowpack season, and more precipitation settling as rain rather than snow, resulting in earlier runoff diversions, increased risk of winter flooding, and reduced summer surface water supplies (Gleick and Chalecki, 1999; Hayhoe et al., 2004; Leung et al., 2004). As one of the world's largest agricultural suppliers and a key U.S. energy supplier, changes in regional temperature, precipitation, snowpack, and water availability in California could significantly affect the state's agricultural economy and future power supply capacity (Tanaka et al., 2006; Hanak and Lund, 2012; California Department of Food and Agriculture, 2016; Bartos and Chester, 2015; Pathak et al., 2018; Arellano-Gonzalez et al., 2021). Specifically, recent downscaling studies have found that under the impacts of climate change, California and the western U.S. will experience significant reductions in snowpack, including reduced winter snowfall, earlier spring snowmelt, and increased interannual variability, with important implications for water management and flood risk (Berg et al., 2016; Hall et al., 2017; Musselman et al., 2017; Rhoades et al., 2017; Walton et al., 2017; Musselman et al., 2018; Rhoades et al., 2018a; Marshall et al., 2019; Sun et al., 2019; Siirila-Woodburn et al., 2021). In addition, other renewable energy facilities, particularly wind and solar, are growing rapidly in California, with wind deployment plans are projected to provide 14% of the energy supply by 2050 (Edenhofer et al., 2011; Barthelmie and Pryor, 2014). Predictions of future wind and solar generation in California have also received attention (Crook et al., 2011; Wang et al., 2018).

California's winter precipitation fluctuates dramatically from year to year due to changes in the location of the jet stream, and this strong precipitation volatility can subject California to extreme hydrological events such as megafloods and extreme droughts (Swain et al., 2018; Dettinger, 2016). While the majority of California typically remains dry during the summer months, the high-elevation deserts in the southeast portion of the state can experience brief but intense thunderstorms due to the southwest monsoon (Adams and Comrie, 1997; Prein et al., 2022; Higgins et al., 1999). Virtually all parts of California are vulnerable to relatively long-duration heat waves during the summer months (Gershunov et al., 2009), with inland communities being most affected. These heat-waves not only pose major health risks, but they often contribute to increased wildfire activity. In the autumn, strong Santa Ana/Diablo winds from the interior desert plateau rapidly increase the risk of wildfires (Williams et al., 2019; Keeley et al., 2009). In addition, the intricate variability of temperatures and climates over very short distances across California, such as the cool downslope mountain-valley circulations at night (Zängl, 2005; Pagès et al., 2017; Jin et al., 2016; Junquas et al., 2018) and the elevation dependence of snow-rain transition (Guo et al., 2016; Minder et al., 2018; Winter et al., 2017; Rhoades et al., 2016, 2017), underscore the state's vulnerability to diverse climate extremes. The processes that give rise to these microclimates require the use of high-resolution models to understand their interactions and project how they may become altered under climate change.

To reliably predict climate change in California and assess the impacts of extreme events in the future, high-resolution climate simulations are needed to resolve microclimate features that are highly dependent on the fine-scale heterogeneity. These include topographic precipitation, mountain snowpack, coastal fog, Santa Ana winds, etc. For example, Caldwell et al. (2019) found that 25 km was necessary to capture general mountain topography and the associated climatological precipitation patterns with fidelity. Tang et al. (2023) showed that the topographic precipitation and mountain snowpack are improved in the E3SMv2 North American 25 km Regionally Refined Model overview relative to the 100 km configuration. However, Huang et al. (2020) found that even higher resolution, specifically ~3 km, was needed to accurately simulate and predict precipitation distributions and potential hazard impacts related to AR events. Rhoades et al. (2023) recently evaluated RRM-E3SM at 14km vs 7km vs 3.5 km horizontal resolutions and demonstrated the forecast skill of 3.5 km in recreating extreme floods. A 3-km resolution represents the typical convection-permitting scale, and thus the resolution advantages go far beyond the ability to simulate ARs since the uncertainties associated with deep convection parameterizations can be avoided (Hohenegger et al., 2008; Chikira and Sugiyama, 2010; Kendon et al., 2012; Ban et al., 2014; Prein et al., 2015; Yano et al., 2018; Neumann et al., 2019; Stevens et al., 2019; Lucas-Picher et al., 2021; Gao et al., 2022, 2023). As an example, Caldwell et al. (2021) found that many long-standing biases typically associated with conventionally parameterized GCMs are significantly reduced when run at ~3 km horizontal resolution.

Recently, global convection-permitting models (GCPMs) have become a reality thanks to advances in high performance computing (HPC), algorithms, and software optimizations (Satoh et al., 2019). However, it is still very computationally expensive and difficult to perform interannual climate simulations using GCPMs and most simulations using these type of models have thus far focused on durations of $\sim$40 days (Stevens et al., 2019; Caldwell et al., 2021; Hohenegger et al., 2023). Higher resolution requires smaller time steps to achieve numerical stability, which contributes greatly to the cost. In addition, managing the large volumes of data produced by GPCMs further adds more complication. Given the expensive cost of GCPMs, regional climate models (RCMs) have played an important role in the last decades (Giorgi, 2019; Gutowski et al., 2020), allowing for low-resolution boundary condition data to be dynamically downscaled to high resolution over regions of interest. The low-resolution GCMs have been able to provide plausible large- and synoptic-scale climatologies given large scale forcing (e.g., future emissions and land use changes described in future scenario projections). The sub-grid scale processes are represented by downscaling techniques (Giorgi, 2019). While RCMs were developed based on limited-area nesting models, GCMs now have the capability to employ variable resolution grids and regionally refined meshes by capitalizing on unstructured grid development (Fox-Rabinovitz et al., 2006; Abiodun et al., 2008; Tomita, 2008; Zarzycki et al., 2014; Skamarock et al., 2018). In contrast to regional CPMs, which refer to regional climate models with limited areas (e.g., Prein et al., 2015; Kendon et al., 2017), RRMs are global models. When RRM is run freely, it works exactly like a typical GCM (e.g., Tang et al., 2023), and there are studies discussing the upscale effects of the refined area in large-scale circulations (e.g., Sakaguchi et al., 2016). Thus, although both can be pushed to a CP resolution, RRM and limited-area regional models are fundamentally different in terms of grid structure and evolutionary history.

Modern regionally refined models (RRM) allow for a gradual transition of the grid from the synoptic scale to the kilometer scale (Harris and Lin, 2013; Zarzycki and Jablonowski, 2014; Guba et al., 2014; Zarzycki et al., 2014; Rauscher and Ringler,

2014; Harris et al., 2016; Tang et al., 2019). A unique feature of RRM is that it allows for a seamless transition from coarse to fine resolution regions, provided that the model has physical parameterizations that are scale aware. It can also be implemented as a configuration that more closely resembles a RCM by "relaxing" or "nudging" the refined region to atmospheric and land/oceanic boundary conditions outside the region of interest (Gutowski et al., 2020). RRM methods have been used in idealized aquaplanet simulations (Rauscher et al., 2013; Rauscher and Ringler, 2014; Zarzycki et al., 2014) and AMIP and

fully-coupled simulations (Rhoades et al., 2016; Wu et al., 2017; Huang and Ullrich, 2017; Tang et al., 2019; Rhoades et al., 2020a; Tang et al., 2023). RRM is a powerful tool because it has the ability to replicate results in a region of interest when compared to global simulations with uniform high resolution (Bogenschutz et al., 2022; Liu et al., 2023). The cost of RRM is dominated by the high resolution region, meaning that a high resolution mesh that covers about 10% of the globe would roughly be equal to about 10% of the cost of running the entire globe at this resolution. Thus, the substantial cost savings RRM

provides enables one to run longer duration simulations or to produce larger ensemble size compared to a GCPM.

Given 1) the impact of climate change on California and the effects it has on the U.S economy and energy infrastructure, 2) the requirements of California's complex fine-scale heterogeneity for convection-permitting scale modeling, and 3) the purpose of exploring climate change response in long duration integrations; this work proposes to develop a California convection-permitting climate modeling framework. This framework is based on the Simple Cloud-Resolving E3SM Atmosphere Model

(SCREAM) developed under the United States (U.S) Department of Energy (DOE) Energy Exascale Earth System Model (E3SM) project (Caldwell et al., 2021) and RRM configuration (Tang et al., 2019, 2023). This is the first time that SCREAM is being used for climate-length simulations. One of the main purposes of this paper is to document the modeling strategy used to perform this ambitious SCREAM RRM simulation, with the idea that one could replicate these methods to be used in other regions and/or time periods. In addition, by comparing our simulation results to that of a traditional GCM, we aim to highlight

the importance of high resolution to accurately simulate regional climate patterns and changes in California.

This paper is organized as follows: Section 2 describes the methodology we used, including the California RRM framework, future projection experiment design, and model evaluation strategy. Section 3 presents the results of SCREAMv0 California RRM including a baseline comparison with observations and an analysis of the future projection. Finally, in section 4 we conclude with a discussion on the implication of our results as well as a summary on the application of SCREAM RRM for

RCMs.

## 2 Methods

In this section we will first focus on the modeling strategy used in this study, which can be used as guidance for future studies aiming to use SCREAM RRM for different regions. It includes the descriptions of SCREAM, the regionally refined model framework, nudging strategy, and future projection experiment. Then we will provide our methodologies for evaluation.

## 2.1 Modeling Strategy

### 2.1.1 SCREAM Description

The framework for the California convection-permitting RRM in this paper is developed using SCREAM version 0 (Caldwell et al., 2021), developed under the U.S. Department of Energy (DOE) funded E3SM project (Golaz et al., 2019). SCREAM has a global resolution of 3.25 km and thus does not parameterize deep convection. SCREAM uses the Simplified Higher Order Closure (SHOC) (Bogenschutz and Krueger, 2013) to serve as a unified cloud macrophysics, turbulence, and shallow convective parameterization, the Predicted Particles Properties (P3) cloud microphysics scheme of (Morrison and Milbrandt, 2015), and the RTE + RRTMGP radiative transfer package to calculate gas optical properties and radiative fluxes (Pincus et al., 2019). The average aerosol climatology is interpolated from an 1° E3SMv1 simulation (Zhang et al., 2013; Wang et al., 2020; Zhang et al., 2022). Caldwell et al. (2021) shows that SCREAM has an excellent performance in the simulation of vertical profile of tropical clouds and coastal stratocumulus, tropical/extratropical cyclones, ARs, and cold air outbreaks; making it well suited to serve as the model for the California RRM framework.

SCREAM's dycore enables the numerical solution of the nonhydrostatic equations of motion (Taylor et al., 2020) using the High Order Method Modeling Environment (HOMME). HOMME uses virtual potential temperature as the thermodynamic variable with semi-Lagrangian tracer transport, which enables the use of much larger time steps while maintaining advective stability compared to explicit Eulerian methods. The time discretization uses an IMplicit-EXplicit (IMEX) Runge Kutta method in which an implicit Butcher table for terms responsible for vertically propagating acoustic waves and an explicit Butcher table used for most equations. The HOMME dycore consists of spectral elements, with each element containing $4 \times 4$ grid of Gauss-Lobatto-Legendre (GLL) nodes, while the physics is handled by a uniformly spaced $2 \times 2$ grid (called "pg2" grid), which substantially increases the model throughput (Hannah et al., 2021).

SCREAM contains 128 layers in the vertical, compared to the 72 vertical layers in E3SM, though the model top in SCREAM is lower (40 km vs. 60 km). Thus, the vertical resolution in SCREAM is nearly twice that of E3SM at most layers, with enhanced vertical resolution in the lower troposphere. In particular, the improved vertical resolution of the lower troposphere was found to be a factor that improved marine stratocumulus (Bogenschutz et al., 2021, 2022), which is important for representing the California coastal climate.

E3SMv1 land model (ELM) Golaz et al. (2019) is placed on the same RRM mesh as the atmosphere model. The river routing model (Model for Scale Adaptive River Transport) uses a lat-lon grid with the spacing of 0.125° (Li et al., 2013). The prescribed-ice mode from the Los Alamos sea ice model CICE4 Hunke et al. (2008) and the data ocean model are used in our study.

### 2.1.2 RRM in California

The configuration of the California 3.25 km RRM (hereafter referred to as "CARRM") in this work consists of two main parts, the first of which deals with the design of the regionally refined grid and its associated model configuration files (e.g., domain

files, topography, atmospheric initial condition, land surface, etc.). The second part handles the generation of the boundary conditions from the low-resolution (1°) GCM and nudging settings (to be described in section 2.1.4).

The CARRM grid is progressively refined from the outer global resolution of ne32 (corresponding roughly to a resolution of ~100 km) to the convection-permitting scale for California (ne1024, 3.25 km), with a 8th order ($2^8$) refinement between them (Fig. 1). We created the CARRM grid using the offline software tool Spherical Quadrilateral Mesh Generator (SQuadGen; https://github.com/ClimateGlobalChange/squadgen, last access: 21 January 2024). The choice of the finest domain may affect the RRM simulation behavior, but there are no precise rules on how to choose the best domain. Our basic considerations include: 1) suitablility for the science applications, 2) the need for the domain to cover the entire state of California, 3) avoiding having the domain boundary reside near substantial topography, and 4) the desire to keep the domain as small as possible to avoid excessive computational expense and allow for long integrations. We note that atmospheric rivers originating from the central/eastern Pacific are important to California precipitation, but 1° GCMs are sufficient to resolve the synoptic scale features of these systems (Giorgi, 2019; Neumann et al., 2019). The sensitivity of the size of the refined mesh for the simulation of atmospheric rivers was explored with CESM (Rhoades et al., 2020a).

The topography file was generated using the NCAR topography toolchain (Lauritzen et al., 2015), with tensor hyperviscosity enabled for the RRM grid. Figure 1 shows the topography used for 1° E3SMv1, the E3SMv2 North American 25 km RRM, and the California 3.25 km RRM used in this study, respectively. Since the topography files are on the GLL node, we used matplotlib's "tricolor" function to represent the native spectral element data as accurately as possible, with each triangle's color taken from three GLL vertexes (https://matplotlib.org/stable/api/_as_gen/matplotlib.pyplot.tripcolor.html, last access: 21 January 2024; note that the "tricolor" function doesn't allow a manually specified color levels). As a reference, Fig. 1 displays the 3.25 km topographic data from The United States Geological Survey (USGS) used to downsample to the destination resolution of RRM. Note that 3.25 km is the nominal resolution, and that the effective resolution (fully resolved scale derived from kinetic energy spectra compared to observations) of California is actually at about 6 times the nominal resolution (Neumann et al., 2019; Caldwell et al., 2021). CARRM topography essentially captures the fine spatial patterns shown in the 3 km USGS data, such as features of the Sierra Nevada, coastal ridges, and the Central Valley. This is not surprising, since CARRM's topography was processed from USGS GTOPO 1 km data and then interpolated to 3 km cube sphere.

The atmosphere initial condition was generated with the HICCUP package (https://github.com/E3SM-Project/HICCUP, last access: 21 January 2024), which has a built-in download of ERA5 pressure level data. HICCUP interpolates the ERA5 data to the model's vertical levels using NCO's vertical interpolation algorithm (Zender, 2008) and to the horizontal resolution using a TempestRemap horizontal interpolation algorithm (Ullrich and Taylor, 2015; Ullrich et al., 2016). We adopted the higher order algorithm here. The surface temperature and pressure are adjusted using a procedure described in Trenberth et al. (1993) based on the topography elevation difference plus a dry hydrostatic atmosphere lapse rate. This procedure also avoids extrapolating excessively high/low pressure values by resetting the surface temperature from extremely warm/cold terrain. The CARRM mesh used in this work contains good grid properties (max Dinv-based element distortion is 3.02[1]). The atmosphere

---

[1]It indicates a high quality RRM grid if the maximum Dinv-based element distortion is less than 4. See https://acme-climate.atlassian.net/wiki/spaces/DOC/pages/872579110/Running+E3SM+on+New+Atmosphere+Grids, last access: 19 February 2024.

**Table 1.** Column numbers and timesteps used in E3SMv1, E3SMv2 NARRM and SCREAMv0 CARRM.

| Model | Column no. | | Time steps (s) | | | | |
| | Dynamics | Physics | Dynamics | | | | Physics |
| | | | Dycore | Dycore Remap | Advection | Hyperviscosity | |
| E3SMv1 | 48,602 | 48,602 | 300 | 900 | 300 | 100 | 1800 |
| NARRM | 130,088 | 57,816 | 75 | 150 | 450 | 75 | 1800 |
| CARRM | 152,712 | 67,872 | 9.375 | 18.75 | 75 | 9.375 | 75 |

initial condition is in balance, which is possibly benefited from the surface adjustment (otherwise, instability would occur using this IC directly). As a result, we did not need to spin up the atmosphere and adjust the hyperviscosity incrementally. The hyperviscosity timestep for dynamics are set to the default value used in SCREAM 3.25 km global simulations.

### 2.1.3 Timesteps and computational cost

CARRM has a total of 152,712 GLL columns (dycore) and 67,872 physical columns (pg2 grids). For reference, E3SMv1 has 48,602 physical columns (Golaz et al., 2019) and the E3SMv2 North American 25km RRM (NARRM) has 57,816 physical columns (Tang et al., 2023), representing a slightly higher storage demand for CARRM compared to NARRM (Table 1).

Table 1 provides the timesteps we used for CARRM simulations. Because the timesteps must be uniform globally based on the finest region, our configuration follows the parameters used in the global convection-permitting simulation of SCREAMv0 (Caldwell et al., 2021).

All CARRM simulations were performed using the Livermore Computing (LC) Quartz machine with Intel(R) Xeon(R) CPU E5-2695 v4 @ 2.10GHz 36-core 120 nodes using only MPI processes. We used a 120-node configuration to balance throughput and queue time. Although we did not systematically evaluate the performance of CARRM, we found that scaling from 30 to 120 nodes was quite good in 1-month testing, with almost no loss of scaling performance. Jobs were resubmitted once every simulated month and the total throughput (including I/O) was about 0.68 simulation years per day, or about 240 simulation days per day. For comparison, the global SCREAMv0 simulation Caldwell et al. (2021) run on the National Energy Research Supercomputing Center (NERSC) Cori Knights Landing (KNL) used 1536 nodes (68 physical cores per node) with a throughput of 4-5 simulation days per day. The NARRM was run on Argonne National Laboratory Chrysalis which used 80 AMD Epyc 7532 64-core nodes with a throughput of about 10 simulated years per day (Tang et al., 2023).

In addition to occasional node failures, we encountered several instability failures during the simulation with "EOS bad state: d(phi), dp3d or vtheta_dp < 0" or "negative layer thickness" model produced errors. While the specific cause of these errors is unclear, we note that all errors were produced between the months of November and April, thus could be a result of topography-related baroclinic instability associated with winter storms. The error frequency is: 3 times for 2015-2020, 7 times

for 2029-2034, 2 times for 2044-2049, and 3 times for 2094-2099. We got around these instability failures by temporarily halving the model time steps uniformly. All instances have been properly documented to ensure reproducibility.

### 2.1.4 Nudging Strategy

Since SCREAM does not have a deep convective parameterization, and hence lacks the ability to run with a 100 km resolution, we cannot perform a completely free running integration using CARRM. We use the approach of RCMs, using lower and lateral boundary conditions provided by future scenario simulations from low-resolution GCMs to provide coarse scale fields that drive CARRM.

We reproduced the future projection scenario (to be described in further detail in section 2.1.4) described in Zheng et al. (2022) using the 1° fully coupled E3SMv1. We output the 3-hourly vertical distribution of winds, temperature, and specific humidity. The consistency among the boundary conditions is important because the internal variability is fully dependent on this unique realization. Sea surface temperature (SST) and ice cover were obtained from the same coupled simulation as lower boundary conditions to drive Data Ocean and Prescribed CICE4 (the latest Los Alamos sea ice model) (Hunke et al., 2008).

The e3sm_to_cmip tool (https://github.com/E3SM-Project/e3sm_to_cmip, last access: 21 January 2024) was used to get 1° lat-lon timeseries which were further processed to meet the format of the Data Ocean streamfile (https://esmci.github.io/cime/versions/ufs_release_v1.1/html/data_models/data-ocean.html, last access: 21 January 2024). We retrospectively noticed that the step of replacing the missing value of SST to -1.8 °C in the streamfile generation procedure caused the model to regard that the "-1.8 °C" value over land is valid. This caused some points along the coastline to inherit a spurious cold SST from the

1° streamfile. This spurious signature is directly reflected in the SST and surface fluxes from the RRM output with little direct effect on the variables not at the bottom level of the atmosphere.

The nudging capability that has been implemented into E3SM and used by RRM is described in Tang et al. (2019), which allows for selected areas of the globe to be nudged while allowing other regions to be simulated freely. In this work we want to nudge the coarse outer domain, but allow for the high resolution mesh over California to integrate freely. To allow this, a

nudging coefficient is set by a Heaviside window function from 1 (other global areas) to 0 (where California is fully covered, free run) in the lat-lon direction (Fig. 2). The nudging strength is consistent in the vertical direction.

The winds, temperature, and specific humidity profiles were interpolated vertically by NCO and horizontally by the TempestRemp higher order algorithm. Lateral boundary conditions were updated every 3 hours by linearly interpolating each pair of nudging time slices (current timestep and the next 3 hours) onto the model's physical time step with a relaxation timescale

of 2 days. The selection of a 2-day relaxation timescale was not the result of an exhaustive study to find an optimal time scale, as running CARRM is still relatively expensive thus making tuning fairly time-intensive. However, we did test relaxation timescales of 1 h, 6 h, and 24 h. We found that the 2-day timescale gave the most consistent results between RRM and 1° E3SMv1 global precipitation patterns and the smallest bias for California precipitation.

We found that when the nudging strength is very strong (timescale = 1 hour), a spurious circulation formed in California,

which may be due to the inconsistency between the temperature of the boundary forcing and that of the freely integrated spin-up temperature over California; when the two are coupled too frequently, the large gradient of temperature across the nudging

boundary will force the wind shear to adjust by thermal wind balance. Therefore, a very short relaxation timescale is not desirable. The 3-hourly evolution of instantaneous total (vertically integrated) vapor transport for 1° E3SMv1 and SCREAMv0 CARRM on 2097-12-21 are shown in Fig. 2 for an atmospheric river event as it makes landfall on the west coast. This is just

one example to show that the general meteorology and climate of the E3SMv1 simulation are well reproduced in the 100 km domain of SCREAM. Note that there are some differences between them, which is expected to be a natural effect of nudging, especially since we used a weak relaxation timescale.

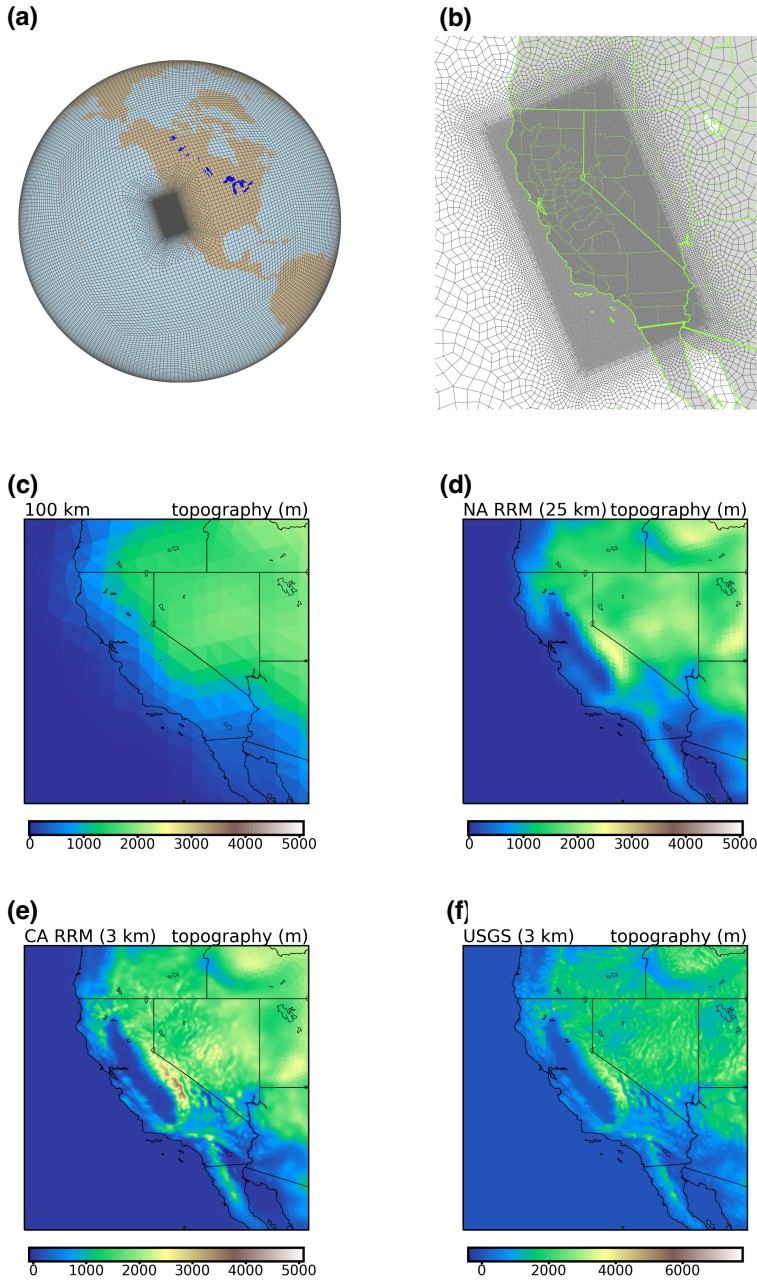

**Figure 1.** Regionally refined grid for CARRM (a-b), topography for (c) 1° E3SMv1, (d) US 25 km RRM, (e) CARRM, and (f) the United States Geological Survey (USGS) topography. All topography data are zoomed to the western United States.

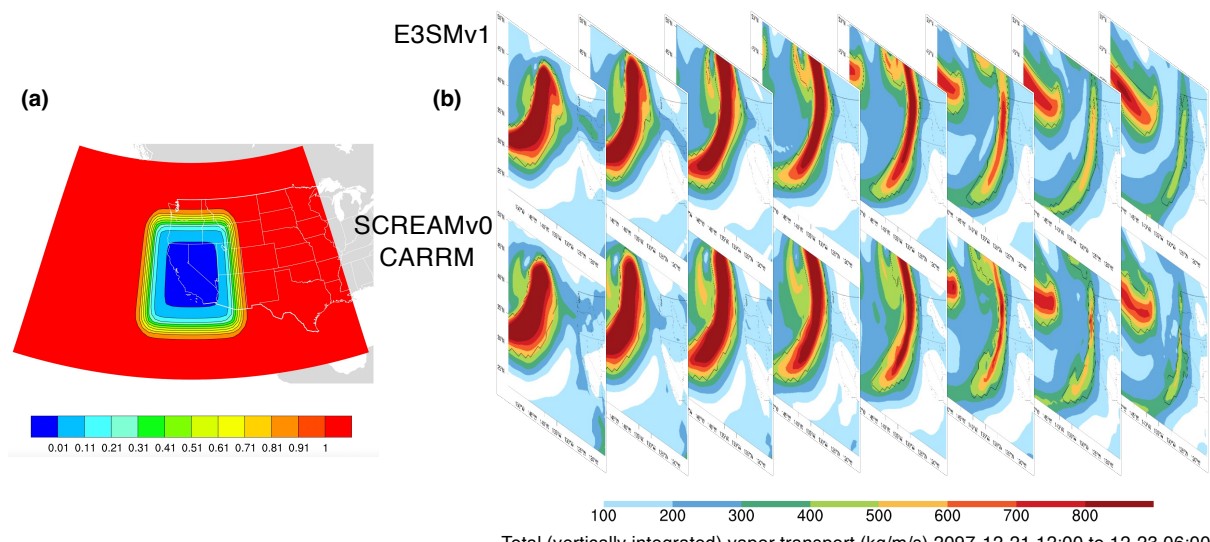

**Figure 2.** (a) Nudging coefficient map over California where nudging is not applied in red areas, (b) 6-hourly evolution of instantaneous total vertically integrated vapor transport in 2097-12-21 for E3SMv1 and SCREAMv0 CARRM.

### 2.1.5 Future Projection Experimental Design

We choose the high-emission SSP5-8.5 scenario for our future climate projection, which is comparable to the radiative forcing path of the highest representative concentration path (RCP8.5). We recognize that SSP5-8.5 is a "worst" case scenario that is unlikely to happen, due to policy interventions that promote carbon emission mitigation and sequestration, and thus represents an upper bound case of the ScenarioMIP (Kriegler et al., 2017). However, the differences between the more plausible SSP3-7.0 and SSP5-8.5 before 2050 are relatively small (Masson-Delmotte et al., 2021; Tebaldi et al., 2021). Both of these scenarios

predict similar development trends, including high GHG emissions, increased energy usage, and limited climate change miti-
gation measures before 2050 (O'Neill et al., 2016). The chief reason for our choice to run the SSP5-8.5 scenario is due to the
fact that we had to re-run the publicly available version of E3SM (i.e., version 1) to produce the necessary nudging data for the
coarse grid region, and SSP5-8.5 is the only scientifically validated scenario for the publicly released v1 future projection.

Given the relatively high cost of CARRM, we choose to run four 5-yr segments rather than integrating the entire 85-yr
SSP5-8.5 simulation. Our goal is to pick segments which represent various points within the 85-yr future projection time
line. In addition, we consider the El Niño-Southern Oscillation (ENSO), which can explain hydrological events in California
(Harrison and Larkin, 1998; Dettinger et al., 1998; Wise, 2012; Hoell et al., 2016; Patricola et al., 2020; Mahajan et al., 2022).
Since internal variability like ENSO is well inherited from the boundary forcing (Giorgi, 2019; Laprise et al., 2000), our
nudging strategy enables us to conduct RRM simulations by selecting the time periods containing a strong ENSO signal.

As an expediency, the spatial and temporal variability of the California climate may be better represented by selecting the
time periods with larger ENSO variability since we are only able to run one ensemble member. Our segment-simulation strategy
also accelerates the validation process of CARRM framework, and in particular allows us to provide simulation outputs as soon
as possible to the downstream energy infrastructure experts, who are more interested in validating a certain time slice (e.g.,
mid-century) or specific extreme events (e.g., heat waves, floods, wildfires) rather than the entire time series. For the full period
2015-2100 of SSP5-8.5, we chose 2015-2020 (as baseline), 2029-2034 (which includes a strong El Niño year followed by a
strong La Niña event lasting three years), 2044-2049 (mid-century of interest to the infrastructure planners), and 2094-2099
(the end of the century), for a total of 20 years (Fig. 3). Here all usage of the word "year" refers to "water year" (from October
to the next September). One can also cast the simulation segments as being run according to different global warming levels
of interest to the IPCC AR6 reports. From another perspective, the four simulation segments provide different levels of global
warming (about 0.9 °C, 1.7 °C, 2.8 °C, 7.6 °C) relative to the 1850-1869 baseline (Zheng et al., 2022).

In retrospect, when we examine the relationship between precipitation and ENSO across the four segments, the 5-yr mean
precipitation barely reflects the ENSO signal. In addition, we do not see a significant modulation of the ENSO on monthly pre-
cipitation. Instead, the climate change signal seems to be more dominant, with heavy precipitation events occurring essentially
every year at the end of the century. Compared to CESMv1-LE, the ENSO variability in E3SMv1 piControl and historical
ensemble simulations is slightly closer to observations, while strongly shifted to a 3-yr period. The overall score for the spatial
pattern compared to observations is also higher, but still muted along the North American coast (Golaz et al., 2019). This may
partially limit the ability of ENSO to modulate the climate in our simulations.

To provide well-established regional climate projections, a three-step approach is usually used (Giorgi, 2019): 1) drive a
high-resolution model with a reanalysis dataset to identify biases in the model dynamics/physics and nudging strategy, akin
to a hindcast as described in Ma et al. (2015), 2) drive the high-resolution model with historical GCM simulations to identify
climate change signals for given historical periods and identify the biases from low-resolution GCMs (baseline), and 3) perform
regional future projections driven by the same GCM to assess climate change signals for future time slices by comparing with
the baseline. One reason for not performing the first step in this paper is that hindcast-style simulations are primarily useful in
short-term simulations to help select the physical schemes with optimal performance in the region of interest. However, unlike

commonly employed regional climate downscaling approaches such as the Weather Research and Forecasting (WRF) model, SCREAM does not have multiple physics options to choose from. We note that we have performed hindcasts of several AR events with CARRM, which will be described in future publication. In addition, we integrate steps 2 and 3 since we treat the first 5 years of SSP5-8.5 as a baseline (2015-2020, akin to a historical run) in which we compare the simulated climatology to observations.

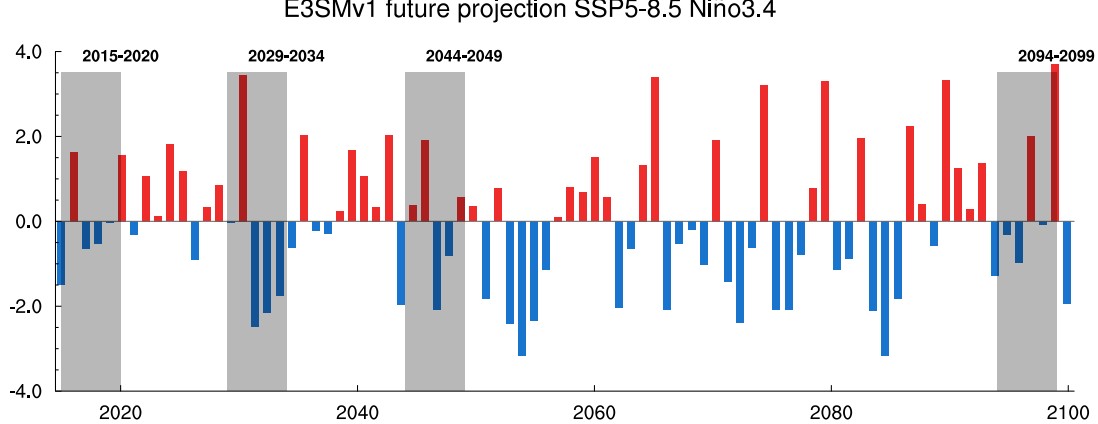

**Figure 3.** Niño 3.4 index from E3SMv1 SSP5-8.5 projection. Shaded areas are labeled with the four segments of the CARRM simulations. The global warming levels for the four simulation segments are about 0.9 °C, 1.7 °C, 2.8 °C, 7.6 °C relative to the 1850-1869 baseline.

## 2.2 Evaluation Strategy

### 2.2.1 Evaluation datasets

To properly evaluate CARRM, it is important to compare against observational datasets of sufficient temporal and horizontal resolution since one would expect typical added values from convection-permitting simulations most likely to occur at small temporal and spatial scales. Moreover, it is desirable that the observational datasets cover a long record to account for the possible range of natural variability.

In this study we use the 4 km PRISM (Parameter-elevation Regressions on Independent Slopes Model) observation-based gridded dataset of 30-yr normal to evaluate maximum, average, minimum temperature and precipitation (PRISM Climate Group, Oregon State University, https://prism.oregonstate.edu, last access: 21 January 2024). PRISM adopts the primary assumption that "elevation is the most important factor in the distribution of climate variables" for a localized region, and calculates the local climate-elevation relationship by considering coastline, temperature inversion, cold pool, topographic factors, etc. to weigh the in-situ data. For example, PRISM calculates precipitation-elevation regression functions under each category based on slope orientation categories to distinguish precipitation on windward and leeward slopes. PRISM and other observation-based gridded products has been known to underestimate extreme precipitation (particularly from ARs) (Lundquist et al., 2019; Rhoades et al., 2023). Given this issue in PRISM and other gridded products and the potential to "falsely" attribute

an over precipitation bias, we also use a probabilistic gridded product for daily extreme precipitation (Risser et al., 2019).
This probabilistic data provides 10- to 100-year return values for the largest seasonal daily precipitation. For CARRM, we compute the 10-year return values of the largest seasonal daily precipitation based on the 20 years of available outputs. The location, shape, and scale parameters for the Generalized Extreme-Value (GEV) distribution were estimated using Maximum-Likelihood Estimation in NCL. However, we only have 20 years of simulation in total, so we cannot reasonably estimate the parameters for the GEV distribution for daily precipitation extremes. In addition to PRISM, we use the unsplit Livneh gridded product, which does not underestimate the extreme precipitation as much when compared to the time-adjusted Livneh (Pierce et al., 2021). To evaluate the snow water equivalent (SWE) we use assimilated snow observations developed by the University of Arizona (UA-SWE), and Western United States UCLA Daily Snow Reanalysis (WUS-SR) Version 1. The UA-SWE data (Zeng et al., 2018; Broxton et al., 2019) were derived from in-situ measurements from Snow Telemetry network and Cooperative Observer Program with assimilated temperature and precipitation from PRISM. This is a 40-yr dataset and has a spatial resolution of 4 km. The WUS Snow Reanalysis (Fang et al., 2022a) has an ultra-high resolution of 500 m from water years 1985 to 2021, which assimilated cloud-free Landsat observations (Fang et al., 2022b). For consistency in the analysis period, all observation-based gridded products were analyzed for the water years 1984 to 2020, unless otherwise stated.

In addition to the observation-based gridded products, we use in-situ temperature and precipitation measurements from Global Historical Climatology Network (GHCN) (Menne et al., 2012a, b), and SWE from Snow Telemetry (SNOTEL) network (https://nwcc-apps.sc.egov.usda.gov/imap/, last access: 15 February 2024). Four representative sites are chosen for GHCN: Sacramento, San Francisco, Tahoe City, and Death Valley. The stations for SWE are Tahoe City, Adin Mtn, Truckee, and Leavitt Lake so that they match the available SNOTEL sites. We choose those stations to represent the varying microclimate across California and for their proximity to populated cities. Only values with an empty QFLAG field are kept in GHCN records, meaning they pass all quality assurance checks. The temperatures of -60.3 F in SNOTEL records seem to be invalid and are set to missing. We also obtained the timeseries of PRISM and UA-SWE for the same locations. The period for in-situ records is different among stations, dataset and variables. The 1989-2020 water years are used in all station analyses. In addition to serving as the "truth" in the comparison to CARRM, the in-situ observations also provide an additional comparison to the observation-based gridded products and highlight uncertainties from the gridding process/statistical co-variate assumptions employed in these products.

To characterize unstructured grids and model/observation raw resolutions as directly as possible, all analyses in this paper are based on the model's native grids (unless otherwise stated). Most output variables of SCREAMv0 reside on physical columns, except for those output from the dycore (GLL columns). Each coordinate of the physical (pg2) grid corresponds to four vertices and can be drawn directly by NCL's CellFill method without interpolation, where each color block represents the cell average of the physical column data. To match the pg2 grid of CARRM, we interpolated the GLL column output of E3SMv1 to the physical column with the higher order (atmosphere output) or monotune (land output) algorithm via TempestRemap. For the calculation of California regional averages, a mask file was generated using a high-resolution California shapefile, then the regional averages were obtained by NCO's ncra calculator with mask and grid-area weights being applied. The statistics of a single grid point are obtained directly by extracting the time series of that point.

### 2.2.2  Atmospheric river tracking with TempestExtremes

The response of atmospheric river (AR) contributed precipitation with climate change in California is briefly analyzed in Section 3.3. We used TempestExtremes 2.2.1 (Ullrich and Zarzycki, 2017; Ullrich et al., 2021) to track the 6-hourly instantaneous IVT (total vertically integrated vapor transport) with the key parameters including: 1) minimum laplacian of IVT = 20000 kg/m/s, 2) latitude of AR tagged grid point > 15°, and 3) blob area of IVT > $4 \times 10^5 \text{km}^2$. We did not isolate single AR events in TempestExtremes using StitchBlobs in order to compute the PDFs with as large a sample size as possible. Using StitchBlobs
would make the sample size of variables corresponding to individual AR events in each 5-year winter very small. As a result, we did not divide ARs into a category-based definition such as in Ralph et al. (2019) and Rhoades et al. (2020b). Therefore, the terminology "AR" in the context of this paper is strictly AR-related IVT 6-hourly samples.

Note that the tracker must be applied to an orthogonal grid, and we interpolated the model output to the 1° lat-lon grid by the TempestRemap higher order algorithm in advance. For simplicity, we did not stitch AR tracks and treat AR and California
precipitation as one-to-one samples every 6 hours. To explore the relationship between AR and California precipitation, we calculated the following statistics for each simulation period December-January-February (DJF):

- The percentage of California precipitation contributed by ARs. AR-contributed California precipitation was obtained by interpolating the 1° AR mask back to the model's native pg2 grid and then associating any precipitation as AR-produced when AR masks exist over California.

- The highest latitude reached for each AR making landfall on California.

- The "duration" of an "AR" after California landfall, obtained by counting the sample size of AR mask that makes landfall in California and multiplying 6 hours. We recognize this is different from the concept of an event's duration and does not require the samples to be sequential.

- The maximum IVT (the intensity of AR snapshots) within each AR mask that makes landfall in California

- The average TMQ (total vertically integrated precipitable water) of each AR mask that makes landfall in California

- The average 850 hPa zonal wind speed of each AR mask that makes landfall in California

## 3   Results

### 3.1   Baseline comparison with observations

To compare with observations, we use a baseline with the first five water years (2015.10-2020.09) of the SSP5-8.5 projection.
Since the simulation period is not corresponding to the "real world" (because our simulations are not hindcasts using realistic boundary conditions), the simulation can only be compared to observations in a statistical sense (e.g. long-term averages).

For air temperatures at 2 m height (hereafter referred to as "T2m"), Fig. 4 clearly shows much richer spatial patterns simulated in CARRM than the 1° E3SMv1. The 1° E3SMv1 largely fails to capture prominent temperature gradients associated

with the coastline, Central Valley, Sierra Nevada, and Mojave/Colorado Desert. Compared to PRISM, CARRM produces a
very realistic spatial distribution of daily maximum, mean, and minimum T2m. Good representation of complex topography
can form temperature gradients simply by lapse rate effect, and cooler/denser air masses at night tend to drive subsidence
warming in the valley. Note that the daily maximum T2m are slightly higher in CARRM than in PRISM in parts of the Central
Valley (up to 2°C), while the maximum T2m is underestimated by 2-4°C over the Colorado Desert and by 0-3°C in the Sierra
Nevada. Daily minimum T2m is overall warmer (up to 2-5 °C) in CARRM than in PRISM (also see Fig. 9), and the mean T2m
is fairly similar in RRM against PRISM. Caldwell et al. (2021) reported that SCREAMv0 does have an overall warm bias for
T2m, especially at high latitudes, while we also see the cold bias in daily maximum T2m. A further comparison with GHCN
and PRISM at Tahoe City shows that the seasonal mean of maximum T2m in JJA is 1-2 °C colder than GHCN/PRISM (Fig.
8c), while the minimum T2m in SON is about 2 °C warmer than GHCN/PRISM ((Fig. 9c). Note that the simulations represent
only 5-year averages whereas PRISM represents 30-year averages. This is especially important given the large interannual
variability in the California climate, and might obscure "warm" or "cold" biases (and is relevant for the results to be presented
for precipitation/snowpack).

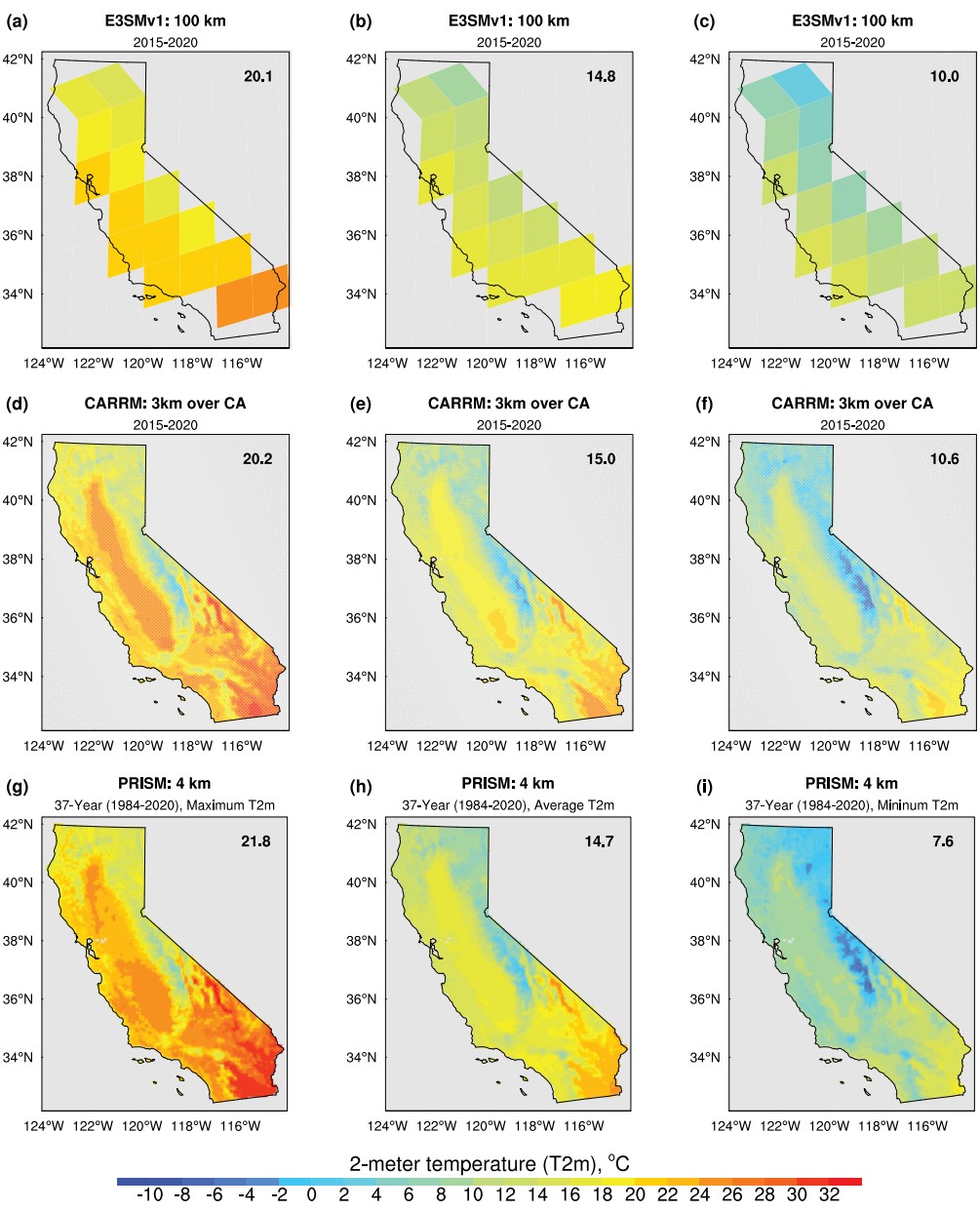

**Figure 4.** Baseline (2015-2020 water years) multi-year average daily maximum (left column), mean (middle column), and minimum (right column) 2-meter temperatures (referred to as "T2m", °C) from 1° E3SMv1 (top row), SCREAMv0 CARRM (middle row), and PRISM observation-based gridded product (bottom row). The Statewide average is shown in the right-top corner.

The temporal and spatial variability of precipitation is more pronounced than that of temperature in the state of California. Dettinger et al. (2011) highlights the large interannual variability of California precipitation, which warns of the potential issues with comparing 5-year vs 30-year normals. Therefore, we also show the five wettest and driest water years during 1981-2020 for PRISM analysis and unsplit Livneh gridded product in addition to the 30-yr average to characterize the observed natural variability (Fig. A1a-f). The high mountains (i.e. the Sierra Nevada, the Cascade Range, and the Klamath Mountains) manifest a significant topographic precipitation pattern with moist air coming from the northwest, and higher annual rainfall in the north than in the south. In addition, the relatively smaller ranges, such as the Transverse and Peninsular ranges of southern and central CA, also receive considerable annual-mean precipitation. The northern part of the Central Valley can receive a substantial amount of precipitation, while the southeastern desert east of the Sierra Nevada highlands typically receives very little.

CARRM essentially captures the spatial distribution of precipitation in PRISM and provides much better details than 1° E3SMv1, e.g., the local precipitation maxima in the Sierra Nevada and the Coast Ranges, and the relative dry area in the Central Valley (Fig. 5). Despite this large internal variability, it is clear the CARRM precipitation is significantly higher than observed, with the wettest year in 2015-2020 even exceeding the wettest years of PRISM/Livneh (Fig. A1g-i). One could argue, given the large interannual variability in California, that we need at least 15-20 years of baseline to determine if the CARRM's meteorology (temperature/precipitation/SWE) statistics are converged. Given that observation-based gridded products might underestimate extreme precipitation (Lundquist et al., 2019; Rhoades et al., 2023), we also use a probabilistic gridded product for daily extreme precipitation (Risser et al., 2019). The 10-year return values for the largest seasonal daily precipitation are compared in Fig. B1. Again, the return levels are much higher in CARRM. Note that we only have 20 years of simulation to estimate the parameters of the GEV distribution, and we found the extreme values weakened quite a bit using 20 years of data than using 10 years of data. Therefore, the return values of CARRM may not be robust.

We have formulated several hypotheses regarding the overestimated precipitation in CARRM. First, the wet bias is partially inherited from the large-scale biases in 1° E3SMv1. Note the larger statewide-mean precipitation (2.9 mm/day) than that in PRISM (1.7 mm/day) (Fig. 5). We also note a slightly stronger meridional moisture flux across the Coastline of California in E3SMv1 when compared to ERA5 reanalysis (Fig. E1), which may contribute to the overprediction of California precipitation.

Secondly, GCMs typically underestimate the strength and duration of high pressure blocking ridges that dominates the dry years in California (Davini and D'Andrea, 2020; Schiemann et al., 2020); this can be seen in the comparison with ERA5 (Fig. F1). Additionally, SCREAM physics likely contain their own biases (e.g., cloud microphysics) that are currently not well understood, which will be explored in future work by utilizing CARRM for atmospheric river hindcast experiments. Caldwell et al. (2009) suggested that the overestimated precipitation in California may be a common issue for physics of RCMs as reanalysis-driven RCMs tend produce more precipitation and higher relative humidity than reanalysis. The 3 km WRF hindcasts in Huang et al. (2020) did not show a wet bias, while 3 km RRM-E3SM in Rhoades et al. (2023) and 3 km / 800 m SCREAM CARRM hindcasts in Bogenschutz et al. (2024) found a wet bias especially in the Sierra Nevada. Bogenschutz et al. (2024) serves as a direct comparison to this work because we use the same code base (SCREAM) and RRM configuration; the main difference is that our simulations are not hindcasts (i.e., our boundary conditions are prescribed from

a GCM simulation). The wet bias found in Bogenschutz et al. (2024) is much weaker than our current work, suggesting that most of the bias produced by CARRM climate runs is likely due to the large-scale forcing rather than biases in the physics.

Lastly, ~3 km is a convection-permitting scale, not a fully convection-resolving scale. Unresolved processes at convection-permitting scales may spuriously accumulate energy on the effective resolution (the fully resolved scale derived from kinetic energy spectra compared to observations, ~20 km for CARRM), which can detrimentally affect the synoptic scales (Neumann et al., 2019). The convergence of convection-permitting models is suggested to require the resolution of large eddy simulations O(100 m) (Bryan et al., 2003; Petch, 2006; Langhans et al., 2012), and vertical mass fluxes at O(1-5 km) km may be too strong (Chan et al., 2012). In idealized rising thermal bubble experiments, the 900 hPa vertical velocity in non-hydrostatic SCREAM dycore at 3 hours were found to converge at 1.56 km (Liu et al., 2022). The wet bias in CARRM may reveal the insufficiency of convection-permitting resolution and suggest an even higher resolution requirement to represent convective mass fluxes more realistically.

Snowpack is the most prominent quantity to demonstrate the added value of using CARRM (when compared to the poorly resolved snowpack in the low-resolution simulations), which is represented by snow water equivalent (SWE, or water equivalent snow depth, i.e., the amount of water that would be produced by the snowpack if it were instantaneously melted) (Fig. 5). SWE reflects the variability of snow density and snow melt. The statewide mean SWE is similar for UA-SWE and WUS-SR reanalysis, as shown in the MAM and JDF averages from 1984 to 2020 water years (Fig. C1c-d, g-h), despite that WUS-SR better resolves the fine structures in the Sierra Nevada due to its ultra-high resolution (Fig. D1). WUS-SR would be a great reference for California SWE when the model resolution goes beyond 1 km in CP models. 1° E3SMv1 produces negligible SWE (SWE < 0.1 m), while CARRM essentially captures the spatial distribution of SWE in the Sierra Nevada. Note that similar to precipitation, the SWE simulated by CARRM has a positive bias when compared to UA observations.

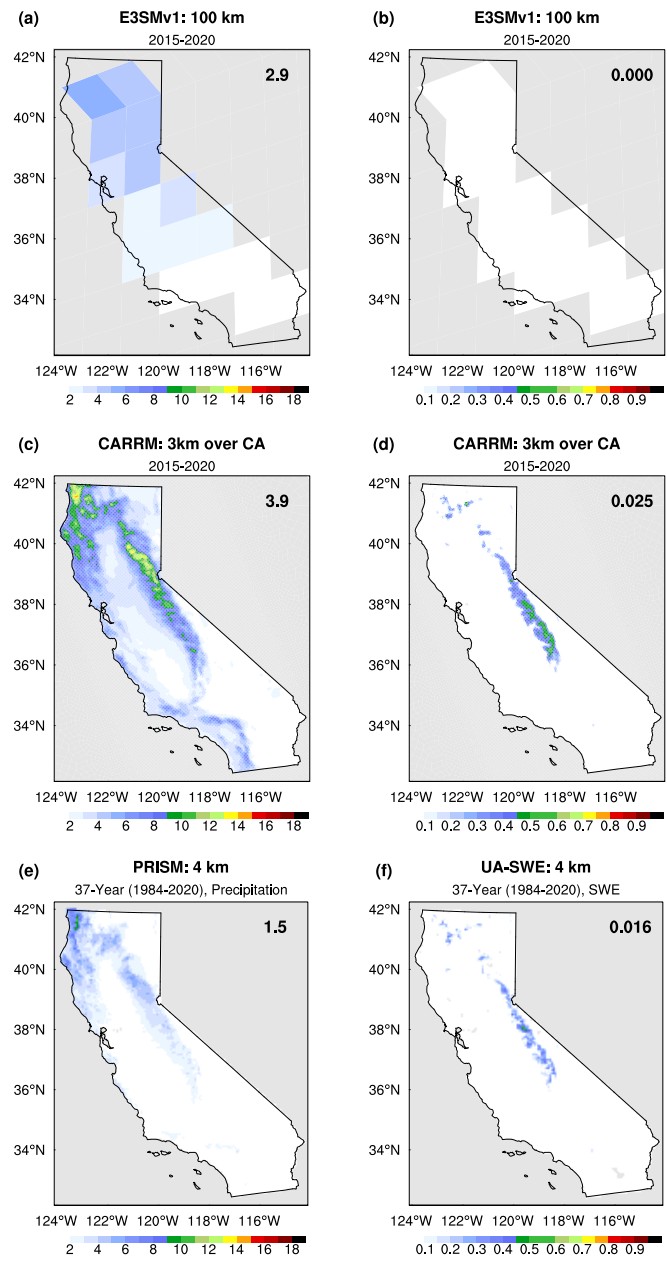

**Figure 5.** Same as Fig. 4, but for precipitation (mm/day, left column) and snow water equivalent (referred to as "SWE", m, right column). The observation-based gridded products used for comparison is PRISM (e) and UA-SWE (f). The megaton (Mt) of multi-year mean statewide SWE storage is 0.16 Mt, 10 Mt, 6.3 Mt for E3SMv1, CARRM, and UA-SWE, respectively.

### 3.2 General characteristics of the future projection

This section will present climate statistics for four time periods (2015-2020, 2029-2034, 2044-2049, and 2094-2099 water years). It will include spatial distributions of seasonal averages, statewide seasonal averages (time series), and daily intra-seasonal statistics at selected locations. The spatial distributions will highlight the seasons in which a variable of interest exhibits the most distinct patterns.

#### 3.2.1 2m air temperature

Figure 6 depicts the spatial distribution of daily maximum T2m during the summer seasons (June-July-August) for the SSP5-8.5 scenario. This figure roughly indicates a general trend in the likelihood of heat waves. In the Central Valley, daily maximum T2m are projected to rise from the current average of 36 °C to approximately 43.5 °C by the end of the century (also shown in the difference plots Fig. G1). Similarly, the Mojave/Colorado Desert is expected to experience temperatures exceeding 48 °C by the end of the century. Moreover, the Sierra Nevada is projected to undergo general warming of approximately 10 °C. The warming level of daily minimum T2m is even more prominent (not shown). For comparison, the warming level from 1981–2000 to 2081–2100 is 6-8 °C in July using a hybrid dynamical–statistical downscaling (Walton et al., 2017). By employing the definition of heat waves based on the current climate regime, e.g., three consecutive days with maximum T2m surpassing 37.8 °C, it is anticipated that nearly half of the Central Valley and California Desert will be subjected to continuous heat waves by mid-century. Moreover, by the end of the century, most of California is expected to experience prolonged periods of heat waves according to CARRM projections. The DJF daily maximum T2m in DJF is shown in Fig. H1 and Fig. I1. The warming level over the Sierra Nevada is about 9 °C in DJF. This is expected to have a significant impact on snow.

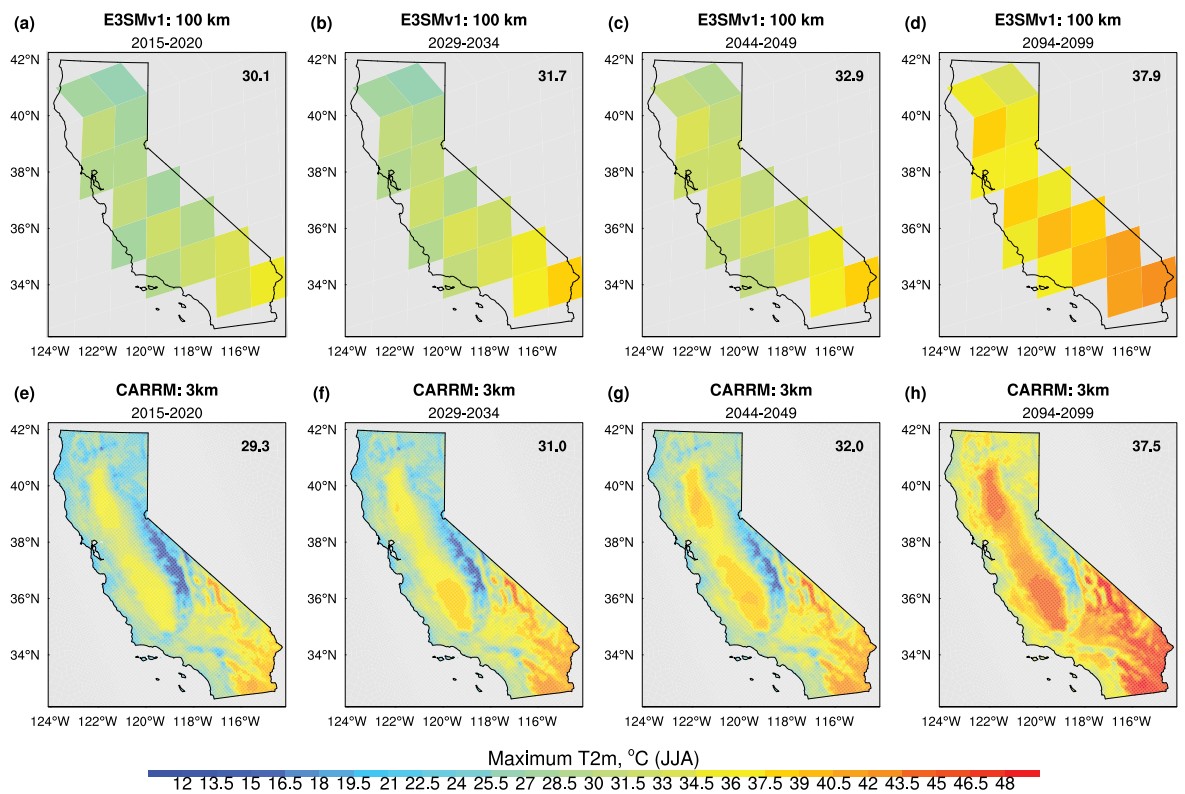

**Figure 6.** Multi-year summer average of daily maximum T2m in 2015-2020, 2029-2034, 2044-2049, 2094-2099 water years (from left to right columns) simulated by 1° E3SMv1 (top row) and SCREAMv0 CARRM (bottom row). The statewide average is shown in the right-top corner.

The statewide-average T2m is essentially inherited from 1° E3SMv1 (Fig. 7). The response of statewide-averaged T2m to GHGs is very clear. Across all seasons, there is a consistent and monotonic increase in daily maximum, mean, and minimum T2m over time. Of particular note is that during the summer season, statewide-average daily maximum T2m can approach nearly 40 °C, while daily mean T2m can rise to 20 °C from spring to autumn. This prominent warming is expected to have severe implications for California's agriculture. For example, given that the growth of wine grapes typically commences at around 10 °C, such substantial warming could lead to a pronounced advancement in average grape ripening period and a decline in overall quality (Hayhoe et al., 2004). Even more importantly, extreme temperature and humidity associated with

climate change has a great impact on human survivability, especially for older populations that work in agriculture (Vanos et al., 2023).

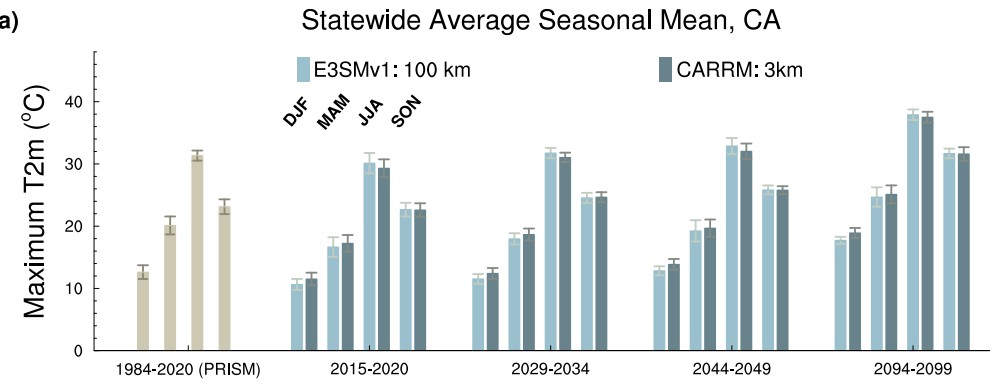

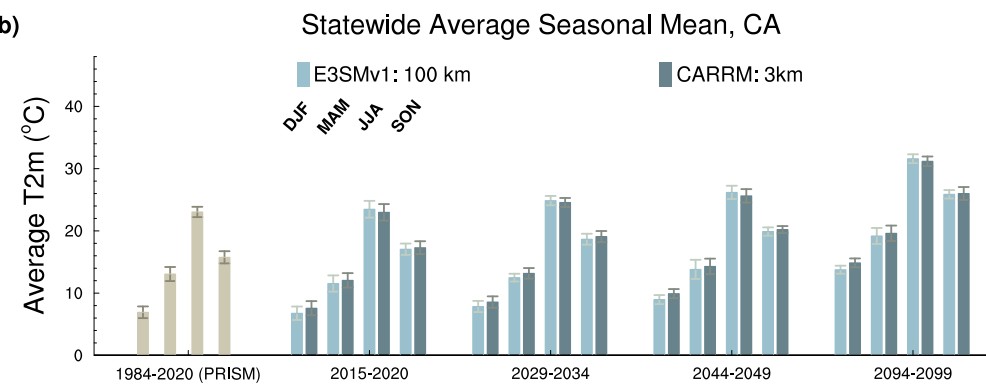

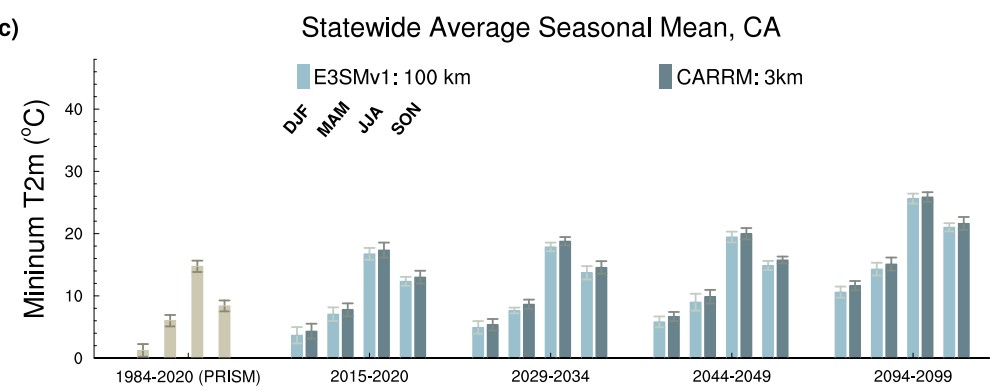

**Figure 7.** Five-year seasonal average and standard deviation of daily (a) maximum, (b) mean, and (c) minimum T2m during four simulation segments). SCREAMv0 CARRM (1° E3SMv1) is denoted by dark (light) blue histograms. Each segment shows winter (December-January-February, DJF), spring (March-April-May, MAM), summer (June-July- August, JJA), and autumn (September-October-November, SON) in order. PRISM (yellow histograms) from 1984 to 2020 water years is shown in the leftmost column for a baseline comparison with the 2015-2020 simulation period.

While CARRM may return essentially the same result in terms of statewide mean temperature statistics, the superior representation of spatial distribution allows one to examine temperature trends at specific locations. As an example, we compared four representative locations for their daily statistics: Sacramento (a point in Central Valley), Death Valley (one of the hottest points in Mojave Desert), Tahoe City (a city representative of the High Sierra), and San Francisco (a major city in the Bay Area, typically subjected to the marine layer) (Fig. 8, 9). Box plots give the minimum, lower quartile, median, upper quartile and maximum of daily samples for each segment per season, with a sample size of ~450 (5 year x 3 month x 30 day) samples per box. Overall, the distribution of daily maximum T2m in the CARRM baseline is very consistent with GHCN in-situ observation and PRISM gridded reanalysis, while CARRM shows a general warm bias in daily minimum T2m.

Though the overall warming trend is comparable between 1° E3SMv1 and CARRM, CARRM can better differentiate temperatures across geographical locations. For example, the daily maximum T2m in Death Valley is 15-20 °C higher in CARRM than in 1° E3SMv1, while the daily minimum T2m in Tahoe city is 5-10 °C lower in CARRM, representing a wide range of temperature spatiotemporal variability across California landscapes in Death Valley and the Tahoe city, respectively. This discrepancy directly reflects the influence of topography and elevation differences. As 1° E3SMv1 cannot resolve such topographical details, the contrast in daily maximum T2m between Death Valley and Tahoe city is smoothed out (Fig. 1c).

The local variations of temperature are better captured by CARRM. For example, in CARRM, although the maximum daily T2m in summer is similar between Sacramento and Death Valley (rising from 45 °C at present to nearly 60 °C by the end of the century), the mean daily T2m is approximately 10° higher in Death Valley compared to Sacramento. It's alarming that 60 °C would be substantially higher than the historical all-time record reached this past year (which is about 56.67 °C). Note that the record of daily maximum T2m in the GHCN observational data in Fig. 8d is 54.4 °C during the 1989-2020 water years. This indicates that the daily temperature variability in Death Valley is relatively small, implying a much warmer body temperature one could feel in Death Valley.

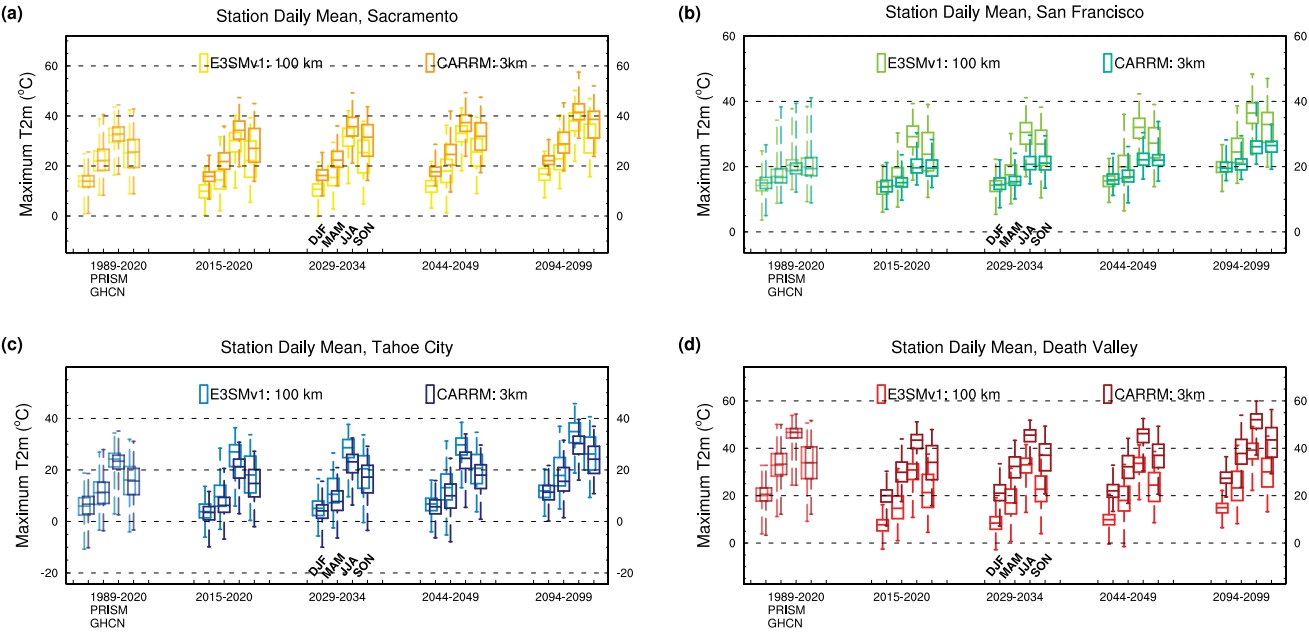

**Figure 8.** Daily maximum 2-meter temperature statistics for different seasons and different segments in (a) Sacramento (yellow), (b) San Francisco (green), (c) Tahoe city (blue), and (d) Death Valley (red). Each box gives the minimum, lower quartile, median, upper quartile and maximum, with a sample size of ~450 (5 year x 3 month x 30 day). The order of seasons in each segment is winter, spring, summer and autumn. The light color of each pair of boxes indicates 1° E3SMv1 and the dark color indicates SCREAMv0 CARRM. The in-situ GHCN observations (darker color) and the observation-based gridded product PRISM (lighter color) from 1989 to 2020 water years are shown in the leftmost column for a baseline comparison with the 2015-2020 simulation period.

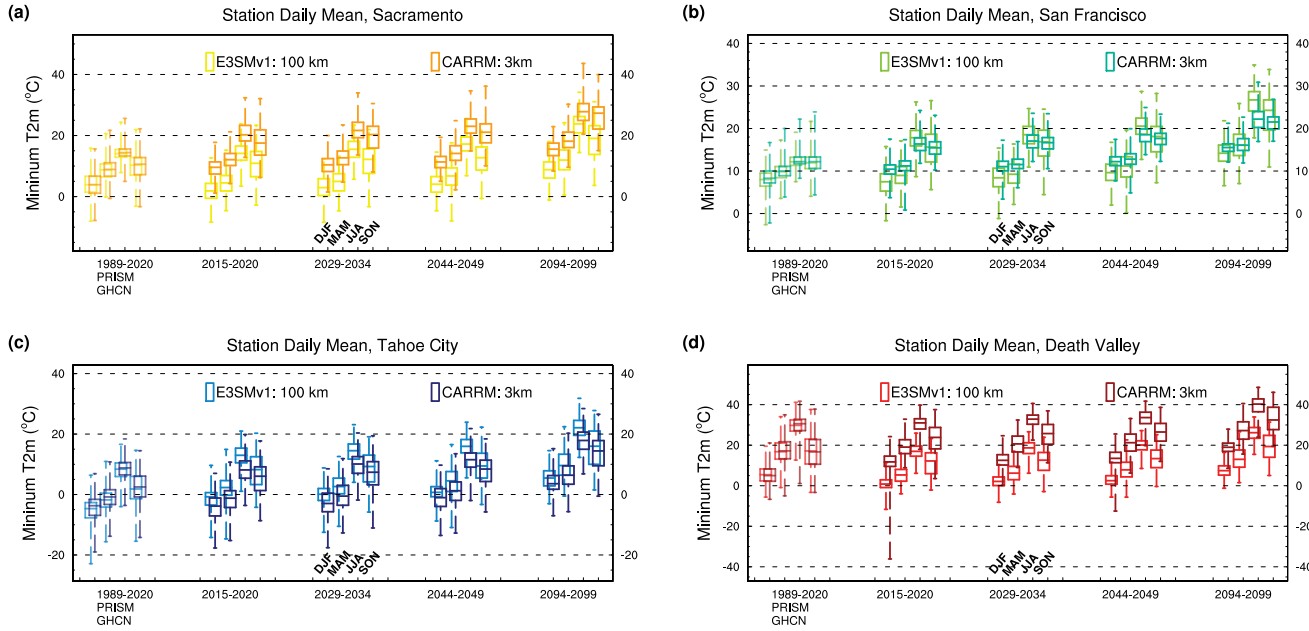

**Figure 9.** Same as Fig. 8 but for daily minimum 2-meter temperatures.

### 3.2.2 Precipitation

The spatial variability of winter precipitation (December-January-February) is shown in Fig. 10. As we found that CARRM has a wet bias when compared to observations, the key takeaway from future projection simulations using CARRM lies in the relative trends rather than absolute magnitudes. In our simulations, the signal of the forced response of precipitation to GHG in California remains obscure during the first half of the century, but shows a significant increase towards the end of the century. Note that the sign of precipitation change is the same in 1° E3SMv1 and CARRM, but the magnitude is amplified along terrain in CARRM.

Regarding the spatial distribution, the two segments before mid-century show contrasting changes across different regions in CARRM: precipitation in the Sierra Nevada is weaker compared to the baseline period (particularly up to 3 mm/day less during 2044-2094), while the western Northwest Coastal Range experiences an increase in precipitation (up to 2-3 mm/day). In addition, the Transverse-Peninsular Ranges in southern California exhibit drier conditions than the baseline during 2029-2034, while they receive more rainfall than the baseline during 2044-2049. By the end of the century, under this scenario, the majority of California may experience a significant increase in precipitation except for the southern Sierra Nevada and the southernmost desert of California. Compared to the baseline period, annual total precipitation is projected to increase by 30% in the northern, eastern, and southern Ranges (Fig. J1). Some areas in the Great Basin Desert are projected to received more than 50% of annual total precipitation. The Central Valley is expected to increase by 0-24% of total annual precipitation. In contrast, the signals of Transverse-Peninsular Ranges, Great Basin Desert and Mojave Desert are very weak in 1° E3SMv1.

Note that the 5-yr average hardly reflects the ENSO signal. For example, the 2029-2034 segment contains an extremely strong El Niño year followed by a strong 3-yr La Niña event and thus its overall impact on California precipitation may largely cancel out. However, we did not see a significant modulation of the ENSO signal on precipitation even upon examining monthly precipitation. Towards the end of the century, heavy precipitation events occur at least once per year (not shown). We note that the spatial pattern of ENSO in the E3SMv1 historical ensemble is not sufficient along the North American coast (Golaz et al., 2019). In CESMv1-LE, a high correlation between ENSO and the Pacific–North American pattern/east Pacific pattern was identified, but it was also noted that considerable variability remains within the midlatitude dynamics that cannot be attributed to ENSO influences alone (O'Brien and Deser, 2023). This suggests a notable chance of failed hydroclimate responses in the western U.S. to ENSO events in the fully coupled ensemble. As the prescribed SSTs in CARRM were derived from fully coupled E3SMv1 projections, some of the effects of air-sea interactions have been included, whereas the interactions at fine scale is not represented here.

As the baseline five years of CARRM future projections are not hindcasts (i.e., the forcing data are not from reanalysis/observations), they are not suitable for comparison with individual extreme events in observations as was done in Huang et al. (2020) and Rhoades et al. (2023). Bogenschutz et al. (2024) simulated and evaluated representative AR events using SCREAMv0 CARRM under the hindcast framework. The model performance and sensitivities of the RRM configurations are discussed in detail in that work.

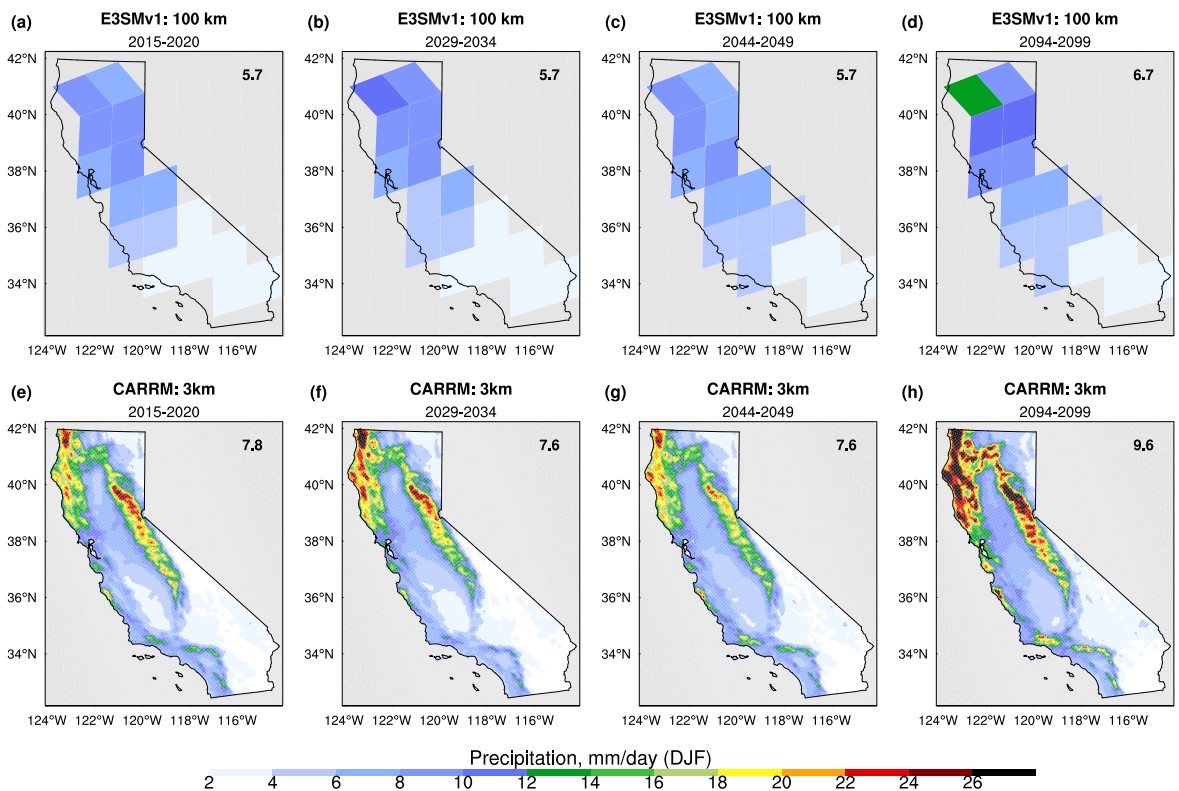

**Figure 10.** Same as Fig. 6 but for winter precipitation.

The response of statewide-averaged precipitation to GHGs is not as clear as T2m, which is not surprising (Fig. 11a). In contrast to temperature, the parameterizations of precipitation processes involve a higher number of assumptions and exhibit increased inter-model variability. Additionally, precipitation displays greater spatial inhomogeneity, even in the absence of topography. Note that we nudged temperature, humidity and horizontal winds in the coarse outer domain, so temperature is directly constrained by the low-resolution simulations, but precipitation can still be significantly different with the constrained atmospheric conditions.

Unlike temperature, the statewide-average precipitation is consistently higher in CARRM compared to 1° E3SMv1. This discrepancy of precipitation (especially in winter) shows a non-stationary increase over time (Fig. 11a). This exemplifies the model differences, as well as the potential issues with the model physics, such as the wet bias seen in the comparison with observations (Fig. 5). It is important to note that SON precipitation decreases with time. This is significant because despite the

relatively modest contribution to annual precipitation, SON is historically the most active period for wildfires in CA, therefore precipitation is crucial during this season to dampen the worst impacts (Swain, 2021). This is also consistent with recent

observational evidence and multi-model analysis. For example, Goss et al. (2020) showed that decreases in California SON precipitation over the past 40 years have led to increases in fire weather indices, while Luković et al. (2021) provided evidence of a significant decrease in November precipitation in California, and CESM large ensemble, CMIP5 and NA-CORDEX all found that "shoulder season" precipitation is likely to decrease by mid-century (Swain et al., 2018; Dong et al., 2019; Mahoney et al., 2021).

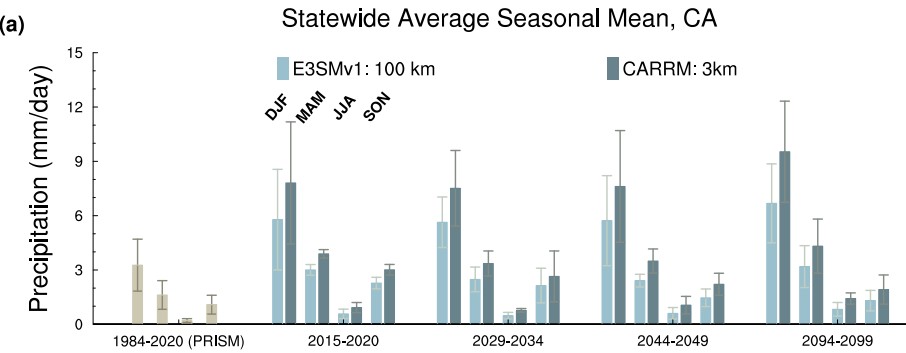

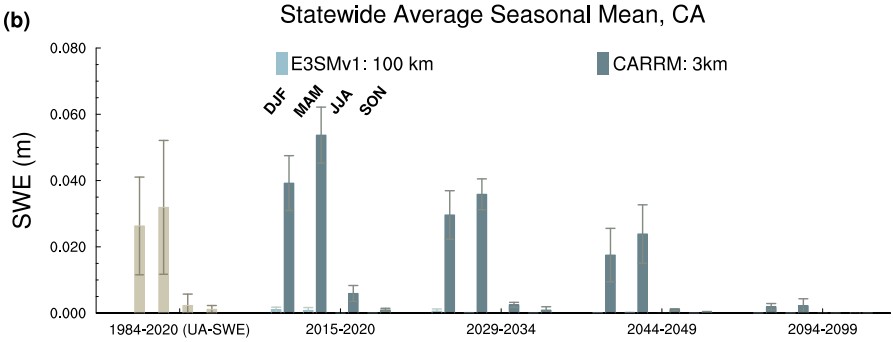

**Figure 11.** Same as Fig. 7 but for (a) precipitation and (b) SWE. The observation-based gridded products used for comparison is PRISM (a) and UA-SWE (b).

Despite not receiving as much attention as winter precipitation for California, summer precipitation (JJA) also appears to increase towards the end of the century. We noticed a few mesoscale convective system-like convective systems that can originate locally or propagate into California from the east during the summer, especially at the end of the century (not shown). They are characterized by prominent longwave radiative cooling which can rival the magnitude of mesoscale convective systems and tropical cyclones. This pattern is partially depicted in the 5-yr averaged JJA precipitation, especially over the Sierra Nevada

(Fig. 12). Unlike DJF, JJA precipitation at the end of the century does not exhibit a distinct topographic precipitation signature along the mountain range. Instead, it shows local extremes at a few specific locations. The small area and significant gradient of these precipitation hot spots may indicate a series of highly intermittent but intense organized convective systems.

The primary source of summer precipitation in Southern Desert is the southwest monsoon (Adams and Comrie, 1997; Prein et al., 2022). The monsoon contributes up to 45% of the annual precipitation in the southwest desert (Higgins et al., 1999) and can trigger severe weather events such as lightning, thunderstorms, wildfires, and floods (Nauslar et al., 2018; Griffiths et al., 2009). By the end of the century, summer precipitation is generally projected to increase by 10-20% of annual precipitation over most of the Southern Desert (Fig. K1). The notable increase in precipitation over the Southern Desert may be associated with an amplified temperature gradient and increased moisture transport from the Gulf of California (Jana et al., 2018; Johnson and Delworth, 2023). In addition, since the monsoon season is characterized by intense localized thunderstorm activity, accurate monsoon simulations require models that capture the spatial heterogeneity of temperature and precipitation. Specifically, some thunderstorms are triggered by local temperature extremes near the surface in tandem with increased humidity in the Southern Desert. The higher resolution provided by RCMs has been found to impact the quantification of various mechanisms of the North American monsoon warming response (Meyer and Jin, 2016).

Given that precipitation in California is primarily influenced by large-scale processes such as atmospheric rivers and mid-latitude cyclones, the diurnal cycle is not as significant a consideration as it is in the central Great Plains. However, as with other GCPMs, CARRM's host model SCREAMv0 captures diurnal cycles that are generally consistent with observations (Caldwell et al., 2021). Though, we do recognize that studying the diurnal cycle of precipitation in California during summertime monsoon events over the Sierra Nevada and southeastern portion of the state could warrant some investigation in the future.

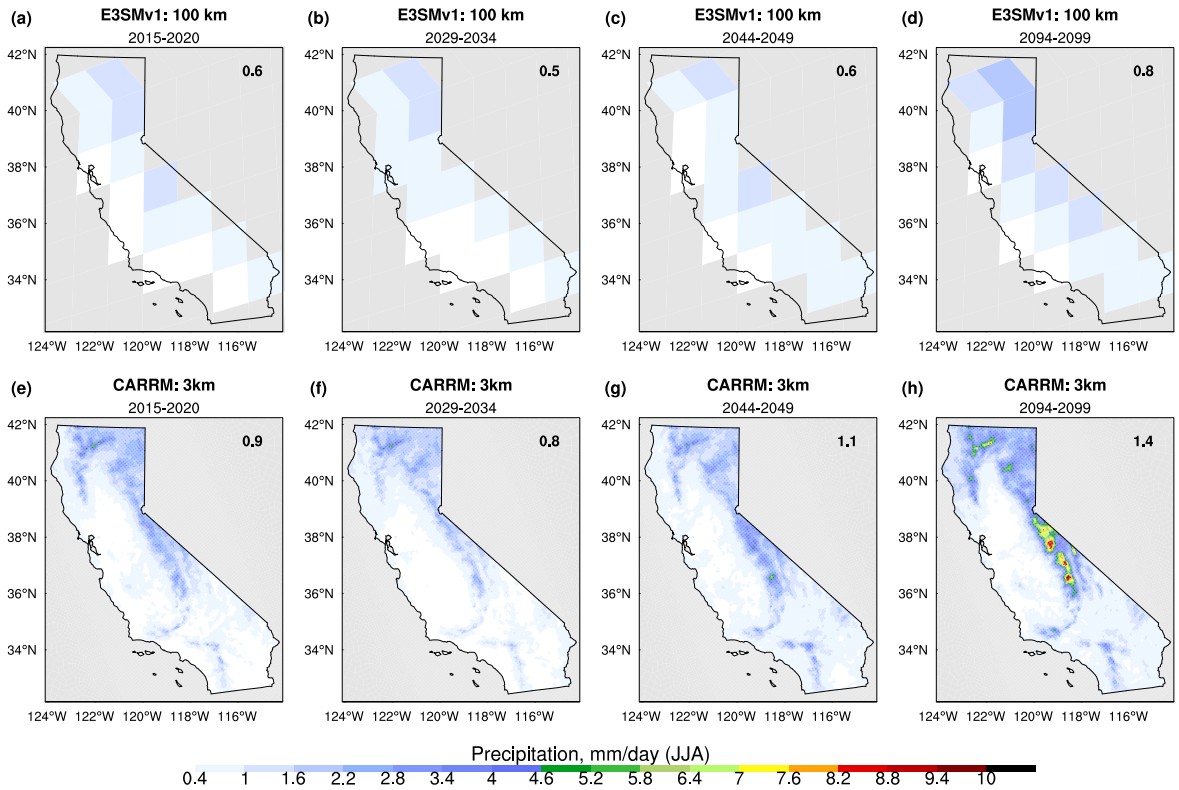

**Figure 12.** Same as Fig. 6 but for summer precipitation.

### 3.2.3 SWE

In the Sierra Nevada, SWE is typically thickest during the spring season (March-April-May) (Fig. 13). SWE serves as a compelling indicator that highlights the benefit of high resolution, as 1° E3SMv1 fails to represent SWE in the Sierra. This is particularly evident in the California average SWE (Fig. 11). Furthermore, SWE is expected to be one of the variables most significantly impacted by GHG forcing. California is projected to be essentially devoid of snow by the end of the century (Fig. 11), except for scattered areas in the central Sierra Nevada (Fig. 13). Note that unlike precipitation, which showed minimal

changes until the end of the century, SWE exhibits a clear decline by mid-century. A local warming of 6 °C can greatly affect the majority of SWE in the Sierra Nevada (Bales et al., 2015), so it is not surprising that such a pronounced decline in SWE would occur due to temperature changes (Figs. 7, 9, 10). The response of snow sensitivity to warming in the Sierra Nevada is consistent with recent works (Berg et al., 2016; Rhoades et al., 2017, 2018a; Sun et al., 2019; Siirila-Woodburn et al., 2021).

They found that under the impacts of climate change, California and the western U.S. will experience significant reductions in SWE, including reduced winter snowfall, and earlier spring snowmelt.

Given that SWE contributes approximately 3/4 of the annual freshwater supply for the western United States (Palmer, 1988; Cayan, 1996; Bales et al., 2011), the retreat of SWE by mid-century will have significant implications for water management throughout California. Consequently, this will impact agriculture yields and energy supplies (Rhoades et al., 2017; Belmecheri et al., 2015). Additionally, the shortening of the snow season and early snowmelt are closely linked to fire activity, as this would lead to drier soils and vegetation, and thus will increase the wildfire frequency and extend the fire seasons (Westerling et al., 2006; Holden et al., 2018). Lastly, the complete recession of SWE is anticipated to have a substantial impact on California's ski industry. The start of a typical ski season requires snow depths above 2-4 ft (with a corresponding SWE threshold of ~0.2 m) (Hayhoe et al., 2004; Hill et al., 2019), despite that snow-to-liquid ratio can vary substantially from season to season and across mountain regions, especially in maritime vs continental mountain ranges.

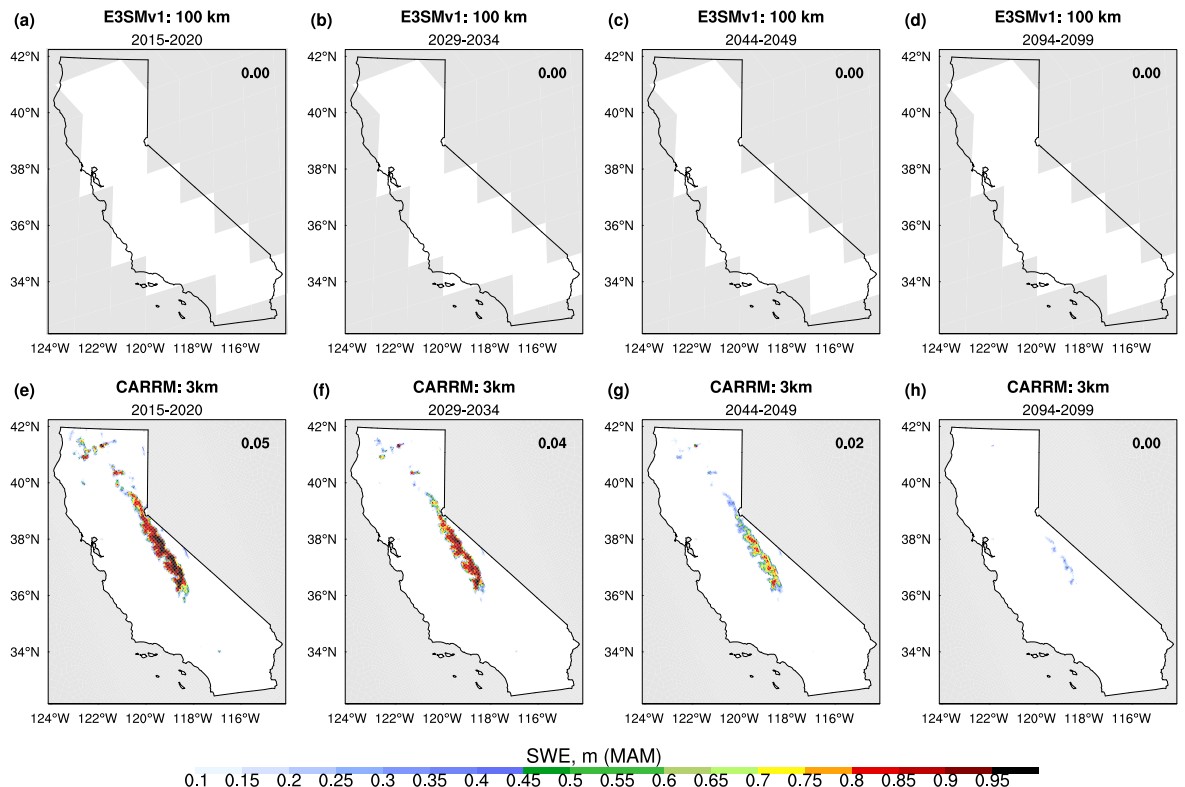

**Figure 13.** Same as Fig. 6 but for spring SWE.

To further investigate the response of SWE at different latitudes in the Sierra Nevada and to demonstrate the added value of CARRM in simulating SWE, we selected four specific locations: Adin Mtn, Truckee, Tahoe City, and Leavitt Lake. The elevations recorded in SNOTEL are 1886.7, 1983.9, 2071.7, 2927.3 m, respectively. We compare their climate statistics using monthly-mean SWE (Fig. 14, a sample size of 15 (5 year x 3 month) per box), because we did not output daily SWE for the re-run of E3SMv1. Figures 14a-d also show the monthly mean statistics for the in-situ SNOTEL observation and the observation-based gridded product UA-SWE from 1989 to 2020 water years. In addition, Fig. 14e shows the daily mean annual cycle of SWE simulated by CARRM. The observed annual cycle of SWE, the associated daily maximum/minimum T2m, and precipitation during water years 1989-2020 are shown in Fig. 15.

First, it is reconfirmed that 1° E3SMv1 has essentially no ability to simulate SWE, as depicted by the light blue box in Fig. 14a-d, where SWE simulated by E3SMv1 is consistently close to zero. Second, CARRM has biases relative to the in-situ

SNOTEL observation and the observation-based gridded product UA-SWE. Note that UA-SWE also has a dry bias (0-0.5 m)
       relative to SNOTEL (Fig. 14a-d, Fig. 15g,h). Interestingly, while the CARRM-simulated statewide mean SWE is significantly
       higher than UA-SWE (Fig. 5d,f), SWE at Adin Mtn has a dry bias relative to UA-SWE (Fig. 14a). Compared to SNOTEL,
       CARRM has a lower SWE for all stations except for Tahoe City, which has a higher SWE. The SWE bias in CARRM may be
       related to temperature and precipitation biases. Finally, as expected, the distribution and variability of SWE is influenced by
elevation. Leavitt Lake, characterized by the highest elevation, has the largest observed SWE. CARRM predicts that Leavitt
       Lake will still have 0.5 m winter SWE and 0.7 m spring SWE by mid-century (Fig. 14d). The snowmelt response is fastest in the
       spring, as shown in the observations (Fig. 15g,h) and the CARRM simulations (Fig. 14e). The rate of snowmelt is proportional
       to the SWE of each station. Snowpack retreat by the end of the century is significant at all four sites examined in the CARRM
       simulations.

We emphasize the substantial reduction in summer (June-July-August) SWE projected at all locations, and the complete
       absence in some cases. This would have significant implications for increased wildfire threats (i.e. more frequent wildfires and
       a much longer wild fire season) (e.g., Westerling et al., 2006; Holden et al., 2018).

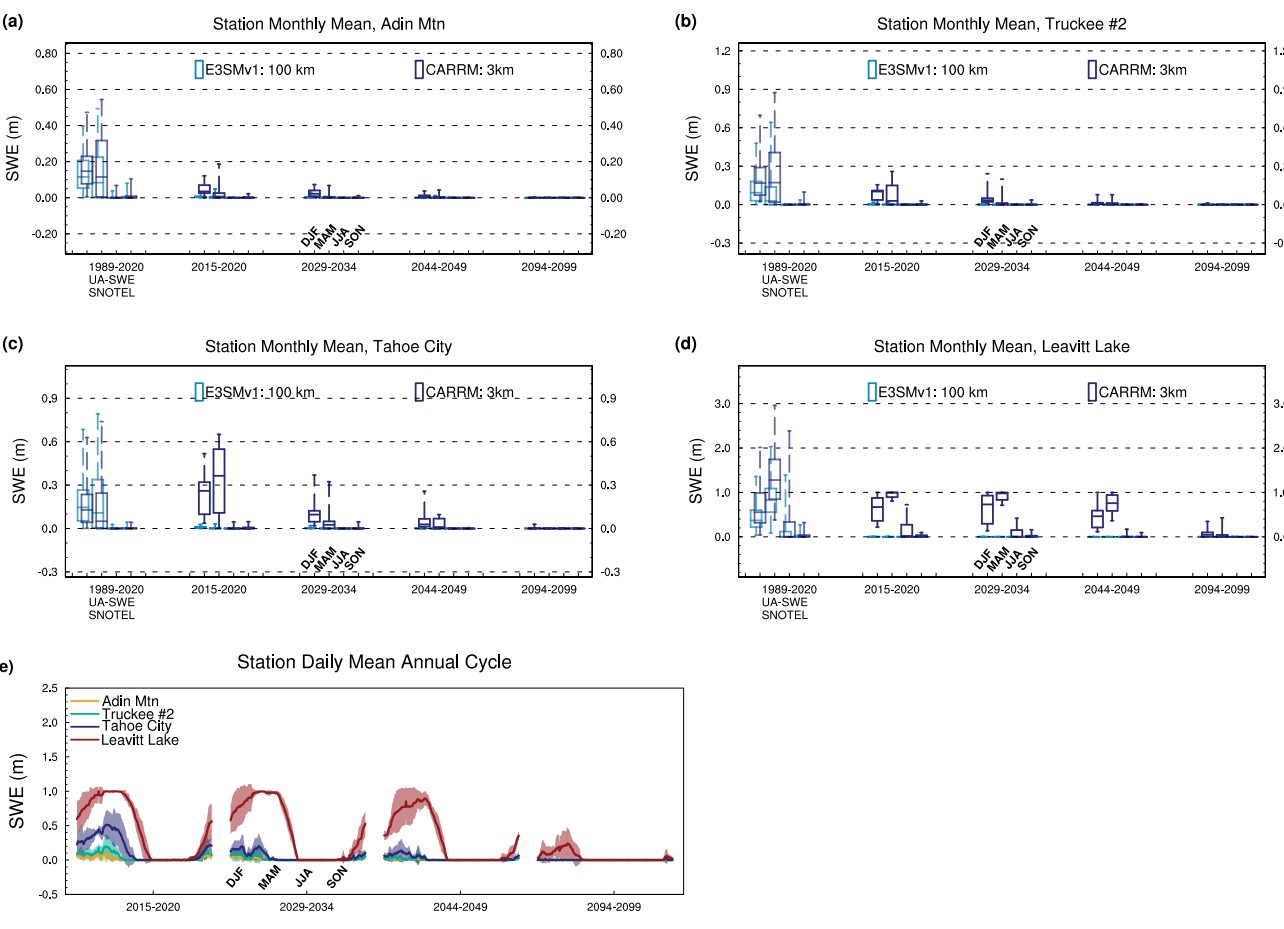

**Figure 14.** Same as Fig. 8 but for monthly mean of SWE in (a) Adin Mtn, (b) Truckee, (c) Tahoe City, and (d) Leavitt Lake. (e) Daily mean annual cycle at four stations. The shading shows the standard deviation of each segment.

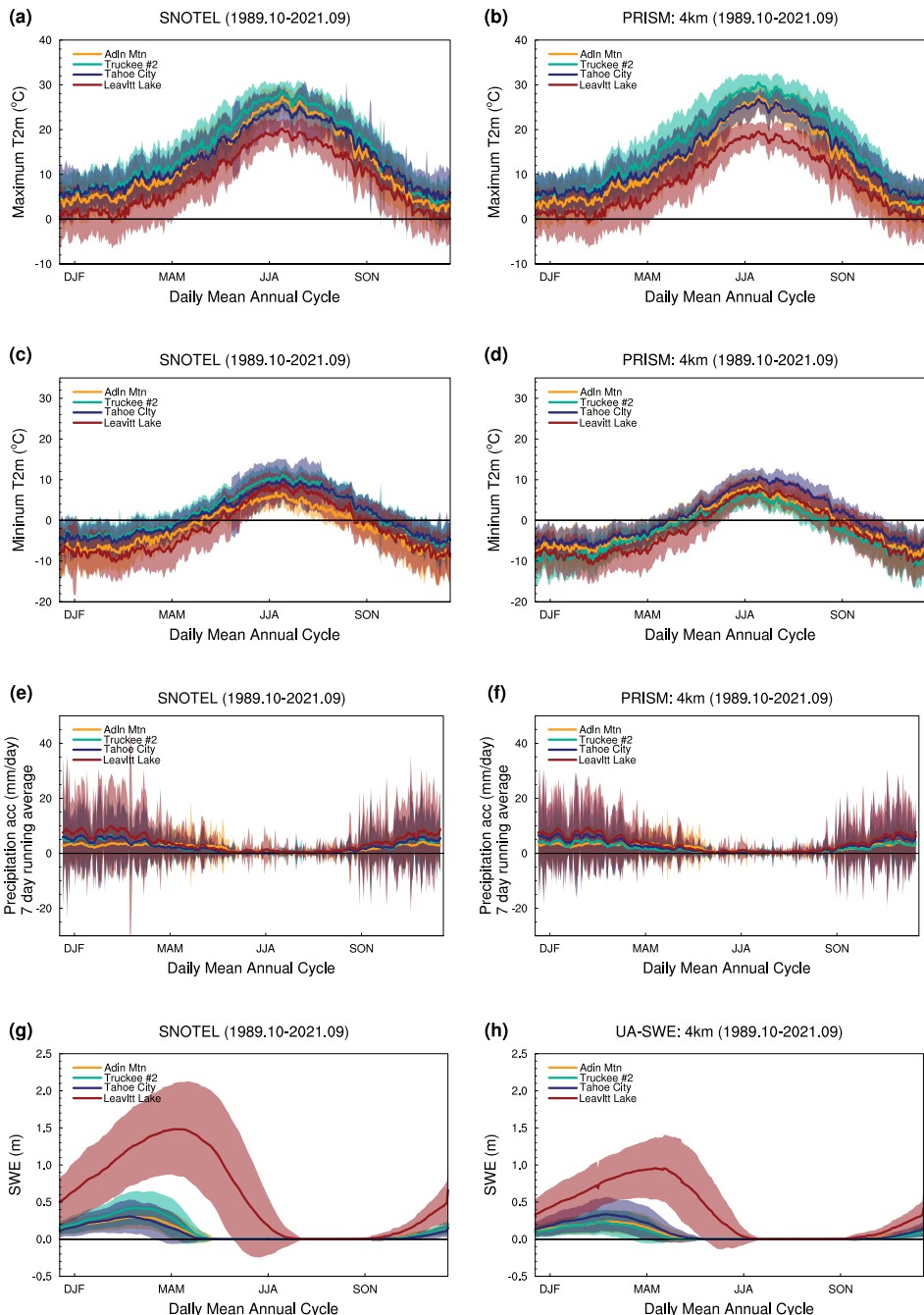

**Figure 15.** Comparison of in-situ SNOTEL observations (left column) and 4 km observation-based gridded products (right column). It shows the daily mean annual cycle for (a)-(b) maximum T2m, (c)-(d) minimum T2m, (e)-(f) 7-day running average precipitation, and (g)-(h) SWE at Adin Mtn (yellow), Truckee (green), Tahoe City (purple), and Leavitt Lake (dark red). In the right column, PRISM is used for T2m and precipitation, and UA-SWE is used for SWE.

### 3.2.4 Marine Stratocumulus

Along the west coast of California fog plays a crucial role in maintaining the redwood ecosystem, helps to moderate hot summer temperature influenced by the coastal Mediterranean climate, and increases humidity to help curb wildfire ignitions (Lewis, 2003; Johnstone and Dawson, 2010). A major mechanism for the formation of coastal fog is strong large-scale subsidence near the coast pushes low-level inversions near the surface acting to lower the base of marine stratocumulus clouds (O'Brien et al., 2012; Koračin et al., 2001). Note that while coastal fog lies within the 3.25 km mesh, the California stratocumulus found upstream over the ocean falls within the transition region. Nevertheless, SCREAM's turbulence scheme (SHOC) is scale aware and should be able to properly parameterize the maritime low clouds across resolutions (Bogenschutz et al., 2023).

The lack of marine stratocumulus is a common issue in low-resolution GCMs, adding to the uncertainty of shortwave cloud feedback. The improved marine stratocumulus is a great achievement of the SCREAM global 3.25 km simulations (Caldwell et al., 2021), which is partially due to higher horizontal and vertical resolution (Lee et al., 2022; Bogenschutz et al., 2022). In our CARRM baseline (2015-2020), the shortwave cloud radiative forcing ($SWCF = FSNTOA - FSNTOA_C$, where FSNTOA is net solar flux at TOA, $FSNTOA_C$ is clear-sky net solar flux at TOA) is greatly improved over inland areas (Fig. L1). However, it is also worth noting that near the western edge of the RRM domain (~100 km), the SWCF of the RRM simulation is stronger when compared to the CERES-EBAF observation.

Given the unaffordable cost of GCPMs, CARRM provides an excellent opportunity to explore the response of marine stratocumulus near California to GHGs under a convection-permitting scale (Fig. 16). The climate change signal of SWCF simulated by 1° E3SMv1 is weak. However, the SWCF simulated by CARRM is much stronger (more negative) and manifests a significant weakening over time, which indicates a decrease in stratocumulus and strong positive shortwave cloud feedback along the west coast of California. This suggests that under warming, boundary layer turbulence becomes more effective at entraining dry air from above the cloud tops. Note that the Data Ocean in CARRM uses 1° lat-lon SSTs which cannot resolve the cold coastal upwelling, which partially hampers the ability to properly capture the marine stratocumulus and coastal fog.

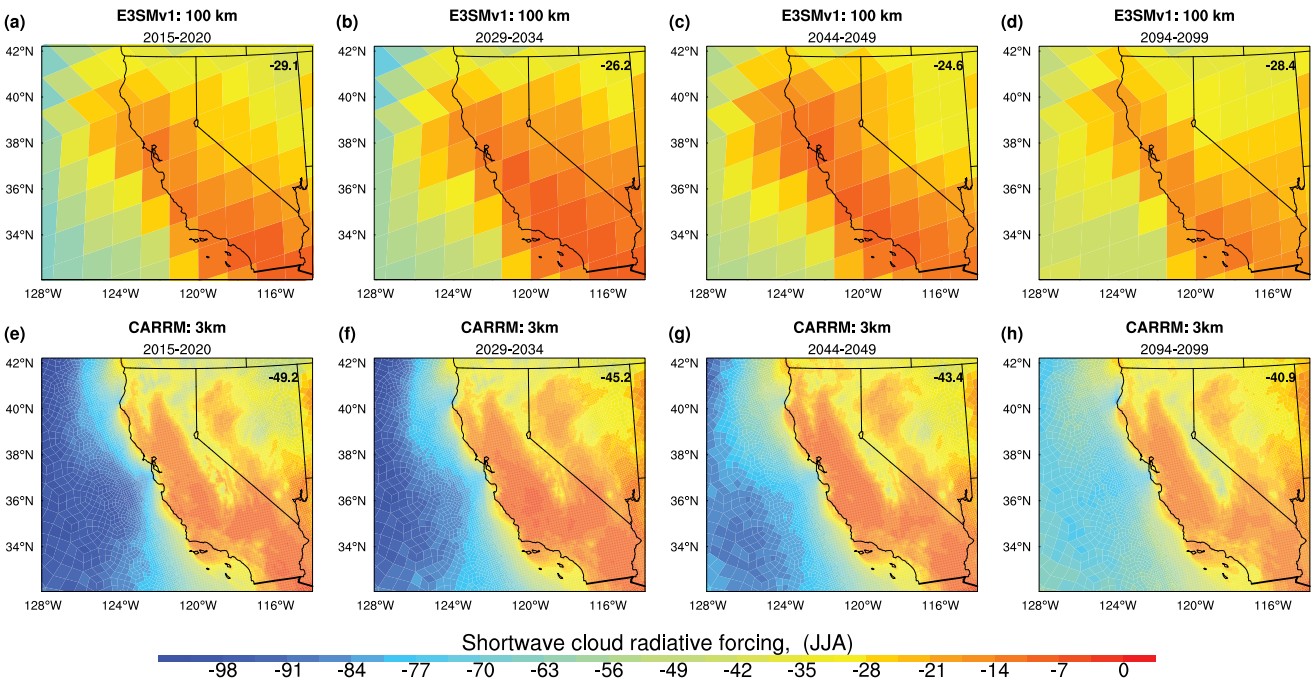

**Figure 16.** Same as Fig. 6 but for summer shortwave cloud radiative forcing.

## 3.3 Atmospheric River trends over California

ARkStorm is considered as a rare atmospheric river (AR) phenomenon transpiring once every 500 to 1000 years (Porter et al., 2011; Wing et al., 2016). ARkStorm is a hypothetical scenario that refers to a near continuous series of strong AR events capable of causing a massive flooding event similar to the Great Flood of 1862 (Engstrom, 1996; Porter et al., 2011). This storm series is estimated to have dumped 3000 mm of water in California in the 43 days from 1861.12-1862.01, triggering devastating floods that wreaked havoc across the state. A modern ARkStorm could cause $725 billion to $1 trillion in damage. Since ARs have been identified as a critical contributor to wintertime precipitation in California but can also be quite hazardous

(Ralph et al., 2006; Swain et al., 2018; Huang and Swain, 2022; Dettinger et al., 2011; Rhoades et al., 2021), we are curious about assessing the changes of ARkStorm possibility in CARRM with warming. Here, we examine the statewide 30-day precipitation and AR activity. As introduced in Methods, ARs were tracked using TempestExtremes (Ullrich and Zarzycki, 2017; Ullrich et al., 2021).

A more refined definition of the ARkStorm event in previous studies consists of two key aspects. First, it is defined based on extreme events by calculating the return period that depends on model performance rather than an absolute threshold (Swain et al., 2018). Second, considering the spatial heterogeneity across local sites, the focus is placed more on the spatial distribution rather than a statewide average (Huang and Swain, 2022). Unfortunately, we are unable to follow the first step because the calculation of return period for such an extreme event requires a large sample size, while we only have a sample of 20 years. For example, more than 1000 years of PI-control simulations and 40 multi-year ensemble numbers of future projections are typically needed. Instead, we adopted a simple approach in this work: the 30-day mean for the statewide precipitation is used to assess the possibility of ARkStorm events. The ARkStorm event is indicated by an estimated threshold (14 mm/d statewide precipitation) based on most ARkStorm studies. Since ARkStorm is too rare to capture in a small sample size, one might suggest looking at 1-20 year events instead. However, the CARRM simulations are also inadequate to answer 1-20 year events, as samples up to 20 years do not yield reliable GEV parameter estimates. On the other hand, hindcasts are useful to investigate whether CARRM has the ability to capture extreme AR events by an apple-to-apple comparison with observations, as demonstrated in Bogenschutz et al. (2024).

The statewide 30-day average or cumulative precipitation effectively diminishes the heavy-tailed distribution observed in daily or sub-daily precipitation over single sites. However, a noticeable increase in the median and upper quartile of statewide 30-day precipitation is projected at the end of the century (Fig. 17). Considering the wet bias of CARRM as shown in Fig. 17 and Fig. 5, the 14 mm/d statewide precipitation may underestimate the intensity of ARkStorm events. Nevertheless, it is evident that the possibility of end-of-century ARkStorm events is significantly increased in the realization inherited from 1° E3SMv1. More importantly, although it is not currently practical to perform multidecadal and multi-ensemble simulations directly with CARRM, Fig. 5 illustrates that CARRM provides a significantly different change in probability over time for any given extreme event reference than the low-resolution model.

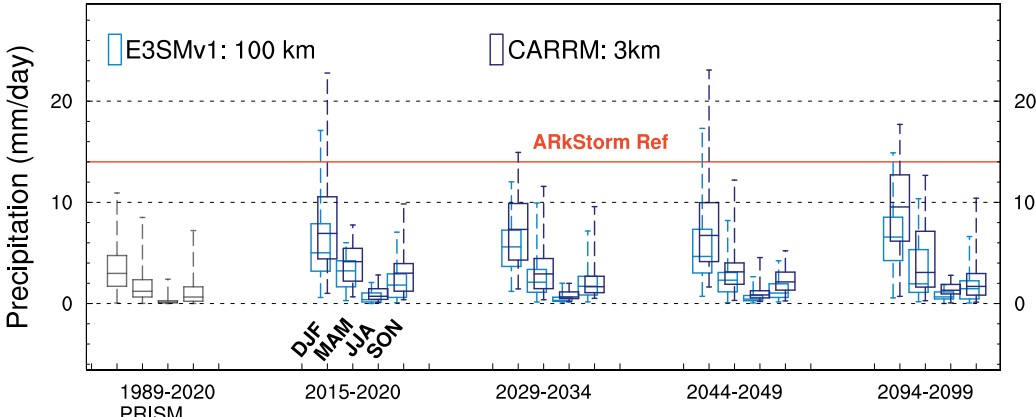

**Figure 17.** Same as Fig. 8 but for 30-day mean statewide precipitation. The observation-based gridded products used for comparison is PRISM (grey). The red line indicates the threshold corresponding to the ARkStorm event.

Consistent with the significant increase of statewide precipitation projected at the end of the century (more than 30% in winter and more than 70% in summer, Fig. 17), we also see evidence of increased AR contribution to California precipitation (about 50% in DJF, Fig. 18a). This is consistent with the increase in AR strength represented by the shift of the PDF of maximum IVT values over California toward the tail (Fig. 18b), which is also evident in the spatial distribution of IVT (Fig. M1). However, there is no clear shift in the PDFs of AR-related maximum IVT location or in the number of AR-related IVT samples making landfall in California (not shown). The low-level winds that shape AR latitudinal variability are quite similar across the four segments (Fig. M1).

As the climate warms, the PDF of the precipitable water (increased by 36%) shifts towards its tail, consistent with higher extreme IVT under warming (Fig. 18c). The overall PDF of 850 hPa zonal wind is projected to experience a small leftward shift with minimal change in shape (Fig. 18d). Precipitable water is controlled by the Clausius-Clapeyron relation, which imposes that a warmer atmosphere can contain more water vapor. The slight leftward shift in the 850 hPa zonal wind in the PDF indicates an overall weakening of the westerlies, which may slightly reduce the frequency of AR hitting California. The

differences between 1° E3SMv1 and CARRM are noted. CARRM produces larger AR contribution to California precipitation,
stronger AR intensity, and weaker westerlies.

In our simulations, the large increase in total precipitation in California by the end of the century is primarily due to larger
amounts of precipitation falling from stronger rather than more frequent moisture surges hitting California, which is dominated
by larger precipitable water under the significant warming scenario.

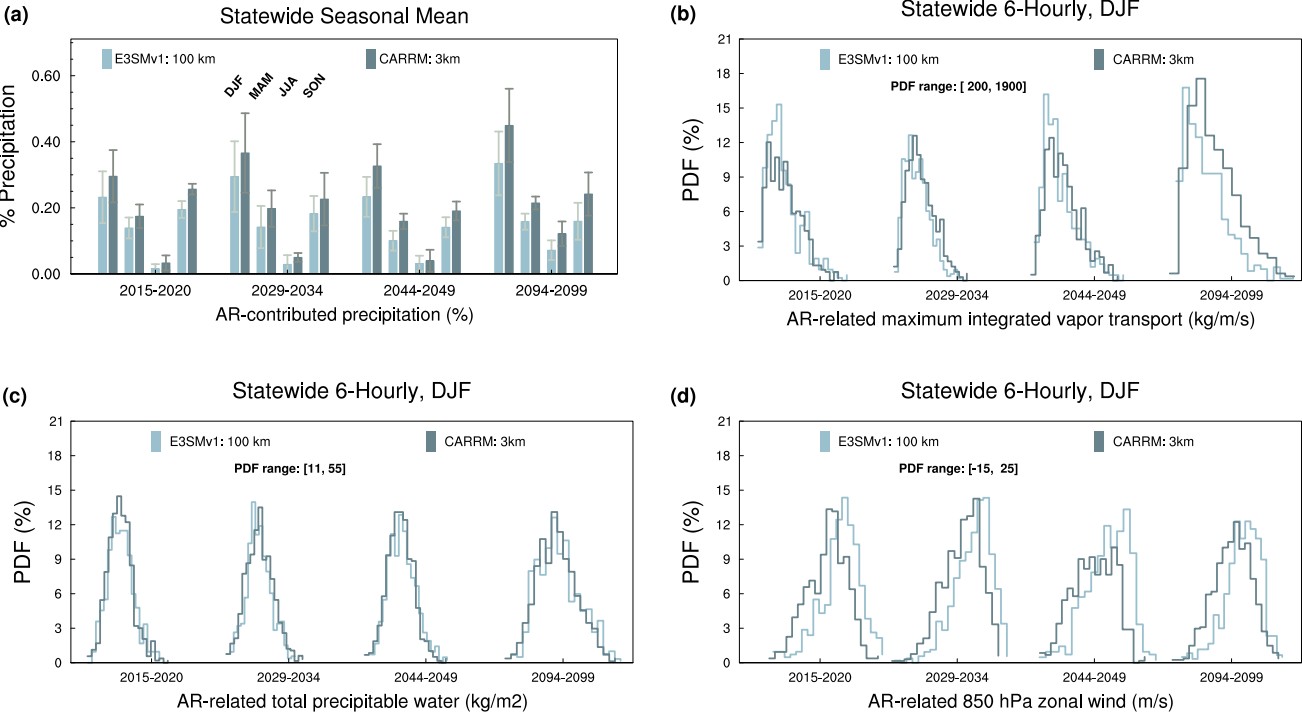

**Figure 18.** Statewide atmospheric river (referred to as "AR") related statistics. (a) Seasonal mean AR contribution to California for each
season in four segments. SCREAMv0 CARRM (1° E3SMv1) is denoted by dark (light) blue histograms. Each segment shows winter (DJF),
spring (MAM), summer (JJA), and autumn (SON) in order. PDFs of AR-related (b) maximum integrated vapor transport, (c) total precipitable
water, and (d) 850 hPa zonal wind in DJF. The range of PDF values for each variable is shown in bold.

## 4 Discussion and Conclusions

This work marks the first time SCREAM has been used for climate length simulations, which was only made possible by leveraging RRM. Our RRM is centered on California and includes parts of the West Coast at a resolution of 3.25 km and 1° resolution covering the remainder of the globe. We evaluated California's future projections under the highest emission scenario by selecting four 5-yr time periods.

To produce CARRM simulations in this study, we first established a California-specific RRM framework. This involved designing the new RRM grid and generating the necessary model configurations. Thanks to the development efforts and documentation provided by the E3SM RRM community, the tool chains and workflows for generating new RRM grids are relatively mature. Then, we nudged CARRM to 1° E3SMv1 SSP5-8.5 scenario and generated future projections for California.

Unlike our work, which was nudged from an E3SMv1 simulation, one may argue that it would be desirable to run RRM freely with an active deep convection scheme. This would indeed avoid the necessary step of having to rerun the E3SM model to generate the forcing at the time scales needed. An advantage of RRM over regional climate models lies in its seamless transition from typical GCM resolution to the finest resolution. However, running RRM freely requires a scale-aware deep cumulus parameterization, which is currently lacking in SCREAM for a proper handling of the transition from 100 km to 3.25 km. Hence, we adopted a nudging strategy to force RRM with a low-resolution GCM.

There are several advantages by adopting nudging in our work. By utilizing known boundary conditions (atmospheric state, SST, sea ice), we can pre-select years with extreme phasing of climate modes of variability (e.g., ENSO) as simulation segments, thereby expanding the range of sampling. Furthermore, instead of strictly following a chronological order, we can simulate several segments simultaneously by nudging to the target state. This greatly reduces simulation time as well as wall-clock time (i.e. the ability to run seperate periods in parallel) and expedites data delivery and model validation. Finally, since we are nudging from an E3SMv1 scientifically validated simulation, we are not subjected to time consuming and tedious tuning efforts that would be required in a free-running simulation to ensure top-of-atmosphere radiation balance and potential issues with a drifting climate.

CARRM represents a very efficient configuration compared to the global 3.25 km SCREAM (249 simulated days per day compared to 4 to 5 simulation days per day, respectively, with approximately one third of the computational cores used) and serves as a powerful tool for studying climate change and resilience in California. With its complex topography and coastline, California is a microclimate-rich region, characterized by significant spatial heterogeneity. Therefore, high-resolution modeling becomes essential to capture the complexities associated with California climate. The convection-permitting scale has manifested great values in accurately representing the highly volatile storm-induced precipitation in winter. The Sierra Nevada snowpack, which holds the lifeblood of California's water resources, relies heavily on high-resolution representation in climate models. Thus, California provides an excellent test bed for SCREAM-RRM climate framework.

By comparing to 4 km observation-based gridded products and in-situ observations, the baseline climate of CARRM demonstrates the significant added value of the 3.25 km resolution for California. In particular, it accurately captures high temperatures in the Central Valley, and realistically depicts the spatial distribution of rainfall and snowpack in the Sierra Nevada. In contrast,

1° E3SM essentially fails to represent these fine-scale features which are closely related to topography. The response of marine stratocumulus along the west coast can also be explored in CARRM as improvements in resolution have been found to be important for marine stratocumulus clouds (Bogenschutz et al., 2022; Lee et al., 2022). In our simulations coastal stratocumulus decrease significantly with warming towards the end of the century, and the positive magnitude of the shortwave cloud feedback is likely moderately high when compared to the CMIP6 spectrum.

Under the SSP5-8.5 scenario, our CARRM simulations indicate that daily maximum temperatures in the Central Valley may increase from 36 °C in the current climate to 43.5 °C by the end of century. A widespread warming of 6-10 °C is anticipated across most of California. By the end of the century, statewide 30-day average winter-spring precipitation in California is projected to increase by 38% compared to the present day. This increase is primarily due to larger amounts of precipitation falling from stronger rather than more frequent moisture surges hitting California. This aligns with the thermodynamic reaction to warming, resulting in an increased amount of precipitable water. On the other hand, our results suggest there could be a notable decrease in precipitation during the fall, which has consequences for fire season. In our simulation, California's SWE was cut in half by the 2050s and almost completely absent by the end of the century. This is consistent with the significant reductions in snowpack found in the recent downscaling studies over California and the western U.S. (Berg et al., 2016; Rhoades et al., 2017, 2018a; Sun et al., 2019; Siirila-Woodburn et al., 2021). These projections hold critical implications for California's future water resources, agriculture, energy, natural disasters (floods, droughts, wildfires), public health, etc.

Due to the nudging strategy, CARRM's mean temperature is basically inherited from 1° E3SM. However, the statewide-average precipitation shows significant differences between CARRM and 1° E3SM that increase over time (i.e., non-stationarity issue as discussed in Maraun et al. (2010)). Specifically, CARRM demonstrates superior proficiency compared to 1° E3SM in accurately representing snowpack. This is evident as the 1° E3SM model essentially fails to capture snowfall in the Sierra Nevada region. This suggests that 100 km may be sufficient if one is only concerned with the warming response in a statewide-average context, but in terms of understanding changes at the regional level, the high resolution provided by CARRM is essential; in the latter case, it is a challenge to make valid projections based on coarse-resolution models alone.

The observation-based gridded products and in-situ observations reveal a small warm bias of daily minimum temperatures, a cold bias of daily maximum temperatures in mountain regions in CARRM. As a comparison, Rhoades et al. (2018b) found a systemic mountain cold bias from 55 to 7 km variable-resolution CESM simulations. Moreover, a significant wet bias is found in CARRM. In particular, CARRM amplifies the wet bias which is already present in the 1° E3SM, which may suggest problems with the physical parameterization of SCREAM and the inadequacy of 3.25 km to fully resolve precipitation systems. For comparison, 3 km WRF hindcasts in Huang et al. (2020) did not show a wet bias, while 3 km RRM-E3SM in Rhoades et al. (2023) and 3 km / 800 m SCREAM CARRM hindcasts in Bogenschutz et al. (2024) found a wet bias especially in the Sierra Nevada. Using the hindcast strategy based on the same SCREAM-CARRM framework, the wet bias found in Bogenschutz et al. (2024) is much weaker than what we found here, suggesting that most of the bias produced by CARRM climate runs is likely due to the large-scale forcing rather than biases in the physics. A further increase in grid resolution could help clarify the resolution issue, as computational resources allow. It is also an open question whether a deep convection scheme can still

play a role in helping to better represent the fraction of convection that is not fully resolved, i.e., mitigating the overprediction of mass fluxes.

Our endeavor demonstrates the engineering feasibility and scientific validity of SCREAM-RRM for conducting decades-long climate simulations in regions of interest. SCREAM-RRM represents an excellent bridge to global convection-permitting simulations. The initial set of CARRM simulations has been employed to investigate the climate resilience of California's energy infrastructure. We anticipate further opportunities for application and iterative enhancements, including refining resolution and model physics. Given the significant benefits of high resolution, this work provides guidance and encourages the replication of SCREAM-RRM in other parts of the globe.

*Code and data availability.* The SCREAM California Convection-Permitting Regionally Refined Model 0.0 version code, in addition to the model output, can be found at https://doi.org/10.5281/zenodo.8303184. The SCREAM CARRM source code is also available on GitHub at https://github.com/E3SM-Project/scream/compare/bogensch/CA_32xRRM (last access: 30 Aug 2023), and a maintenance branch (CARRM-v0.0; https://github.com/jsbamboo/scream/releases/tag/CARRM-v0.0, last access: 30 Aug 2023). The code we used to generate the California RRM configurations for SCREAMv0 is documented in our technical note (https://acme-climate.atlassian.net/wiki/spaces/DOC/pages/3804299340/SCREAM+California+RRM+v0+Technical+Note). Specifically, the code used to generate the boundary conditions can be found at Section 5.

# Appendix A

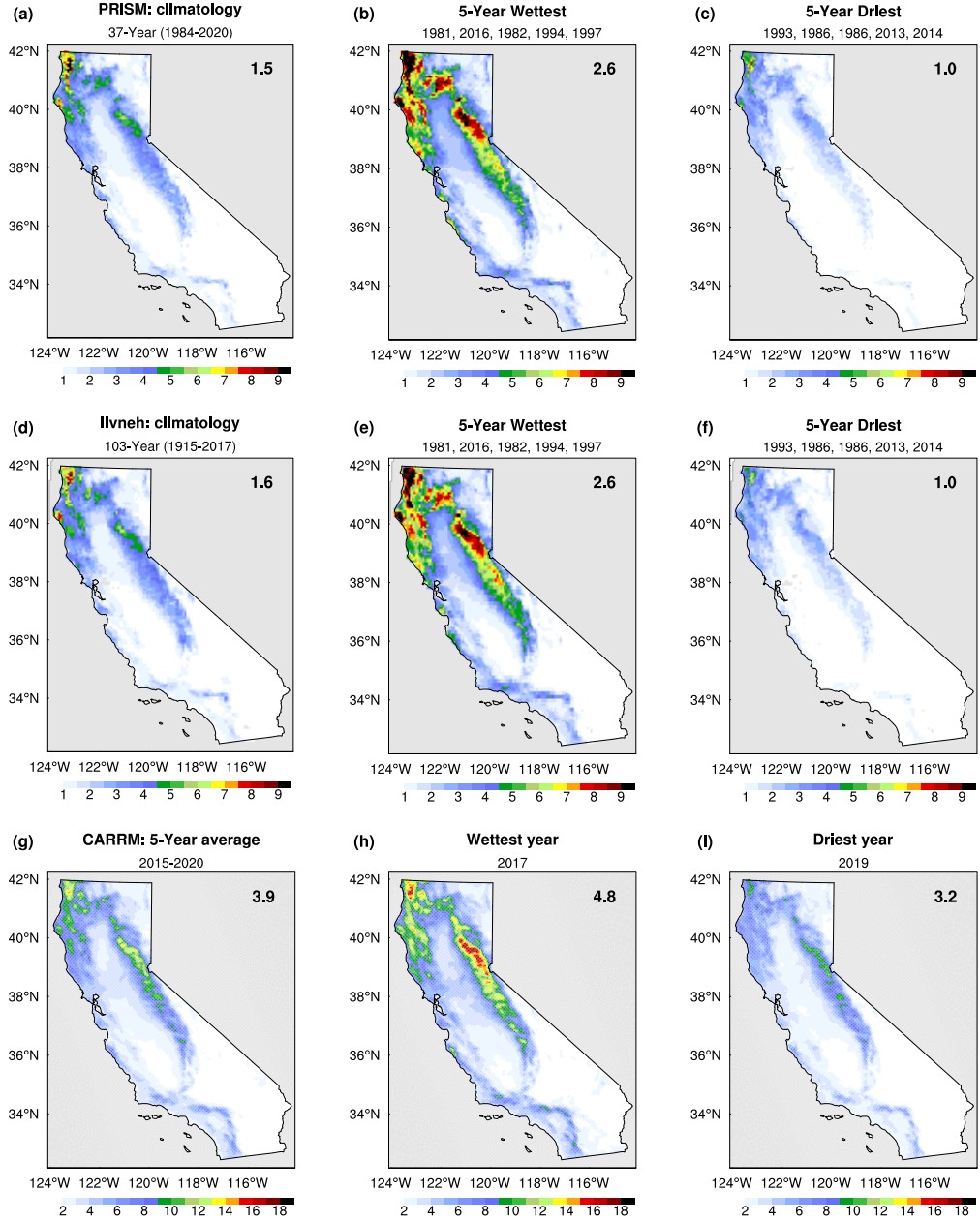

**Figure A1.** Multi-year precipitation of climatological mean (left column), wettest 5-yr mean (middle column), and driest 5-yr mean (right

column) for PRISM (top row), unsplit Livneh (middle row) observation-based gridded products, and CARRM (bottom row). The statewide average is shown in the right-top corner. The analysis period of climatological mean is 1984-2020 for PRISM, 1915-2017 for Livneh, and 2015-2020 for CARRM.

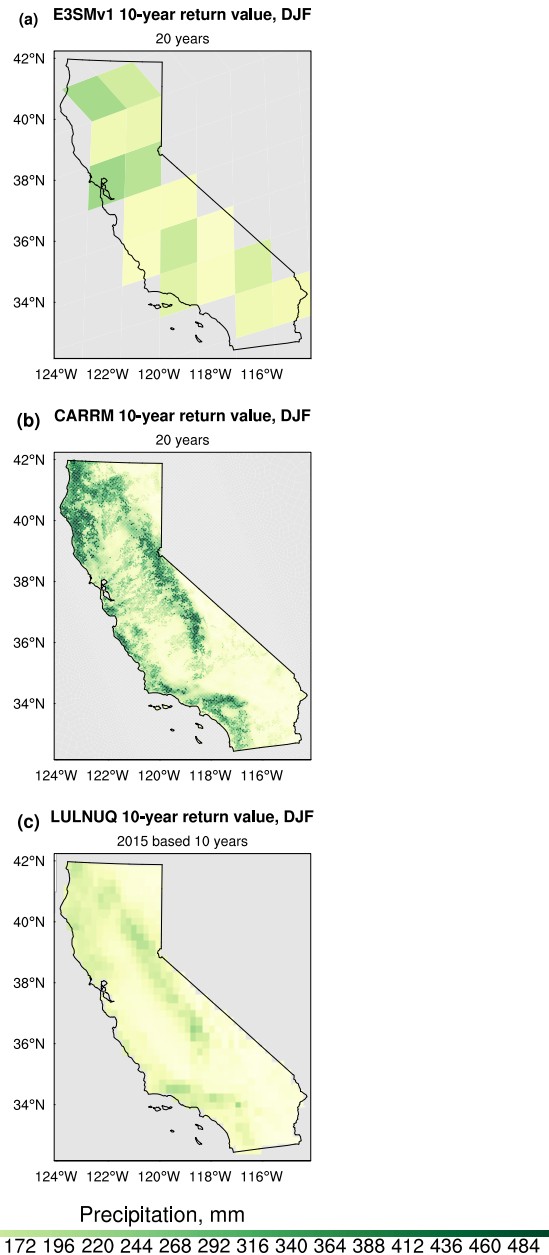

Precipitation, mm

28  52  76  100 124 148 172 196 220 244 268 292 316 340 364 388 412 436 460 484

**Figure B1.** Comparison of 10-year return values for DJF daily precipitation extremes in (a) 1° E3SMv1, (b) SCREAMv0 CARRM, and (c) a probabilistic gridded product (LULNUQ). All 20 years of simulations are used in the generalized extreme value distribution parameter estimates for E3SMv1 and CARRM. The 10-year return values for DJF in 2015 are shown for LULNUQ.

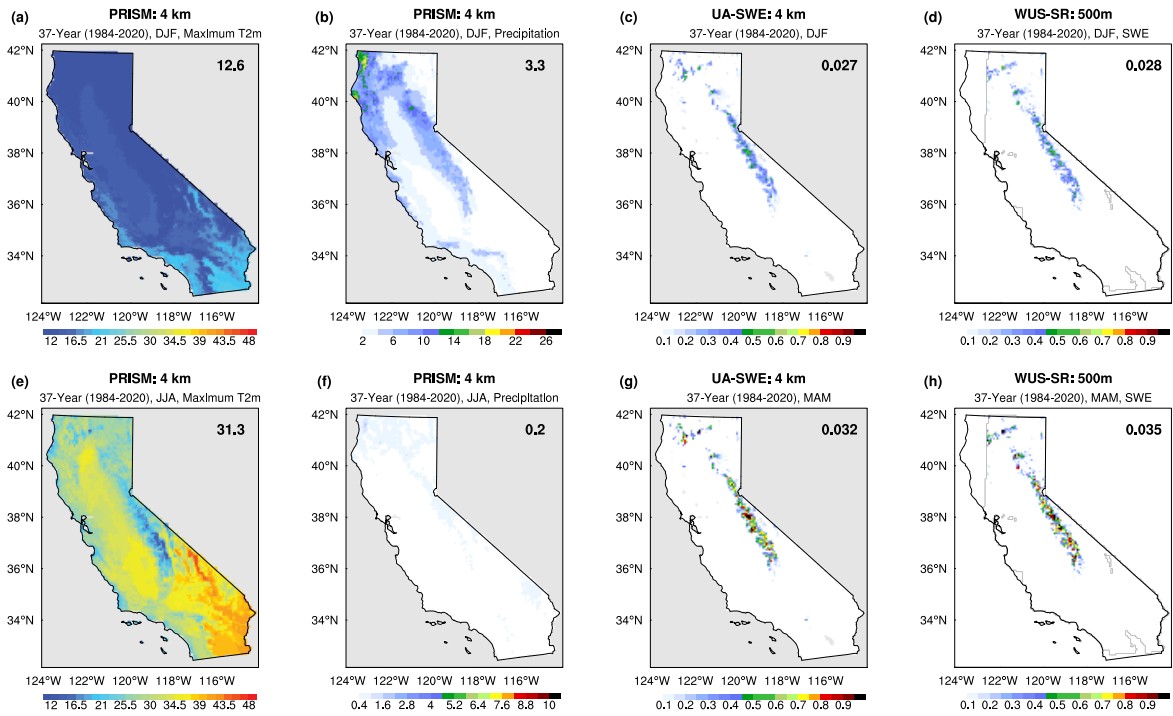

**Figure C1.** Observation-based gridded products for baseline (2015-2020 water years) multi-year average maximum T2m from PRISM in (a) DJF, (e) JJA, precipitation from PRISM in (b) DJF, (f) JJA, SWE from UA-SWE in (c) DJF, (g) MAM, SWE from WUS-SR in (d) DJF, (h) MAM.

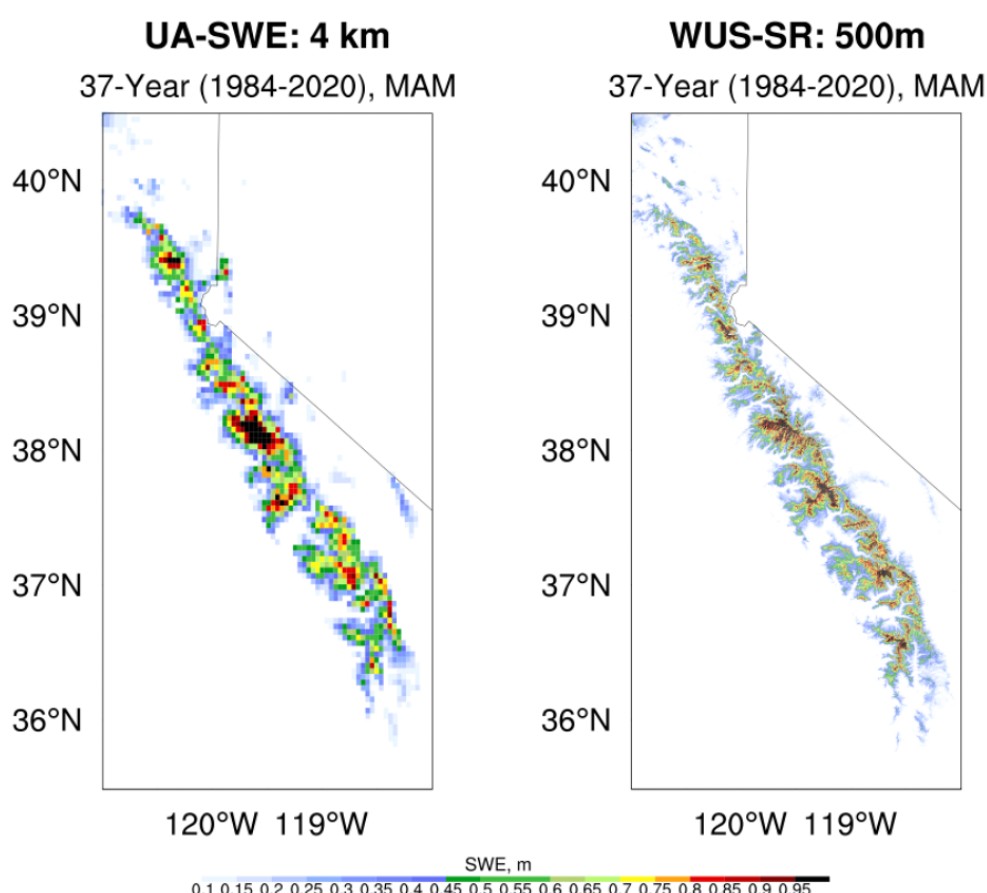

**Figure D1.** Comparison of SWE from (a) UA-SWE and (b) WUS-SR.

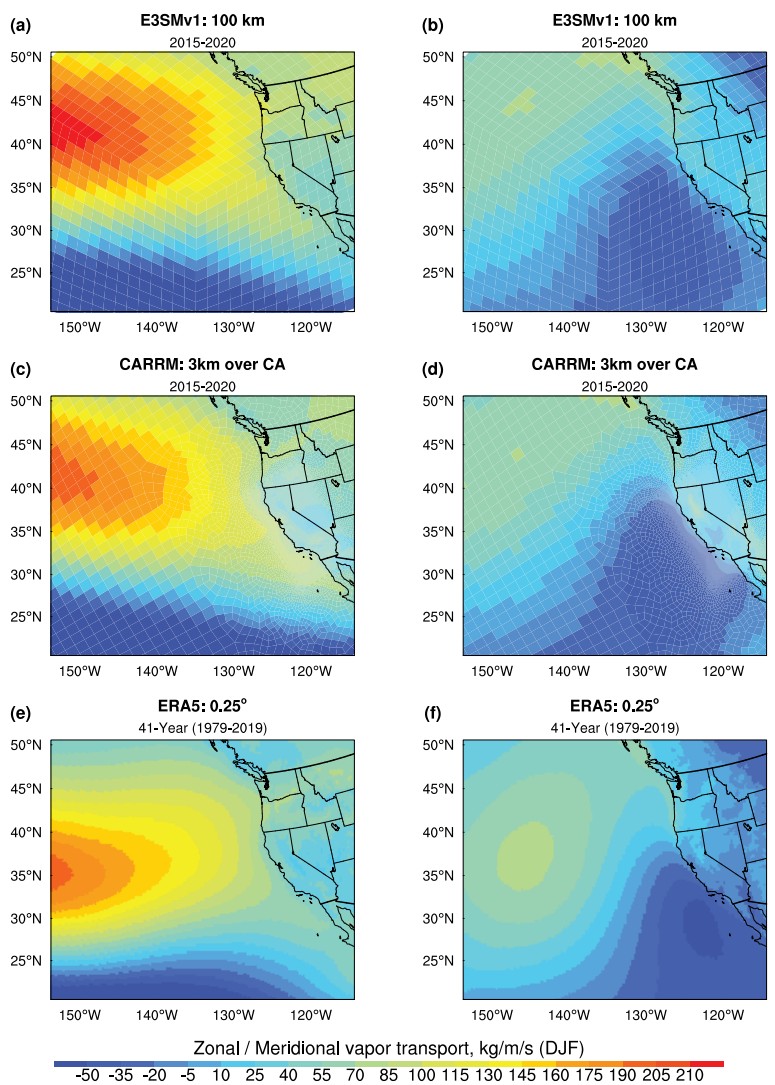

**Figure E1.** Baseline (2015-2020 water years) 1° multi-year total vertically integrated zonal (left column) and meridional (right column) water flux from E3SMv1 (top row), SCREAMv0 CARRM (middle row), and ERA5 observations (bottom row).

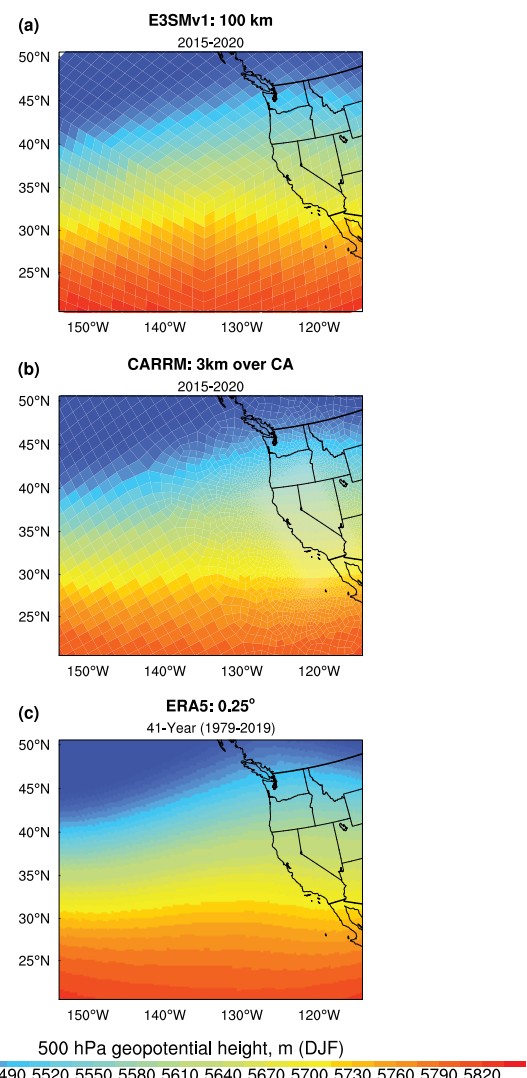

**Figure F1.** Baseline (2015-2020 water years) 1° multi-year 500 hPa geopotential height from E3SMv1 (top row), SCREAMv0 CARRM (middle row), and ERA5 observations (bottom row).

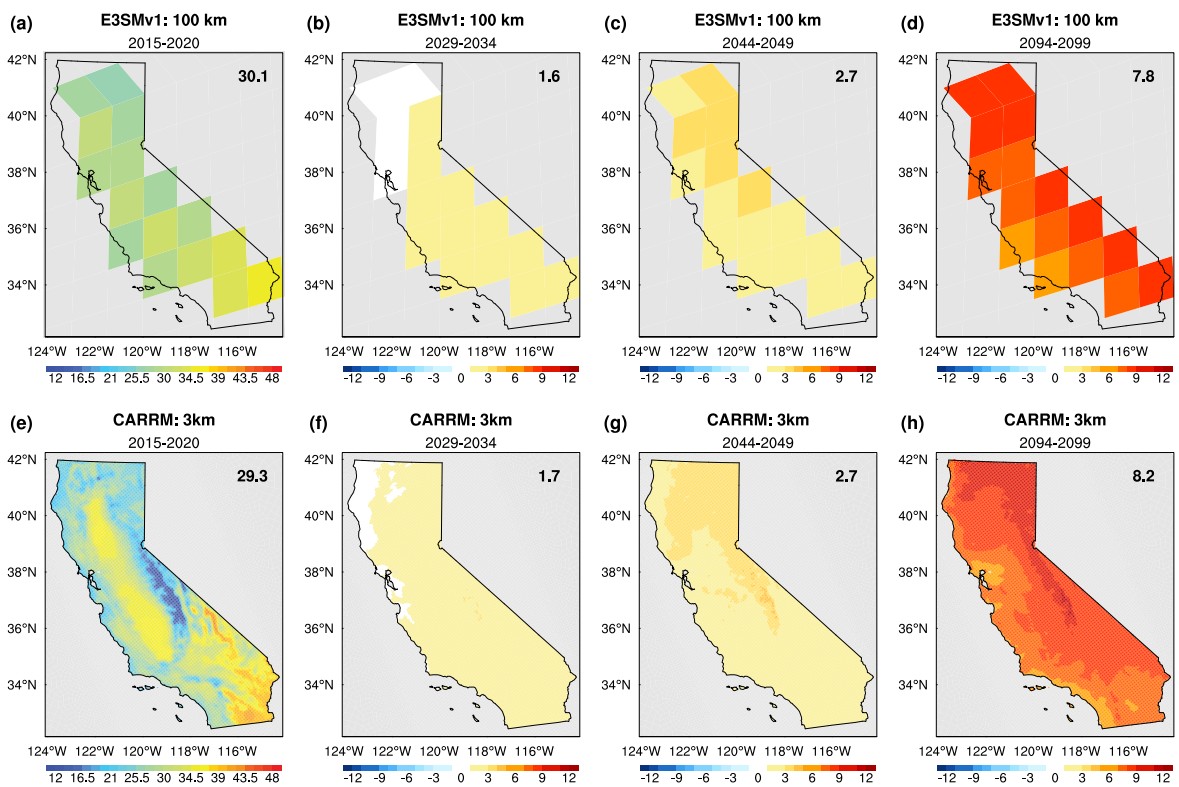

**Figure G1.** Similar to Fig. 6 but showing the differences in summer mean daily maximum T2m (°C) in (b), (f) 2029-2034, (c), (g) 2044-2049, (d), (h) 2094-2099 water years compared to the (a), (e) baseline in 1° E3SMv1 (top row) and SCREAMv0 CARRM (bottom row).

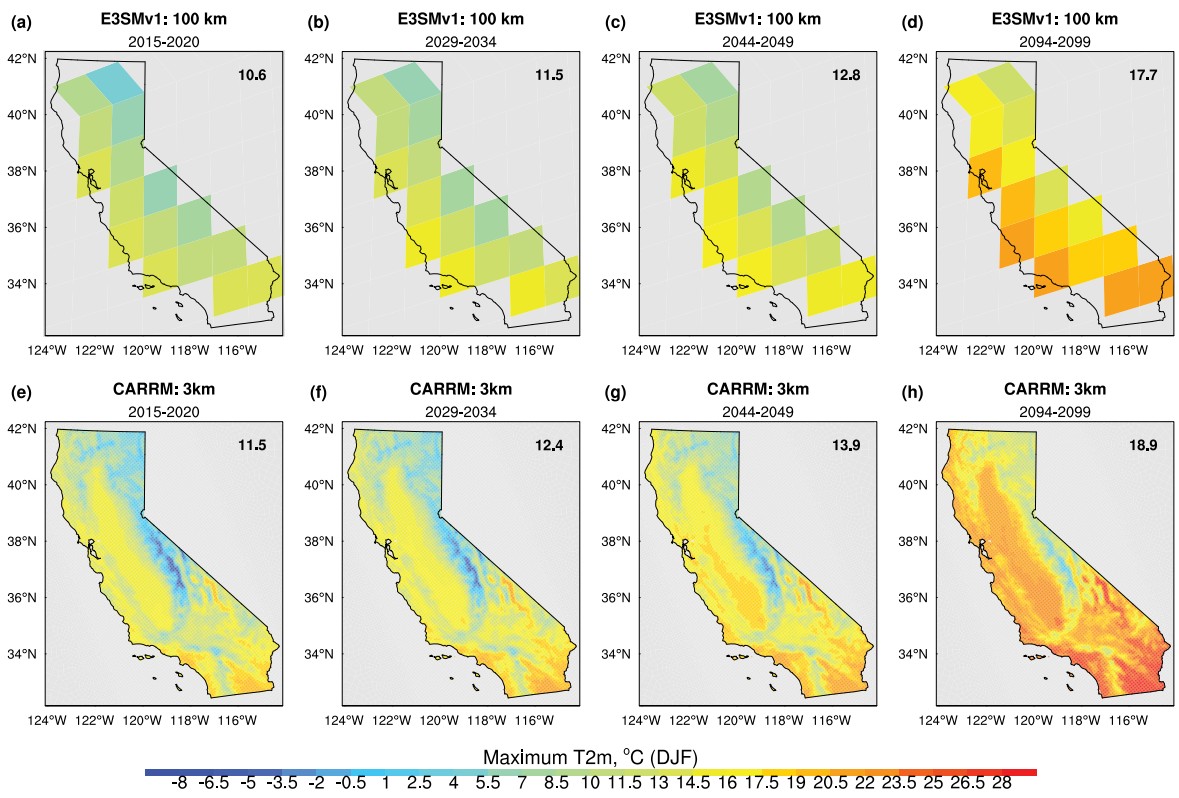

**Figure H1.** Same as Fig. 6 but for winter daily maximum T2m.

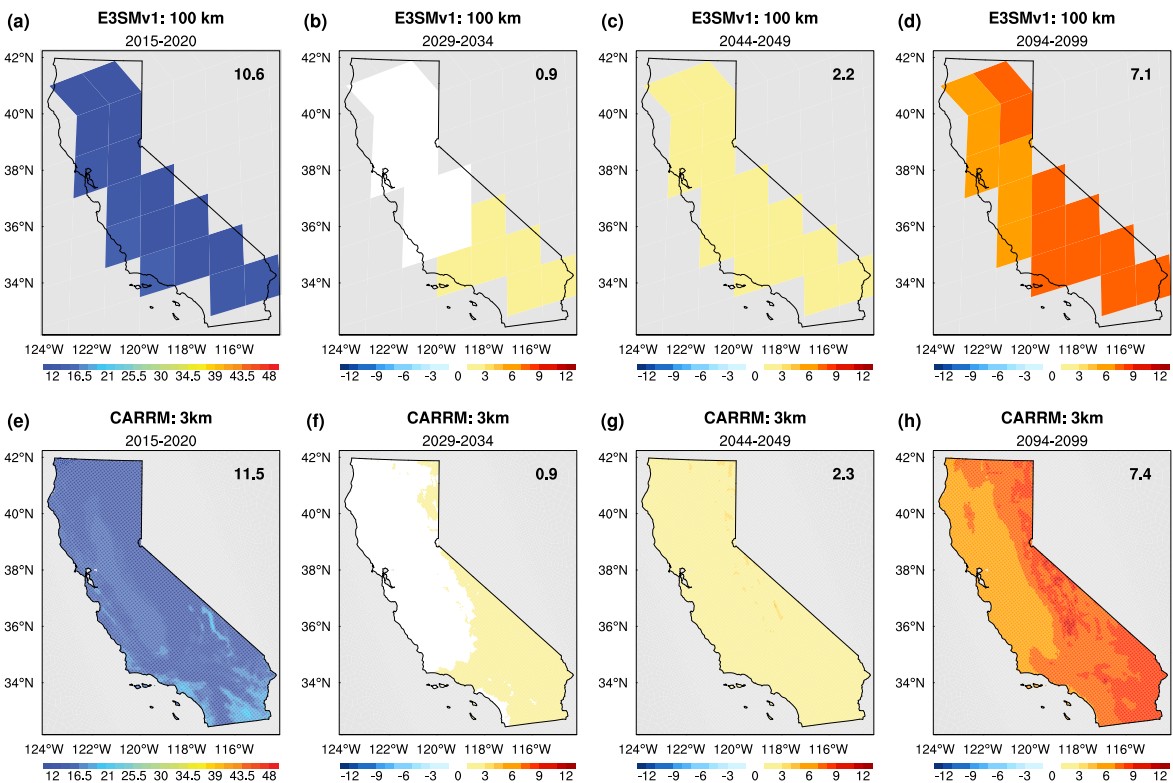

**Figure I1.** Same as Fig. G1 but for winter daily maximum T2m.

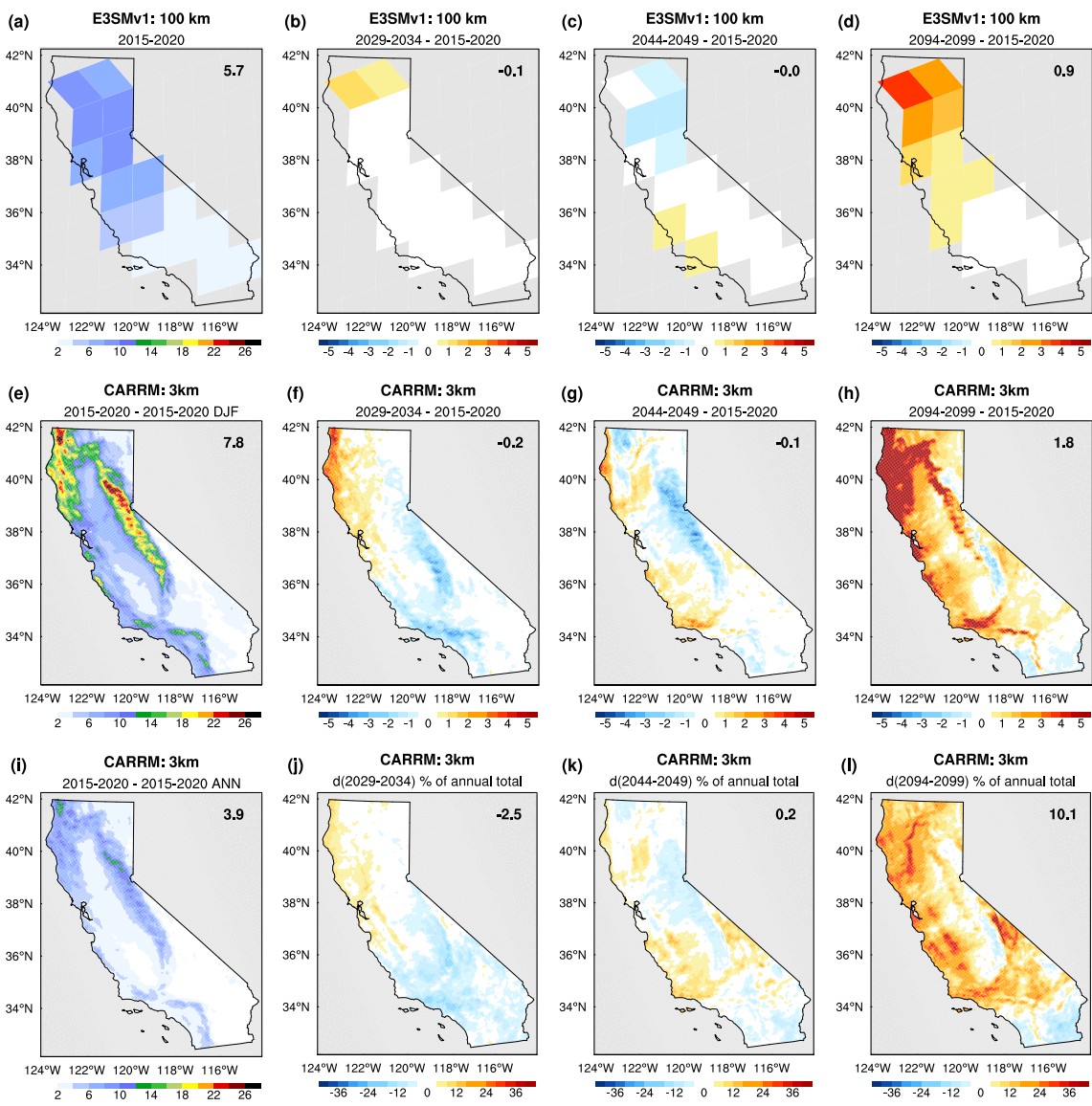

**Figure J1.** Similar to Fig. G1 but for winter precipitation (mm/day) for the top and middle rows. In addition, the difference translated to % of annual total precipitation is shown in the bottom row.

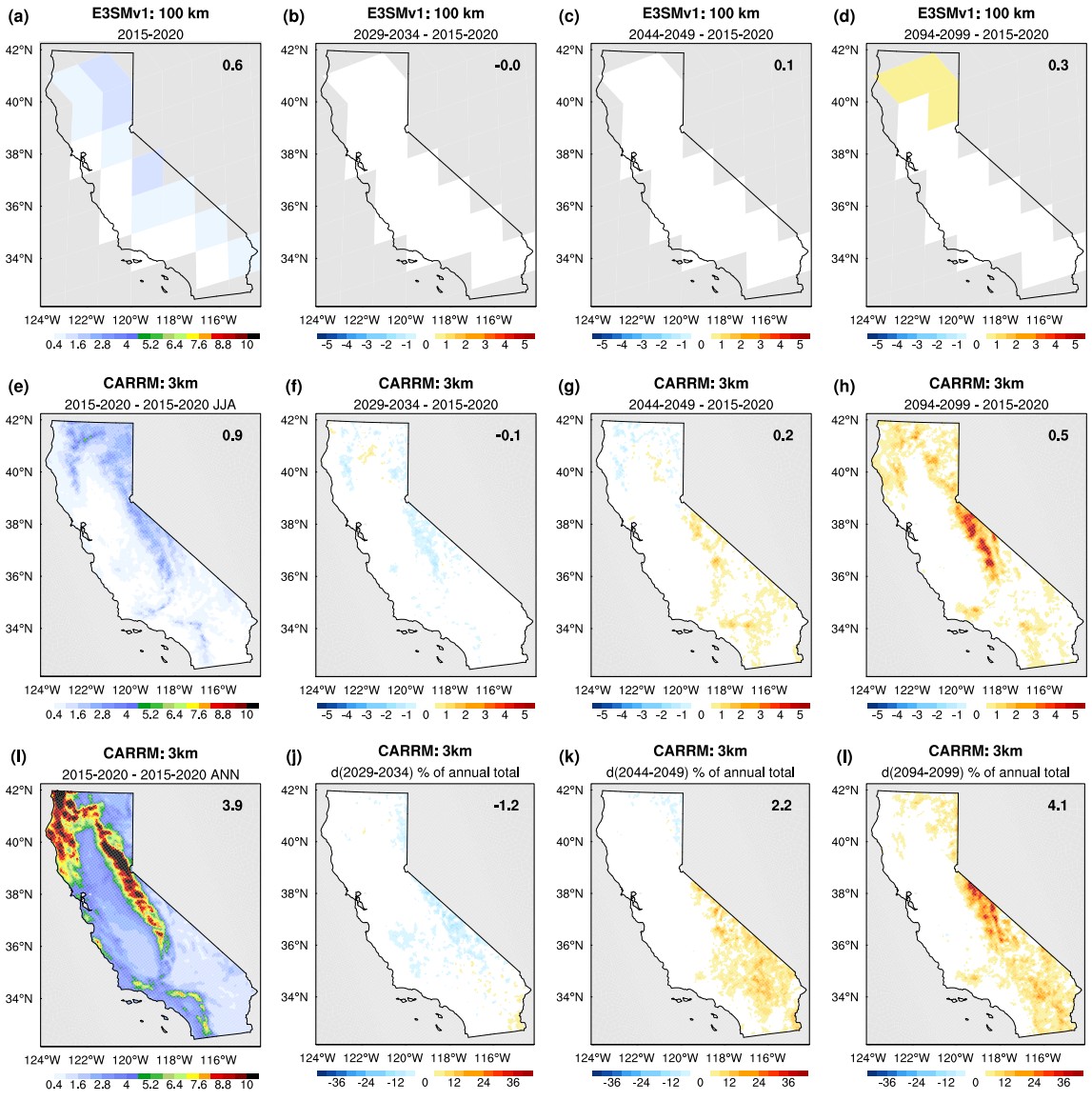

**Figure K1.** Same as Fig. J1 but for summer precipitation.

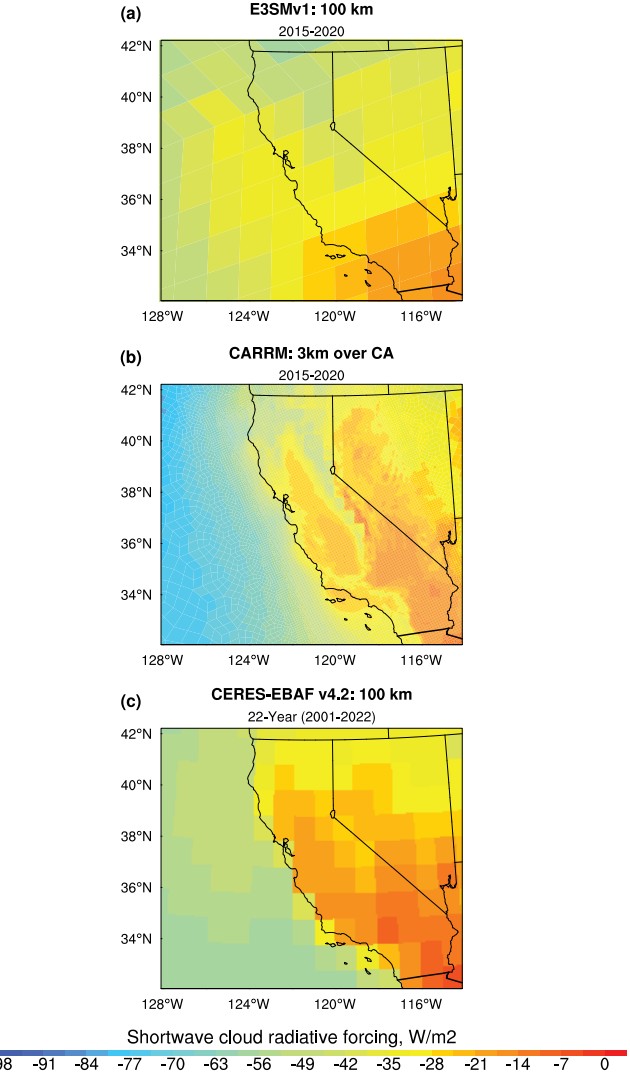

**Figure L1.** Baseline (2015-2020 water years) 1° multi-year shortwave cloud radiative forcing (SWCF) from E3SMv1 (top), SCREAMv0 CARRM (middle), and CERES-EBAF observations (bottom).

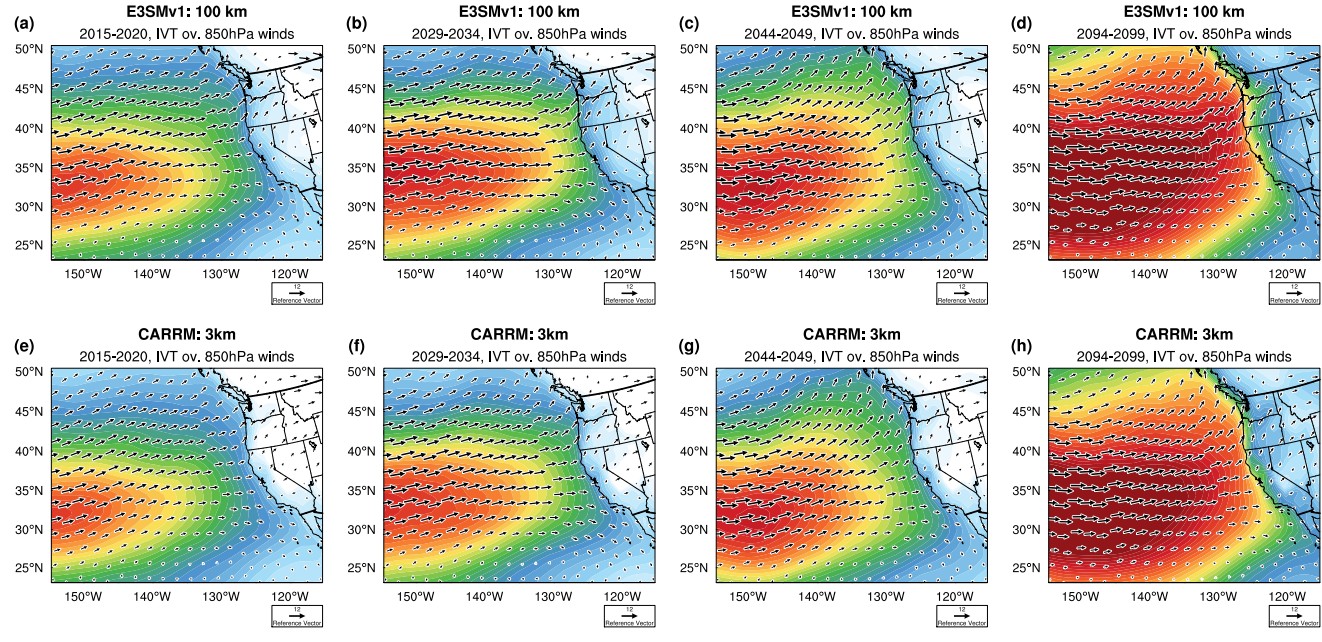

**Figure M1.** Similar to Fig. 6 but for total vertically integrated vapor transport (referred to as "IVT") with 850 hPa horizontal wind vectors.

*Author contributions.* JZ designed, performed and analyzed the 3.25 km SCREAM California RRM simulation, and prepared the first draft of the manuscript. PB took the supervisory role and contributed significantly to the design of the simulation strategy and analysis. QT generated the 3.25 km California RRM source mesh, PB, QT and JZ prepared the RRM configuration files. PB and JZ implemented the code in SCREAM. JZ carried out the SCREAM CARRM and E3SMv1 simulations with assistance from PB, QT, and CZ. PCS was responsible for funding acquisition and contributed to the analysis of the results. JZ and PB designed the paper scope. All co-authors contributed to the manuscript.

*Competing interests.* The authors declare that they have no competing interests.

*Acknowledgements.* The authors thank Walter Hannah for asking SST represented in CARRM, Shiheng Duan for providing the high-resolution Califorina shapefile, Xue Zheng for querying the E3SMv1 archived data, Shuang Yu for discussing on bias correction, Ching An Yang for valuable comments on the manuscript, Greg Lee and Aaron Donahue for their support in troubleshooting an I/O error related to parallel file system. The authors are very grateful to two anonymous reviewers whose comments and suggestions have significantly improved the paper. This documentation is supported by the Lawrence Livermore National Laboratory (LLNL) LDRD projects [22-SI-008], "Climate

Resilience for National Security" and [22-ERD-008], "Multiscale Wildfire Simulation Framework and Remote Sensing", funded by U.S. Department of Energy. Work at LLNL was performed under the auspices of the U.S. DOE by the Lawrence Livermore National Laboratory under contract (grant no. DE-AC52-07NA27344; IM release: LLNL-JRNL-853292).

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
