# Peer review of "Leveraging Regional Mesh Refinement to Simulate Future Climate Projections for California Using the Simplified Convection Permitting E3SM Atmosphere Model Version 0"

_EGUsphere, 2023_

## Author Comment (AC1)

The paper of Zhang et al. simulates future climate projections over California using the Simplified Convection Permitting E3SM Atmosphere Model Version 0. Overall, the paper is well organized, and it can help in understanding future climate change in the California region, providing more detail due to the higher model resolution. However, some issues still need to be improved. Main concerns about this manuscript are listed below.

Thank you for your nice suggestions. We have revised the text and figures (17 figures updated in the main text, 9 supplementary figures added). We have added much of the relevant literature you recommended. Again, we thank you for your suggestions and thorough review.

1.   Why does the author analyze the SSP585 scenario? The author mentions in section 2.1.5 that SSP585 is a worst-case scenario with a high probability that it will not occur. However, many studies show that the warming of the SSP370 scenario cannot be ignored either  (IPCC AR6). Of course, it is not necessary for the author to simulate the SSP370 scenario again, but it is necessary to explain it again in the article. I think just referring to the study of Tebaldi et al. is not convincing enough.

Reference:

Masson-Delmotte V P, Zhai P, Pirani S L, et al. Ipcc, 2021: Summary for policymakers. in: Climate change 2021: The physical science basis. contribution of working group i to the sixth assessment report of the intergovernmental panel on climate change[J]. 2021.

Sorry for the lack of clarity. First, SSP585 and SSP370 are of comparable scientific importance to us, with the former allowing us to study a more extreme scenario that is more relevant to the interests of energy infrastructure users, and the latter being more conservative in its predictions. For work that emphasizes scenario differences, we would choose to simulate both scenarios, but limited resources would limit the number of years we could simulate. The more critical reason is operational. We had to re-run the publicly available version of E3SM as nudging data for the coarse grid region, and we only had E3SMv1 to choose from, while the publicly released v1 future projection only has the SSP585 scenario.

We have added more explanation of the use of SSP585 in the main text according to your nice suggestion:

"We recognize that SSP5-8.5 is a ``worst'' case scenario that is unlikely to happen, due to policy interventions that promote carbon emission mitigation and sequestration, and thus represents an upper bound case of the ScenarioMIP \citep{Kriegler2017}. However, the differences between the more

plausible SSP3-7.0 and SSP5-8.5 before 2050 are relatively small \citep{Masson2021,Tebaldi2021}. Both of these scenarios predict similar development trends, including high GHG emissions, increased energy usage, and limited climate change mitigation measures before 2050 \citep{ONeill2016}. The chief reason for our choice to run the SSP5-8.5 scenario is due to the fact that we had to re-run the publicly available version of E3SM (i.e., version 1) to produce the necessary nudging data for the coarse grid region, and SSP5-8.5 is the only scientifically validated scenario for the publicly released v1 future projection."

2. Sometimes the resolution of the model described by the author is 3.25 km (Line 3), and sometimes it is 3 km (Line 47), which needs to be unified.

When specifying a description of our California RRM, we have uniformly used 3.25 km. When describing the broader concept of the convection-permitting models, the use of ~3km has been standardized.

3. Line 3: the author says that they have "developed" a CARRM model, but after reading the research, I think the author only applies the relevant model to the climate simulation field, so I suggest modifying the relevant expression. In my opinion, designing the new RRM grids and generating the model configurations cannot be considered as "developing" the new model.

Sorry for the inaccurate terminology here. In the main text we use the description "develop a California convection-permitting climate modeling framework", which, in addition to designing the RRM grid, also includes conducting the full suite of future projection simulations (re-running the low-resolution GCM, preparing nudging data and lower boundary conditions, designing simulation strategies, and basic validation). In addition, the design of the RRM grid itself is quite different from limited-area regional climate models, and the preliminary work to prepare the RRM grid has been non-trivial. Instead of simply specifying a user-level name list, the model code must be modified, all grid configuration files must be prepared using a series of stand-alone tools, and the grid may require multiple iterations to be suitable for long-term simulations. I would consider this to be development work, as future projections of CARRM are far from "out of the box". Of course, we agree that we are not developing the "whole" model itself, since the host model SCREAM, on which CARRM is based, is not part of the development effort.

We have modified this sentence to:

"We developed a 3.25 km California climate modeling framework by leveraging regional mesh refinement (CARRM) using the U.S. Department of Energy's (DOE) global Simple Cloud Resolution E3SM Atmospheric Model (SCREAM) version 0."

4.  Figures 1-19: For each subfigure, it is recommended to add legends such as (a) (b).

Thanks for the suggestion. All subfigures have added (a), (b)...

5.  Line 82: The author should briefly introduce the difference between "Modern regionally refined model (RRM)" and convective-permitting model (CPM) in their research. In fact, CPM has been widely used in climate simulations in various regions.

Reference:

Prein A F, Langhans W, Fosser G, et al. A review on regional convection-permitting climate modeling: Demonstrations, prospects, and challenges[J]. Reviews of geophysics, 2015, 53(2): 323-361.

Kendon E J, Ban N, Roberts N M, et al. Do convection-permitting regional climate models improve projections of future precipitation change?[J]. Bulletin of the American Meteorological Society, 2017, 98(1): 79-93.

As we introduced in paragraph 5, CPM is a resolution concept, emphasizing that the resolution reaches 5 km or more, so that deep cumulus parameterization can no longer be needed. CPM includes both global and limited-area (regional) models, the latter of which is the subject of the two papers you mentioned, including WRF, MetUM, COSMO-CLM, RegCM4, and so on. On the other hand, RRM is a global model as is global CPM. Since the simulation we use here is to constrain RRM by nudging it to given lateral forcings at coarse resolution (outside the region of interest), it works "like" a regional climate model, but RRM itself covers the entire globe and can provide global results, just because we focus on California here, so no other regions are shown. When RRM is run freely, it works exactly like a typical GCM (e.g., Tang2023), and there are papers discussing the upscale effects of the refined area in large-scale circulations (e.g., Sakaguchi2016). Thus, RRM and limited-area regional models are fundamentally different in terms of grid structure and evolutionary history, although they both can be classified as CPM when pushed to a CP resolution.

We have added the sentences when we first introduce RRM according to your suggestion:

> "In contrast to regional CPMs, which refer to regional climate models with limited areas \citep[e.g.,][]{Prein2015,Kendon2017}, RRMs are global models. When RRM is run freely, it works exactly like a typical GCM \citep[e.g.,][]{Tang2023}, and there are studies discussing the upscale effects of the refined area in large-scale circulations \citep[e.g.,][]{Sakaguchi2016}. Thus, although both can be pushed to a CP resolution, RRM and limited-area regional models are fundamentally different in terms of grid structure and evolutionary history."

6.    Section 2.1.5: In the historical period, is there any quantitative standard to describe the model's ability to simulate ENSO? This part is very important, because once the model cannot accurately simulate ENSO, the following years are not actually representative. In addition, the author should also emphasize how the simulation results related to future projections are related to ENSO.

Thank you for pointing this out. Section 5.5 of the E3SMv1 review paper discusses ENSO in piControl and historical ensemble simulations (Golaz2019). Overall, the ENSO variability in E3SMv1 is slightly closer to observations compared to CESMv1-LE, but strongly shifted to a 3-yr period. The overall score for spatial pattern compared to observations is higher than CESM-LE, but not enough along the North American coast.

Following your nice suggestions, we have cited the ENSO performance in the historical E3SMv1, and added the discussion of the future projections related to ENSO:

> "In retrospect, when we examine the relationship between precipitation and ENSO across the four segments, the 5-yr mean precipitation barely reflects the ENSO signal. In addition, we do not see a significant modulation of the ENSO on monthly precipitation. Instead, the climate change signal seems to be more dominant, with heavy precipitation events occurring essentially every year at the end of the century. Compared to CESMv1-LE, the ENSO variability in E3SMv1 piControl and historical ensemble simulations is slightly closer to observations, while strongly shifted to a 3-yr period. The overall score for the spatial pattern compared to observations is also higher, but still muted along the North American coast \citep{Golaz2019}. This may partially limit the ability of ENSO to modulate the climate in our simulations."

7.    Line 329: 2-m temperature -> T2m

Modified.

8.    Lines 342-345: Here, the author briefly analyzes the reasons for the biases of model precipitation simulation. In fact, regional climate models generally overestimate the meridional moisture flux. For example, the study of Gao et al. found that the WRF model overestimates southerly wind transport over eastern China. I suggest the authors cite this work to strengthen the reliability of the results

Reference:

Gao Z, Yan X, Dong S, et al. Object-based evaluation of rainfall forecasts over eastern China by eight cumulus parameterization schemes in the WRF model[J]. Atmospheric Research, 2023, 284: 106618.

Thanks for recommending this work. I'm not sure if it's comparable here: CARRM is a global model with 3.25km resolution, no deep cumulus parameterization, constrained by E3SMv1 outside of California, while Gao2023 did WRF 30km simulations with sensitivity to cumulus schemes as its research focus, constrained by observations; precipitation in California and eastern China is also controlled by very different synoptic systems. In terms of moisture fluxes here, we see no significant differences between CARRM and E3SMv1 nudging data.

Since all emphasize cumulus schemes, we have added this paper and the two papers mentioned in question 10 to the introduction that refers to uncertainties associated with deep convection parameterizations.

9.    Section 3.2.2: In fact, for precipitation, the most obvious added value of the convective-permitting model is the simulation of diurnal variations and MCS. Can the authors show some figures related to diurnal variations in the supplementary material?

Reference:

Guo Z, Fang J, Shao M, et al. Improved summer daily and sub-daily precipitation over Eastern China in convection-permitting simulations[J]. Atmospheric Research, 2022, 265: 105929.

Yun Y, Liu C, Luo Y, et al. Warm-season mesoscale convective systems over eastern China: Convection-permitting climate model simulation and observation[J]. Climate Dynamics, 2021, 57: 3599-3617.

Thank you for your question. We did not do an analysis of diurnal cycle of precipitation because precipitation in California occurs mostly in winter and is dominated by largescale processes (atmospheric rivers, mid-latitude cyclones). To the best of my knowledge, the diurnal cycle of precipitation rarely seems to be a research focus in California, and diurnal cycle studies in the U.S. are largely focused on the central Great Plains. Though, we do recognize that studying the diurnal cycle of precipitation in CA during summertime monsoon events over the Sierra Nevada and southeastern portion of the state could warrant some investigation in the future. Since diurnal cycle was not initially included as one of our evaluation objectives, we ran the experiment without outputting hourly precipitation, and E3SMv1 only outputs 6-hourly precipitation, making direct comparisons meaningless. However, the review paper for the host model SCREAMv0 documents the diurnal cycle of precipitation (its Figure 15), showing that SCREAMv0 is able to capture morning peaks over the oceans as well as late-afternoon peaks over land (Caldwell2021).

We have added a brief discussion in the last paragraph of this section:

> "Given that precipitation in California is primarily influenced by large-scale processes such as atmospheric rivers and mid-latitude cyclones, the diurnal cycle is not as significant a consideration as it is in the central Great Plains. However, as with other GCPMs, CARRM's host model SCREAMv0 captures diurnal cycles that are generally consistent with observations \citep{Caldwell2021}. Though, we do recognize that studying the diurnal cycle of precipitation in California during summertime monsoon events over the Sierra Nevada and southeastern portion of the state could warrant some investigation in the future."

10. Section 3.2.4: The author points out that the lack of marine stratocumulus clouds is a common issue in low-resolution model. In fact, for models with higher resolution but not enough to explicitly resolve the cumulus convection process, the simulation of cumulus clouds also has significant shortcomings. Cumulus clouds will release latent heat through condensation, affecting stratus clouds and ground temperature. Authors are advised to cite relevant work:

Reference:

Chikira M, Sugiyama M. A cumulus parameterization with state-dependent entrainment rate. Part I: Description and sensitivity to temperature and humidity profiles[J]. Journal of the Atmospheric Sciences, 2010, 67(7): 2171-2193.

Gao Z, Zhao C, Yan X, et al. Effects of cumulus and radiation parameterization on summer surface air temperature over eastern China[J]. Climate Dynamics, 2023, 61(1-2): 559-577.

Thank you for the recommended works. The cumulus parameterization is undoubtedly important for models that still require it, although the model we use here turns off the deep cumulus parameterization, so exploring the importance and bias of the cumulus parameterization is beyond the scope of our study. In addition, California is not a region dominated by cumulus convection (unlike the central Great Plains), so we have paid little attention to it. Marine stratocumulus clouds are discussed here because - in addition to reflecting improved horizontal and vertical resolution and improved turbulence scheme - they themselves have a significant impact on the climate of coastal California. We do recognize that SCREAM does not have high enough resolution to simulate shallow cumulus clouds and this process is parameterized by the SHOC turbulence scheme (Bogenschutz and Krueger 2013).

We have added these two papers to the introduction that refers to cumulus clouds (please see the response to question 8).

11. Line 498: Although this is a commonly used variable, give the formula for calculating the short-wave radiative forcing.

Thanks for the suggestion. We have added the formula for SWCF here:

"SWCF = FSNTOA - FSNTOA_C, where FSNTOA is net solar flux at TOA, FSNTOA_C is clear-sky net solar flux at TOA."

12. The author calculated multiple variables in the California region in the research. I would like to know whether these variables have an impact on each other? Or, how are they related? Can the author give a schematic diagram like the following article?

Reference:

Wang X, Chen D, Pang G, et al. Effects of cumulus parameterization and land-surface hydrology schemes on Tibetan Plateau climate simulation during the wet season: insights from the RegCM4 model[J]. Climate Dynamics, 2021, 57(7-8): 1853-1879.
Citation: https://doi.org/10.5194/egusphere-2023-1989-RC1

The variables we show here are organized primarily for the interests of the reader who cares about California climate, so attention is given to the most basic variables (temperature, precipitation, snow), and marine stratocumulus clouds which have a large impact on California. It is not organized from the perspective of understanding model biases. As in your recommended work, more sensitivity testing based on the choice of physical schemes is needed to make careful inferences about the relationships between different variables. However, SCREAMv0 does not have an optional physical parameterization suite, and we didn't perturb any physical parameters to get a conclusive plot like Wang2021's Figure 15. Nevertheless, speculatively, the relationships we find here are: a) a link between warming and reduced snowfall (well known), b) an increase in precipitable water due to warming leading to a strengthening of individual storms without significant changes in the winds (not surprising), and c) a link between warming / increased $CO_2$ and reduced stratocumulus clouds (weakened SWCF) through thermodynamic and radiative mechanisms (findings in the stratocumulus field); validation here requires checking variables such as specific humidity gradient, turbulent entrainment rate, cloud-top longwave radiative cooling, etc. The distinct comparison between E3SMv1 and SCREAMv0 CARRM on the shortwave cloud radiative feedback deserves an in-depth separate analysis.

---

## Author Comment (AC2)

Summary

Zhang et al. in "Leveraging Regional Mesh Refinement to Simulate Future Climate Projections for California Using the Simplified Convection Permitting E3SM Atmosphere Model Version 0" evaluate the historical skill and future projections of the U.S. Department of Energy's new cloud-resolving model, SCREAMv0.  The authors evaluate a historical 5-year (2015-2020) period produced by SCREAMv0 under an AMIP experimental protocol with regional refined mesh capabilities (RRM) focused over California (CARRM) compared against a conventional ESM simulation (E3SMv1) and observation-based gridded products (PRISM and UofASWE).  The variables evaluated are temperature, precipitation, and SWE.  The authors then evaluate projections of three, five-year periods (2029-2034; 2044-2049; and 2094-2099) in the future under a high emissions scenario, SSP585.  The future climate simulations project much stronger extreme heat days, more extreme precipitation rates, and substantial losses in winter-to-spring snowpacks.

Overall, I think the paper fits well within the scope of GMD and could be, given more work, a valuable contribution to the scientific community.  The findings have both scientific and societal impact as CARRM is one of the first applications of SCREAMv0 to evaluate its historical skill and investigate projections of future climate change.  CARRM also represents a state-of-the-art methodology leveraging an ESM with RRM capabilities.  This presents an exciting new advance in high-resolution climate modeling (e.g., convective resolving) as well as new capabilities for RRM simulations (e.g., inner grid nudging capabilities).  The experimental design description and methods section is also very clear and would enable reproducibility.  I respected that the authors even provided model error messages when running the simulations, which is both refreshing and uncommon in the literature.  I also appreciated that the authors provided all of their code/post processed simulation data in the "Code and data availability" section.

With that said, I encountered frequent issues with the manuscript's sentence structure (e.g., incomplete sentences, run on sentences, logical flow, etc.) and imprecise description of results.  I also completely understand computational limitations in running cutting-edge RRM simulations, but given the large interannual variability of the California hydroclimate, I'm a bit worried about only using 5-year simulations to interrogate historical skill and future projections of precipitation, temperature, and snowpack (especially in the California Sierra Nevada).  The authors also stated quite strong conclusions from their results, without caveating those findings with any sort of observational and/or internal variability uncertainty estimates.  In my experience, simulations would need to be run for 15-20 years to even begin to see convergence in California hydroclimate summary statistics (i.e., mean, median, IQR).  At the very least, I

would like to see the authors do a more rigorous job of setting the context of the CARRM simulations in the broader literature of GCM, RCM, VR, and RRM simulations conducted thus far and utilize more in-situ observations and, at least, one other observation-based gridded product. This will, at least, better contextualize the uncertainty in leveraging 5-year simulations (internal variability).

Therefore, I think there are still substantial and major revisions needed prior to this paper being accepted in GMD.

Thank you for your careful reading, insightful suggestions, and thorough review. We have added many in-situ observations and new observation-based grid products. We have revised the text and figures (17 figures updated in the main text, 9 supplementary figures added). Our original conclusions were primarily intended to highlight the added values of CARRM, rather than claiming that CARRM sufficiently samples the internal variability. Again, we thank you for your suggestions and hard work in reviewing this manuscript and helping to make it better.

Comments and Suggested Edits

Line 16 – Delete "In addition to the countless smaller valleys and mountain ranges, California's complex topography" and add "This complex topography"

Modified.

Line 18 – Change "accumulated water" to "acting as a key natural store of that precipitation"

Changed.

Line 19 – Delete "extremely"

Modified.

Line 20 – Change "energy sector" to "energy supply"

Changed.

Line 15-25 – the authors use very dated citations and might consider citing some of the more recent literature on California hydrometeorology/hydroclimate

Thanks for pointing this out! We have added more recent literature here:

"Snowpack is also important to California's energy component, with hydroelectric power providing 56\% of the western U.S. energy supply and up to 21\% of California's diverse energy portfolio \citep{Stewart1966,Bartos_Chester2015,Solaun_Cerda2019}.

With climate change, California is likely to experience significantly warmer temperatures, less snowpack, a shorter snowpack season, and more precipitation settling as rain rather than snow, resulting in earlier runoff diversions, increased risk of winter flooding, and reduced summer surface water supplies \citep{Gleick_Chalecki1999,Hayhoe2004,Leung2004}. As one of the world's largest agricultural suppliers and a key U.S. energy supplier, changes in regional temperature, precipitation, snowpack, and water availability in California could significantly affect the state's agricultural economy and future power supply capacity \citep{Tanaka2006,Hanak_Lund2012,CAFoodAgriculture2016,Bartos_Chester2015,Pathak2018,Arellano2021}. Specifically, recent downscaling studies have found that under the impacts of climate change, California and the western U.S. will experience significant reductions in snowpack, including reduced winter snowfall, earlier spring snowmelt, and increased interannual variability, with important implications for water management and flood risk \citep{Berg2016,Hall2017,Musselman2017,Rhoades2017,Walton2017,Musselman2018,Rhoades2018b,Marshall2019,Sun2019,Siirila2021}."

Line 28 – What does "14% generation" mean? Do you mean "provide 14% of the energy supply"?

Yes, modified.

Line 29-30 – this sentence does not make sense and needs to be removed or revised for clarity "The predicted short-term and long-term effects due to energy changes of future wind and radiation on Central Valley temperatures have also received attention"

Sorry for the misleading text. This has been modified to:

"Predictions of future wind and solar generation in California have also received attention \citep{Crook2011,Wang2018}."

Line 32-40 – I think this paragraph should be the first paragraph and portions of the 1st paragraph should be integrated into this paragraph

Thank you for your nice suggestions. We have reorganized paragraphs 1 and 2 so that paragraph 1 now introduces important microclimate features, and paragraph 2 introduces the existing literature on future projections for California and their implications.

Line 48 – Change "sometimes rapidly" to "can rapidly"

Changed.

Line 51 – I think you mean "snow-rain transition" rather than "snow albedo feedback"

Yes, modified.

Line 53-54 – Change "these microclimatic nuances … accurately project and understand of the state's complex climate patterns" to "The processes that give rise to these microclimates … understand their interactions and project how they may become altered under climate change"

Changed to:

> "The processes that give rise to these microclimates require the use of high-resolution models to understand their interactions and project how they may become altered under climate change."

Line 55 – Delete "hazard"

Deleted.

Line 60 – this was also recently evaluated in RRM-E3SM at 14km vs 7km vs 3.5km horizontal resolutions - https://agupubs.onlinelibrary.wiley.com/doi/full/10.1029/2023MS003793

Thanks for pointing to this paper, it's very relevant. We have added it here:

> "\cite{Rhoades2023} recently evaluated RRM-E3SM at 14km vs 7km vs 3.5 km horizontal resolutions and demonstrated the forecast skill of 3.5 km in recreating extreme floods."

Line 66 – Change "simply disappear with running with a resolution of 3km" to "are significantly reduced when run at 3km horizontal resolution"

Changed.

Line 74-75 – Change "with the main purpose of dynamically downscaling…" to "allowing for low-resolution boundary condition data to be dynamically downscaled to high resolution over regions of interest"

Changed.

Line 77 – Change "predictions" to "projections"

Changed.

Line 77-78 – this sentence does not make sense and needs to be removed or revised for clarity "The sub-GCM grid-scale details are represented by downscaling capability"

Modified to:

"The sub-grid scale processes are represented by downscaling techniques \citep{Giorgi2019}."

Line 79 – Change "global GCMs have also undergone a history of…" to "GCMs now have the capability to employ variable resolution grids and regionally refined meshes by capitalizing on unstructured grid development"

Changed.

Line 82 – Change "GCM scale" to "Synoptic scale"

Changed.

Line 88 – Change "more comprehensive simulations" to "AMIP and fully-coupled simulations"

Changed.

Line 113 – change "regionally refined model" to "RRM" (already defined earlier in the manuscript; check in other parts of the manuscript)

Thanks for the reminder. Changed.

Line 117-118 – SCREAM has already been defined earlier in the manuscript

Removed the full name here.

Line 119-120 this sentence does not make sense and needs to be revised for clarity "SCREAM leverages DOE's leadership in computationally intensive frontier designed for convective permitting scales and does not parameterize subgrid-scale deep convection since it uses a resolution of 3.25 km globally"

Modified to:

"SCREAM has a global resolution of 3.25 km and thus does not parameterize deep convection."

Line 125-128 – The authors should remove mention of "Amazon precipitation, tropical precipitation, etc." when making the case that it is well suited for application over California.

Removed.

Line 140 – Change "towards improving marine…which is important for the climate off the California coast" to "that improved marine…which is important for representing the California coastal climate"

Changed.

Line 143 – Change "routine" to "routing"

Changed.

Line 152 – "transition zone between them" it is hard to tell if there is, at least, a 6 dx grid cell transition between refinement regions. Is this the case? If not, do you think this would present any numerical artifacts in the simulation?

It may not be obvious from the figure - it undergoes an 8th order (2^8) refinement from ~100 km (ne32) to 3.25 km. We didn't see any numerical artifacts in our simulations.

Changed this sentence to:

> "The CARRM grid is progressively refined from the outer global resolution of ne32 (corresponding roughly to a resolution of \textasciitilde100 km) to the convection-permitting scale for California (ne1024, 3.25 km), with a 8th order ($2^8$) refinement between them (Fig. \ref{RRMgrid})."

Line 157 – Change "computation" to "computational"

Changed.

Line 158 – Delete "passing westward" to "originating"

Deleted.

Line 160 – FWIW the "sensitivity of the size of the refined mesh for the simulation of atmospheric rivers" was (partly) explored with CESM already - https://agupubs.onlinelibrary.wiley.com/doi/full/10.1029/2019JD031977

Thanks for pointing out this! Added it.

Figure 1 – can the authors standardize the color bar range across all four subpanel plots of topography?  The authors should also add subpanel plot identifiers a) ... f) and reference in the main text.  Also, any acronyms should be defined in the figure caption to make the figure "standalone" from the main text.

We can't standardize the color bar in the topography plots because the "tricolor" function in python doesn't allow a manually specified color levels. Another function "tricontour" can specify color levels (as indicated by its name contour), but we don't want to use this function because it interpolates the data and thus cannot represent the original grid resolution.

Thanks for the suggestion. All subpanels have added (a), (b)...(f).

Line 165 – reference for "tricolor function"?

Added:

> "Since the topography files are on the GLL node, we used matplotlib's ``tricolor'' function to represent the native spectral element data as accurately as possible,

with each triangle's color taken from three GLL vertexes (\url{https://matplotlib.org/stable/api/_as_gen/matplotlib.pyplot.tripcolor.html}, last access: 21 January 2024; note that the ``tricolor'' function doesn't allow a manually specified color levels)."

Line 167-169 – cite this statement (nominal vs effective resolution)

Cited: citep{Neumann2019,Caldwell2021}

Line 173 – "hoirzontal" to "horizontal"

Corrected.

Line 174 – "highorder" to "higher order"

Changed all "highorder" to "higher order".

Line 177-182 – cite this statement about "good grid properties" and explain what is meant by "good"

The metrics used here is "max Dinv-based element distortion". Please see the explanation in E3SM confluence page: "A high quality regionally refined grid will have a value less than 4" (https://acme-climate.atlassian.net/wiki/spaces/DOC/pages/872579110/Running+E3SM+on+New+Atmosphere+Grids). We have cited the concept in a footnote.

Line 177-182 – break this sentence up into several (run on sentence)

Split.

Line 200-204 – I applaud the authors for putting this in their manuscript (esp. for reproducibility). FWIW (I believe) these errors have also been reported in detail here - https://github.com/E3SM-Project/E3SM/issues/5089

Thank you. We noted this issue page too, however, the cold T issue may be a separate problem. One thing to note is that reducing the dycore timestep didn't help for what Colin reported in the issue page, but it did help us pass the instability in our simulations. In addition, we didn't output every timestep to check if there is a cold T issue before the failure.

Line 208 – Change "drive coarse grid" to "provide coarse scale fields that drive CARRM"

Changed.

Line 213 – Define "CICE"

Modified:

"…Prescribed CICE4 (the latest Los Alamos sea ice model) \citep{Hunke2008}"

Line 214 – Change "was" to "were"

Changed.

Figure 2 – can the authors make sure that text does not run onto the figures.  Please define all acronyms to make the figure "standalone" from the main text.  Change "with no nudging is applied in red area" to "where nudging is not applied in red areas"

Thanks for the suggestions. The text position is tuned. The caption is modified.

Line 229 – Delete one of the "the"

Deleted.

Line 238-239 – rather than making two rows of angled maps, couldn't the authors simply difference E3SMv1 and SCREAMv0 CARRM IVT fields and show the difference plot?  This would also allow you to "see" any potential "artifacts" from the various nudging coefficients.

We learned to make this figure from image processing / ML papers. Its main advantage is that it saves space when there are multiple subplots (more than 4 for each row), and emphasizes the main message (contrasts between the top and bottom rows) and omits details (e.g., latitude/longitude, subheadings for each subplots). We did not make the difference plots because the resolutions are different. If we use the interpolated E3SMv1 fields to make the difference plots, we run into a typical problem similar to that of verifying precipitation performance in CP models - a simple spatial offset can cause the difference plots to look drastically different, resulting in low RMSE scores. We therefore think that it may be more intuitive to directly compare the original plots. Nevertheless, to the naked eye there will be some differences between the two. We do not consider this to be artifacts, but a natural effect of nudging (also note that we used a weak relaxation timescale).

We have added a sentence here:

> "Note that there are some differences between them, which is expected to be a natural effect of nudging, especially since we used a weak relaxation timescale."

Line 240 – Change "Experiment" to "Experimental"

Changed.

Line 241-244 – Change "radiative forcing path close to the highest..." to "which is comparable to the radiative forcing path of the highest representative ..." Change "due to policy" to "due to policy interventions that promote carbon emission mitigation and sequestration"

Changed.

Line 241-252 – if the authors are worried about the current debate in the climate literature regarding "hot models" (equilibrium climate sensitivity) and/or "worst case scenario" (climate "doomism") they could also cast their simulations as being run according to the global warming levels of interest to the IPCC AR6 reports (i.e., +1.5 degC, +2.0 degC and +3.0 degC)

Added the global warming levels after introducing the four simulation segments according to your suggestion below.

Line 249 – Delete "to some extent"

Deleted.

Line 258-261 – as mentioned earlier, this might be a good time to provide the global warming levels provided by E3SMv1 (relative to an early 19[th]/20[th] century 30-to-50 year baseline)

Thank you for the suggestion. We have added a sentence here:

> "From another perspective, the four simulation segments provide different levels of global warming (about 0.9 °C, 1.7 °C, 2.8 °C, 7.6 °C) relative to the 1850-1869 baseline \citep{Zheng2022}."

Line 263 – Change "to reanalysis" to "with a reanalysis dataset"

Changed.

Line 264 – Change "regional model to historical" to "with historical"

Changed.

Line 268 – "WRF" has not yet been defined and should probably be mentioned in the following way "like commonly employed regional climate downscaling approaches such as the Weather Research and Forecasting, WRF, model"

Modified.

Line 240-270 – do the authors allow for a "spin-up" of the model prior to analysis? Notably this will be important for the land surface (e.g., soil moisture and, potentially, snowpack).

The spin-up of the atmosphere took less than a week (checked from the timeseries between the simulated fields and the nudging fields). We used a cold-start land initialization. Considerations include: 1) the model response is constrained by the prescribed SST and the nudging strategy, 2) additional resources would be required to get a well spun-up land initial condition (either to rerun historical simulations to get sub-daily 2-meter temperature, humidity, precipitation, radiation as the forcing for the land model, or to run CARRM for longer durations directly; both are time consuming). We admit that a well-spun land initial condition can be important for short-term simulations such as Rhoades et al. (2023). We have also noted the sensitivity of the land initial condition in our hindcasts and pseudo-global-warming simulations for an extreme flood event (in preparation).

Figure 3 – provide the global and/or regional warming levels for each of these 4 time periods

We have added the global warming levels in the figure caption. The regional warming levels were added in Fig. 7, 8.

Line 285-287 – given the large interannual variability of snowpack (i.e., snow depth, SWE and snow covered area) and mismatch/uncertainties across gridded SWE products I would suggest also comparing to the WUS snow reanalysis - https://nsidc.org/data/wus_ucla_sr/versions/1#:~:text=This%20Western%20United%20States%20snow,water%20years%201985%20to%202021.

Thank you for recommending this dataset! We have added it in our analysis. In short, SWE from UA and WUS-SR are very close to each other for a statewide-mean perspective, while WUS-SR has a much finer resolution.

Considering the substantial changes in the baseline comparison part (and the introduction of observational/reanalysis dataset), we didn't paste the updated text here. please see the updated manuscript in \subsubsection{Evaluation datasets} and \subsection{Baseline comparison with observations}.

Line 319 – "...simulated time period does not coincide with observations..." I don't think this is true. I think what you're trying to say is that the CARRM simulations are AMIP-style experiments (i.e., only prescribes observed lower boundary conditions in SST/sea-ice conditions) and should not be expected to exactly recreate the 2015-10-01 to 2020-09-01 period?

Yes, we meant to say the simulation period does not coincide with observations because our simulations are not hindcasts (using realistic boundary conditions such as ERA5 and OISST). Hindcasts are also AMIP-type.

We have changed this sentence to:

> "Since the simulation period is not corresponding to the ``real world" (because our simulations are not hindcasts using realistic boundary conditions), the simulation can only be compared to observations in a statistical sense (e.g. long-term averages)."

Line 321-329 – the last sentence of this paragraph stood out to me. A common and systemic issue that I've seen across models is a cold bias in mountains, especially during winter, this appears to be the case in CARRM too (visual comparison of CARRM to PRISM in the highest elevations of the California Sierra Nevada). Can the authors state how much colder the CARRM simulations are in the California Sierra Nevada? I would also make a note that the simulations represent only 5-year averages whereas PRISM represents 30-year averages. This is especially important given the large interannual variability in California temperature/precipitation (especially compared to other regions around the world) and might obscure "warm" or "cold" biases (and precipitation/snowpack too).

Thanks for pointing this out. "Overall" meant for global latitudinal average (please see Caldwell et al., 2021, Fig. 6, "T2m was uniformly too high at high latitudes"). The comparison for Tahoe City Cross shows that the seasonal mean of maximum T2m in JJA is ~1-2◦C colder than GHCN/PRISM (Fig. 8c), while the minimum T2m in SON is ~2◦C

warmer than GHCN/PRISM (Fig. 9c). Addtionally, the temperatures from GHCN and PRISM are similar for Sacramento, San Francisco, Tahoe city cross, and Death Valley.

We have modified the text according to your suggestions:

> "\cite{Caldwell2021} reported that SCREAMv0 does have an overall warm bias for T2m, especially at high latitudes, while we also see the cold bias in daily maximum T2m. A further comparison with GHCN and PRISM at Tahoe City Cross shows that the seasonal mean of maximum T2m is 1-2 °C colder than GHCN/PRISM (Fig. \ref{boxSP-1dA-TREFHTMX}c vs. Fig. \ref{OBS-ghcn-SNOTEL-SP-ssnbox}a), while the minimum T2m in JJA is about 2 °C warmer than GHCN/PRISM ((Fig. \ref{boxSP-1dA-TREFHTMN}c vs. Fig. \ref{OBS-ghcn-SNOTEL-SP-ssnbox}b). Note that the simulations represent only 5-year averages whereas PRISM represents 30-year averages. This is especially important given the large interannual variability in the California climate, and might obscure ``warm'' or ``cold'' biases (and is relevant for the results to be presented for precipitation/snowpack)."

Figure 4 – I would suggest putting "left column", "right column" … and "top row", "middle row" … Also, TREFHTMX, TREFHT and TREFHTMN are not defined explicitly and I would just delete those acronyms from the figure entirely. I would eliminate redundancies in subpanel plot titles too (e.g., only define E3SMv1: 100 km in one subpanel plot). The authors use CARRM throughout the text, but in this figure they put SCREAM CA RRM, I would either use SCREAM CA RRM or CARRM throughout. Last, could you change "degC" to "°C"

All subpanels now have (a), (b)…(f), as suggested by reviewer #1. All abbreviations have been expanded to maximum/average/minimum T2m (I didn't expand the full "2-meter temperature" due to limited space). I didn't eliminate redundancies in subpanel titles because it will break the uniform panel size in NCL. All "SCREAM CA RRM" in figures have been changed to "CARRM". All "degC" has been changed to "°C".

Line 330 – building off my previous comment, this paper nicely highlights the large interannual variability of California precipitation and the (potential) issues with comparing 5-year vs 30-year normals - https://www.mdpi.com/2073-4441/3/2/445 (see Figure 2)

Thanks for the recommendation! We have added this paper here.

Figure 5 – PRECT in the plot label(s) is not necessary and not defined in the caption either. Also, PRISM is not an observation, but rather an observation-based gridded product. Also, why is CARRM not shown? I would show the CARRM 5-year normal, wettest year, and driest year (for visual comparison).

Changed "PRECT" to "Precipitation" in all related figures. Also changed PRISM to "...for PRISM observation-based gridded product". The wettest, driest, and 5-year average of CARRM are also shown. We have also added the unsplit Livneh gridded product, which shows less underestimation of extreme precipitation compared to the time-adjusted Livneh \citep{Pierce2021}. The PRISM and the unsplit Livneh are very similar (which may not be news to you). We have to use different colorbars for PRSIM/Livneh and CARRM to get a clear visualization (CARRM's driest year in the 2015-2020 period is even wetter than the 5 wettest years in PRISM/Livneh). We have moved this figure to Appendix.

Line 340 – please be consistent with naming conventions "RRM" should be "CARRM". Also, this point about "exceeding the wettest years of PRISM" is important to note, especially since you only simulated 5-years. If CARRM simulated 30-years, would this bias amplify? I would argue given the large interannual variability of temperature/precipitation in California that the authors need, at least, 15-20 years of simulation to know if the CARRM's temperature/precipitation/SWE mean, median, IQR summary statistics are converged. Notably, PRISM (and other observation-based gridded products, such as Livneh, etc.) has been known to underestimate extreme precipitation (particularly from ARs) as discussed in - https://agupubs.onlinelibrary.wiley.com/doi/full/10.1029/2023MS003793 and - https://journals.ametsoc.org/view/journals/bams/100/12/bams-d-19-0001.1.xml . Given this issue in PRISM (and other products) and the potential to "falsely" attribute an over precipitation bias (especially with only five simulated years), could the authors compare CARRM to this observation-based gridded product for extreme precipitation - https://dataverse.harvard.edu/dataset.xhtml?persistentId=doi:10.7910/DVN/LULNUQ (more details in this publication - https://link.springer.com/article/10.1007/s00382-019-04636-0#Ack1 )

Thanks for the recommended dataset and literature. We have tried to get the 10-year return values of the CARRM largest seasonal daily precipitation based on the first two segments, or the full 20 years of available output. The location, shape, and scale parameters for the GEV distribution were estimated using Maximum-Likelihood Estimation. However, it is conceivable that only 10-20 years of data would fail to yield

reliable estimates of the GEV parameters. We did find that the extreme values weakened quite a bit when using 20 years of data as input than when using 10 years of data (though still significantly stronger than the probabilistic gridded product you recommend). We have added Fig. B1 to show the (rough) comparison of return values and we have cautioned against its interpretation.

Considering the substantial changes in the baseline comparison part (and the introduction of observational/reanalysis dataset), we didn't paste the updated text here. please see the updated manuscript in \subsubsection{Evaluation datasets} and \subsection{Baseline comparison with observations}.

Line 350-358 – there is a rich literature in RCMs, GCMs, VR and RRM simulations of California precipitation.  I would compare and contrast more studies here (particularly newer studies) to the CARRM results.  This will further add to the richness of this study.  Also, can the authors produce a plot of total precipitation, convective precipitation, stratiform precipitation, and the % contributed by convective precipitation?  This will provide more precision in highlighting which aspect of the CARRM precipitation simulation is driving the "error" or "mismatch" compared to PRISM (which could inform future CARRM simulations and/or where to target subgrid-scale parameterization development).

Thanks for the suggestions. We have added some recent literature here. Please note that the non-CP scale or purely phenomenological (no discussion of model bias) papers were not included (e.g., Huang and Ullrich, 2017; Rhoades et al., 2018a; Rhoades et al., 2020a; Rhoades et al., 2020b; Huang and Swain, 2022), although they are great resources pertaining to California hydrology. These papers have been added to the introduction or conclusions.

The three papers added are 1) Huang et al. (2020) comparing 3 km vs. 27 km WRF hindcasts for 1980-2017 cases, 2) Rhoades et al. (2023) using 3 km RRM-E3SM hindcasts for the 1997 flood event, 3) Bogenschutz et al. (2024, to be submitted) using 3 km to 800 m CARRM hindcasts for representative AR events. There is no wet bias in the 3 km WRF hindcasts, but there is a wet bias in the Sierra Nevada for the 3 km RRM-E3SM and 3 km / 800 m CARRM hindcasts. Bogenschutz et al. (2024, to be submitted) serves as a direct comparison to this work because we use the same code base (SCREAM) and RRM configurations; the main difference is that our simulations are not hindcasts (i.e., our boundary conditions are quite different from reanalysis). The wet bias found in Bogenschutz et al. (2024, to be submitted) is much weaker than what we found here, suggesting that most of the bias produced by CARRM climate runs is likely due to the large-scale forcing rather than biases in the physics.

The % contributed by convective precipitation is zero because CARRM didn't turn on a deep convection scheme since the resolution is convection-permitting and a scale-aware deep convection scheme hasn't been available in SCREAM. If you're referring to a phenomenological meaning, not a scheme meaning (I don't think you meant that because you said "to target subgrid-scale parameterization development"), we cannot distinguish convective and stratiform precipitation either because an algorithm generally requires a 3D precipitation / radar reflectivity structure (Houze et al., 2015), which we didn't output.

Houze Jr, R. A., Rasmussen, K. L., Zuluaga, M. D., & Brodzik, S. R. (2015). The variable nature of convection in the tropics and subtropics: A legacy of 16 years of the Tropical Rainfall Measuring Mission satellite. Reviews of Geophysics, 53(3), 994-1021.

Line 355 – do the authors think that simulations at 0(100 m) are truly needed in every hydroclimatic region (i.e., is this estimated convergence point in precipitation the same for the Amazon as it is for the California Sierra Nevada)? Wouldn't this 0(100 m) mostly be for regions where precipitation is dominated by convection rather than stratiform precipitation (like in California)? I think these studies makes the case that, in the midlatitudes, nonhydrostatic simulations at convective resolving scales may have diminishing returns in enhanced skill - https://agupubs.onlinelibrary.wiley.com/doi/10.1029/2021MS002805 - and - https://agupubs.onlinelibrary.wiley.com/doi/10.1002/2016JD025287

We have no answer to this. The cited papers are focused on individual clouds or cloud systems or focused on real-case cloud-resolving simulations over a specific region (Alps). If I understand correctly, the second paper discusses whether the non-hydrostatic and hydrostatic simulations would converge at coarser resolutions (36 km), so it's different from the convergence problem itself. We have added Liu et al. (2022) although it's in the context of idealized rising thermal bubble experiments.

Line 359 – why is snowpack the most "prominent" quantity? Also, SWE stands for snow water equivalent (i.e., the amount of water that would be produced by the snowpack if it were instantaneously melted).

We're just trying to convey that the added values of the snowpack are the most obvious (compared to low-resolution simulations, which has almost no snow). We thought snowpack could be expressed as SWE (we just wanted to use a more generic term). Yes, it's the same thing, just in the model output description it's "water equivalent snow depth (kg m-2 / kg m-3)". We have changed this passage to:

"Snowpack is the most prominent quantity to demonstrate the added value of using CARRM (when compared to the poorly resolved snowpack in the low-resolution simulations), which is represented by snow water equivalent (SWE, or water equivalent snow depth, i.e., the amount of water that would be produced by the snowpack if it were instantaneously melted) (Fig. \ref{compOBS1-PRECT-SNOWHLND})."

Figure 6 – what are the units of each of these quantities?  Delete PRECT and SNOWHLND.  Why are 30-year normal and 40-year normal used?  Also, what are the Sierra Nevada only average values for SWE in CARRM vs SWE UA (a statewide average incorporates a lot of 0s into the averaging)?  Or, alternatively, what is the km^3 of statewide SWE storage (this is a more useful metric for water resource managers anyways)?

We have added the units of precipitation (mm/day) and SWE (m) in the figure caption. We have also deleted PRECT and SHOWHLND. We used the default full available period for each dataset, but we have changed to a uniform period (1984-2020). We cannot calculate Sierra Nevada average values as we do not have an adequate shape file to perform such a calculation. We have put "megaton (1 Mt = 1,000,000,000 kg)"  (kg = m * kg/m3 * m2) of statewide SWE storage in the figure caption.

Line 361-362 – what is meant by "essentially captures"?  How is that quantified?  Also, I would argue that the authors need to utilize, at least, one other SWE reanalysis product given the large uncertainties in spatiotemporally estimating SWE combined with the fact that there is large interannual variability of SWE in the California Sierra Nevada.  I would suggest this product for comparison - https://www.nature.com/articles/s41597-022-01768-7

We simply wanted to say CARRM's topography captures the key topographic features in USGS 3 km dataset in California (unsurprising since RRM's topography was processed from USGS GTOPO 1 km data and cubed on 3 km). Thanks for the recommendation again; we have added this dataset in our analysis.

Line 365 – "four segments" or "four time periods"

Changed.

Figure 7 – delete TREFHTMX, "degC" to "∘C", eliminate redundant titles.  Why is JJA only shown?  Add DJF plots too.  Again, inconsistency in using SCREAMv0 CARRM vs RRM vs CARRM, pick one and be consistent throughout.  Can the authors provide difference

plots between the five-year periods, put them in the supplemental, and highlight them in the main text?

All abbreviations have been expanded to maximum/average/minimum T2m. Sorry for the misleading text in the figure caption—we show maximum, not average T2m here. In terms of only showing JJA, JJA is a key season in terms of the CA electrical grid, and daily maximum T2m is related to high risks (heatwaves). We have added DJF and difference plots in the Appendix too (highlighted in the main text).

Line 373-374 – this is an incredible increase in temperature in the Sierra Nevada.  Is this for JJA?  How does that compare to other simulations produced for California?  What about DJF?  In fact, why isn't DJF (or other seasons) discussed?

Yes it's for JJA. The warming level of daily maximum/average/minimum T2m over Sierra Nevada about 9 °C in DJF. Comparison to other works which shows warming patterns:

- Walton et al., 2017 (hybrid dynamical–statistical downscaling): the warming level from 1981–2000 to 2081–2100 is 6-8°C in July.

We added plots for DJF to supplementary material.  In addition, we have added a short discussion for DJF here:

"The DJF daily maximum T2m is shown in Fig. \ref{spa-TREFHTMX-DJF} and Fig. \ref{spa-TREFHTMX-DJF-diff}. The warming level over the Sierra Nevada about 9 °C in DJF. This is expected to have a significant impact on snowpack."

Line 377-378 – citation?

No, simply from the CARRM projection. Changed to:

 "Moreover, by the end of the century, most of California is expected to experience prolonged periods of heat waves according to CARRM projections".

Line 380-381 – "which is less dependent on intricate model physics".  I'm not sure what is meant here because temperature is shaped by land-sea contrasts, lapse rates in complex terrain, cloud spatiotemporal characteristics, land-atmosphere interactions (fog formation), solar radiation seasonality, etc.  I would consider these processes "intricate".

Sorry about the misleading text. What we're trying to say is that compared to temperature, 1) the parameterizations of precipitation process introduce far more assumptions, and far more inter-model spread (you wouldn't expect land-sea contrasts and the lapse rate itself to vary much from model to model, although the reasons behind that are not simple, and the description of solar radiation is even more fairly consistent and deterministic), 2) there is a far greater spatial inhomogeneity in the precipitation even without topography, 3) we nudged temperature, humidity and horizontal winds here, so temperature is directly constrained by the low-resolution simulations, but precipitation can still be significantly different with the constrained atmospheric conditions (the microphysics parameterization is enough to make differences), and 4) the response of statewide-averaged temperatures to GHGs is much more clear than precipitation (a less relevant discussion is that you can use the simplest box model to understand the response of global mean surface temperature to perturbations in the global mean energy flux, if the spatial structure of the response is self-similar). We have modified this part.

Line 384-386 – this single agricultural example is a little random (i.e., why are grapes chosen and not other agricultural commodities) and simplified (i.e., wine grape growth surely responds to more than just 10 °C temperatures).  I would add a few other agricultural impact examples here, if kept.  Also, in my opinion, an even more important issue than wine grapes would be the working conditions that agricultural workers/animals will face in a warmer world?  See this recent study for more details/metrics to employ - https://www.nature.com/articles/s41467-023-43121-5

We recognize the short sightedness of our original text. Thanks for the recommendation! We have added this paper:

> "Even more importantly, extreme temperature and humidity associated with climate change has a great impact on human survivability, especially for older female adults \cite{Vanos2023}."

Figure 8 – add the observation-based gridded product(s) here too.  Since the observation-gridded products have 30-40 years, I would also add +/- standard deviation or confidence intervals vertical lines to the histograms (especially to see if the 5-year simulated periods fall within interannual variability).  I'd do this for all histogram/bar chart plots.  Change "TREFHTMX", "TREFHT", etc. to their actual names (no need for random acronyms that are not used in the main text or caption).  Please be consistent with model naming conventions.  "degC" to "°C"

Thanks for the nice suggestion. We have added the comparison with PRISM and SWE UA. Given the similar statewide mean SWE between UA and WUS-SR, we used UA to speed up the analysis. The standard deviation bar has been added for all seasonal histograms. All abbreviations have been expanded. All "degC" has been changed to "°C".

Line 388-391 – why not use lat/lon locations that have an observation station and compare to the CARRM simulations?  Then you could add these true observations to Figure 9 (and others in the study).  This would also provide an additional comparison to the observation-based gridded products which may/may not include some of these stations and highlight uncertainties from the gridding process/statistical co-variate assumptions employed in these products.

Thanks for the nice suggestion. We have added GHCN for daily maximum/minimum, precipitation, and SNOTEL for SWE as a comparison. The stations have been updated. We keep the similar location around Sacramento, Death Valley and San Francisco, and replaced Conness Glacier to Tahoe City Cross for temperatures. We choose those stations for a varying microclimate across California and for a common interest (around cities/sights).  We also grabbed the daily timeseries of PRISM and SWE UA for the same locations.

We have modified this part substantially to include the new observations and stations.

Figure 9-10 - Change "TREFHTMX", "TREFHT", etc. to their actual names (no need for random acronyms that are not used in the main text or caption).  "degC" to "°C".  Please be consistent with model naming conventions.  The 5yr, 3mon, 30d should be put in the caption and the "seasonal" title deleted.  Add observation-based gridded product(s) here too as this will highlight where the 5-year simulations fall within the distribution of a 30-40 year observation-based gridded product.  This will also highlight if the authors should run additional simulation years.

Thanks for the nice suggestion. All abbreviations have been expanded. All "degC" has been changed to "°C". Figure titles and captions have been updated. The in-situ observation (GHCN) and observation-based gridded product (PRISM) from 1989-2020 water years have been added in the leftmost group.

Line 391-392 – the authors need to define more clearly what is meant by 5x3x30 (this is oddly better defined in the figure than the main text).  Also, delete "approximate".

We have changed this to "5 year x 3 month x 30 day". Deleted "approximate".

Line 395-396 – "highlighting the extreme heat and cold" I think what the authors are trying to say is the "wide range of temperature spatiotemporal variability across California landscapes"?

Yes, we have changed according to your rewriting.

Line 397-398 – cite Figure 1 when referring to topography underrepresentation in E3SMv1

Added.

Line 402-405 – 60 degC would be substantially higher than the historical all-time record reached this past year - https://www.sfgate.com/bayarea/article/what-is-hottest-temperature-in-death-valley-18254957.php - and might be important to highlight to provide context to readers. Also, if locations with station observations were used, this shocking finding would be, in my opinion, even more impactful to readers.

Thanks for sharing this news. We have highlighted it in the main text. Unfortunately I don't know how I can cite the news. But we can find close daily maximum T2m in the GHCN observational data in the updated figure: 54.4°C during 1989-2020 water years. We have added two sentence according to your suggestions:

> "It's alarming that 60°C would be substantially higher than the historical all-time record reached this past year (which is about 56.67°C). Note that the record of daily maximum T2m in the GHCN observational data in Fig. \ref{boxSP-1dA-TREFHTMX}d is 54.4°C during the 1989-2020 water years."

Figure 11 – see previous figure comments about use of random acronyms, repetitive figure titles, etc. Also, please provide difference plots, put into supplemental, and point to them in the main text.

Changed. Added.

Line 419-422 – "observed" or "projected"? Also, these mm/day numbers need to be better contextualized. For example, how does 5 mm/day translate to % of annual total precipitation? Or, km^3 of additional precipitation or …

Projected. We simply wanted to convey "we noticed / saw from the figure". We have changed all misleading "observed" to "projected" in the manuscript. We have added the % of annual total precipitation in the supplementary Figure. We have changed to:

"Compared to the baseline period, an increase of precipitation more than 30\% of annual total precipitation is projected in the northern, eastern, and southern Ranges (Fig. \ref{spa-PRECT-DJF-diff}). Some areas in the Great Basin Desert are projected to received more than 50\% of annual total precipitation. The Central Valley is expected to increase by 0-24\% of total annual precipitation."

Line 423-427 – does E3SM skillfully represent ENSO and its atmosphere-ocean teleconnection response to the coastal western United States?  For example, this study is a good lead for CESM - https://journals.ametsoc.org/view/journals/clim/36/1/JCLI-D-22-0101.1.xml  This might also be why there is weak relationship of the ENSO teleconnection response to warming.  Do the authors think a fully coupled simulation would lead to a different answer compared to an AMIP-style simulation (as used in this study)?

Thanks for sharing this study. I need to digest this paper more. It's impressive that the direct impact of intrinsic midlatitude variability (PNA, EPP) is significant after removing the linear influence of EP ENSO and El Niño Modoki in CESM1 LE (their Fig. 6). They found a high correlation between ENSO and EPP (related to AR activities), although "there is still substantial variability in both the PNA and EPP that is not explained by ENSO" and "imply that there will still be a relatively high probability for failed western U.S. hydroclimate responses to ENSO events". Compared to CESM1-LE, the ENSO variability in E3SMv1 is slightly closer to observations, while strongly shifted to a 3-yr period. The overall score for the spatial pattern compared to observations is also higher, but still muted along the North American coast (Golaz et al., 2019, Fig. 20, 21}. Our simulation lies between the AMIP and coupled simulation. Although our SSTs are prescribed, the SSTs themselves are derived from E3SMv1 fully coupled simulations. Thus, some of the effects of air-sea interactions have been included, whereas the interactions at fine scale are not represented here. If you are referring to a free-running fully coupled RRM, the answer would be yes. Even if the potential problems of free-funning coupled RRMs have been solved (radiative balance, scale awareness of the deep convection scheme, etc.), it would be difficult to find consistent conclusions about ENSO predictability in a single realization. We have added OBrien_Deser2023 here for more discussions here.

Line 426 – why are "heavy precipitation events" not shown?  This seems odd given the five-year simulations produced by CARRM to evaluate California's hydroclimate are likely too few to evaluate annual to decadal scale behavior in the model. Yet, these five-year simulations, evaluated at the event scale, would have a larger sample size to compare/contrast statistics between reference datasets and historical vs future climate

contexts. I think the authors should discuss the spatiotemporal characteristics of (for example) the largest 5-10 storms simulated and compare to in-situ, observation-based gridded products and future warming simulations. This shouldn't be left to "future work".

How can we make direct case-by-case comparisons with observations when we were not running hindcasts (the forcing data are not from reanalysis/observations)? Our baseline compares observations because it focuses on climatological statistics, but extreme precipitation events are so rare that picking a set of simulated cases will always find a set that "looks" comparable to another set in the observations, but this is meaningless. Comparisons with observations should be made in hindcasts (e.g., applying the storyline simulations in Rhoades2023 to multiple hydrologic extremes). A note on why this sentence is here: while this paper was being prepared, a corresponding hindcast paper by Peter Bogenschutz et al. was also being prepared. Since it was not ready at that time, we said that the examination of extreme events "will be investigated in future work". The default CARRM configuration used in that paper is the same as here, with the main difference being the nudging to ERA5 for specific AR events and is therefore suitable for comparison with observations, discussing the model performance of SCREAM-RRM, and discussing the sensitivities of the RRM grid configurations / experimental setups in more detail. Fortunately, Bogenschutz et al. (2024) is in the final stages of preparation with submission imminent.

Line 431 – "observations" – again, the authors do not compare CARRM to actual observations (but should!)

Added PRISM here.

Figure 12 – see previous figure comments about use of random acronyms, figure titles, etc. Also, please provide observation-based estimate histograms with +/- standard deviation/95% confidence intervals to see if the historical experiments are within interannual variability range AND the warming level histograms fall inside/outside historical interannual variability.

Thanks for the nice suggestions. Acronyms and titles have been modified. The standard deviation bar has been added for each histogram.

Line 440-442 – citation on the importance of organized convective systems for California precipitation? Also, similar to an earlier comment, please provide a plot showing the total precipitation, convective precipitation, stratiform precipitation, and % contributed by stratiform precipitation across all five-year CARRM experiments.

Recently, looking at the animations of precipitation and outgoing longwave radiation at the end of the century, I noticed a few suspected MCS. These are very large, with both local origins or being advected from the east during the summer. Precipitation is sparse the rest of the summer. I've rewritten the sentences:

> "We noticed a few MCS-like convective systems that can originate locally or propagate into California from the east during the summer, especially at the end of the century (not shown). They are characterized by prominent longwave radiative cooling which can rival the magnitude of mesoscale convective systems and tropical cyclones."

As we replied to an earlier comment, the % contributed by convective precipitation is zero because CARRM doesn't have a deep convection scheme and it is not trivial to separate convective from stratiform precipitation.

Line 442-444 – again, it is odd that the authors state that 6-hourly precipitation is important but say it will be covered in future work, especially since hourly or event scale analysis would be more fit for purpose when using 5-year simulations (much more robust sample size of storm events than annual/seasonal averages), particularly in the context of California hydroclimate/hydrometeorology

Please see our response to an earlier comment of comparing extreme events in CARRM to observations. Also, the MCS-like or convective system "events" in that sentence are rather rare, it's just striking with the scarce summer precipitation.

Line 439-447 – it is striking that mesoscale convective systems get significantly more mention in this entire section than atmospheric rivers (ARs) and extratropical cyclones (ETCs), yet ARs and ETCs are the dominant sources of precipitation for California…

This is partially due to the fact that we have a whole section on ARs later in the text… And partly because we want to highlight the implications of CARRM in this part. Winter precipitation / ARs are well known to be important, while the summer and fall precipitation have received much less attention and CARRM is likely more capable at simulating these events than 1° E3SM.

Line 449-455 – I'm highly skeptical that the southwest monsoon/North American monsoon is driving these precipitation change signals in CARRM over California (Figure 13)… the largest changes in precipitation occur over mountains and, therefore, are likely orographic convection.  Also, to definitively make this statement with actual evidence, the authors would need to do feature tracking of MCS/monsoon events…

Sorry for the misleading text. The subject of this entire discussion is the desert highlands southeast of the Sierra, not Sierra Nevada itself. By the end of the century, summer precipitation is generally projected to increase by 10-20% of annual precipitation over most of the Southern Desert. We have clarified the object to "Southern Desert" in this paragraph. We have removed the statements attributing the precipitation to MCS/monsoon in CARRM while retaining the literature on the southwest monsoon.

Line 460 – "compelling indicator", how so and why more so than other climate variables?

Simply because there is no snow in low-resolution simulations. We just wanted to say a compelling indicator for the "added values" of high resolution.

Line 460-463 – add citations to back these statements about "thickest during the spring season", snow sensitivity to warming and that the Sierra Nevada will be "essentially devoid of snow by end of the century" There is a rich literature (esp. in recent years) investigating all of these points made.

Thanks for pointing this out. We have added some literature here:

> "The response of snow sensitivity to warming in the Sierra Nevada is consistent with recent works \citep{Berg2016,Rhoades2017,Rhoades2018b,Sun2019,Siirila2021}. They found that under the impacts of climate change, California and the western U.S. will experience significant reductions in SWE, including reduced winter snowfall, and earlier spring snowmelt."

Line 465 – "6 degC" I would state "a local warming of 6 degC"

Modified.

Line 473-474 – "SWE threshold of ~0.2 m"? Do you mean to say snow-to-liquid ratio (i.e., the amount of water produced for a given amount of snow depth)? If so, this snow-to-liquid ratio can vary substantially from season to season and across mountain regions, especially in maritime versus continental mountain ranges.

Yes. We have added your caution for this threshold.

Figure 13 and 14 – see earlier figure comments about redundant titles, acronym usage, etc. Also, please add observation-based gridded products for comparison.

The titles cannot be removed (the subplot size would be different in NCL). The acronym has been expanded. The observation-based gridded products have been added and put in the Appendix.

Figure 15 – this figure seems unnecessary. Why not just add these lat/lon grid cell locations (and those used for Figure 9, etc.) onto Figure 1? Also, why not use lat/lon locations where in-situ observations exist and add them to compare to CARRM?

Removed this figure.

Thanks for suggestion of station locations. The stations for SWE have been updated to Tahoe City Cross, Adin Mtn, Truckee #2, and Leavitt Lake to match the available SNOTEL sites.

Figure 16 – see earlier figure comments related to box and whisker plots, etc. Please add observation-based gridded products for comparison.

Changed and added.

Line 479-486 – "rapid melting in spring" this is not shown. Can the authors provide a water year xy plot of daily/hourly SWE accumulation/melt cycles that shows this spring snowmelt response? This is especially important for water resource managers in California and would not show up clearly in the MAM average map plots and/or box and whisker plots.

Thanks for the nice suggestion. We have added the annual cycle of SWE for observations and simulations. This paragraph has been substantially updated since we have changed the stations according to SNOTEL.

Line 497 – "is scale aware should" to "is scale aware and should"

Corrected.

Line 512 – "very 500" to "every 500"

Corrected.

Line 515 – "dumped" to "is estimated to have dumped"

Modified.

Line 511-512 – "Atmospheric River trends over California" by this time in the study, readers have forgotten that the authors have tracked atmospheric rivers using TempestExtremes.  Please remind them here right away.  Also, how many AR events are tracked in the CARRM simulations across each five-year period?

Thanks, we have reminded readers here that we tracked ARs using TempestExtremes. In our simulations ARs always show up one after the other in the eastern Pacific winter, with very few breaks, it's just that some hit California, and some don't.  In addition, because we were interested in the change in PDF over time for the variables of interest, and in order to compute the PDF with as large a sample size as possible, we did not isolate individual AR events with StitchBlobs. This would make the sample size of variables corresponding to single AR events in every 5-year winter very scarce. We have added an explanation of why we chose a simpler method in the PDF figure in Methods.

Line 512-533 – these five-year CARRM simulations/experiments are definitely not fit for purpose in evaluating an 1-500 to 1-1000 year event (ARkStorm)… so why is this an major stated goal of the study?  At best, these CARRM simulations would confidently estimate the 1-20 year storm (unless the authors were extremely lucky in sampling internal variability in E3SMv1 and/or SCREAMv0)?  With that said, I would advise the authors completely revise this entire paragraph to be within the flood return frequency context that the authors feel their CARRM simulations are fit for purpose in answering (e.g., 1-20 year event).  Even if the authors don't discuss the ARkStorm, I think evaluating, for example, the 1-20 year event is still useful and important to the California Department of Water (DWR) as they will likely not overengineer their built infrastructure/management system to be resilient to the 1-500 to 1-1000 year storm ARkStorm like event.  For example, see CA DWR's Central Valley Flood Protection Plan on the major flood events of the last 160 years for guidance on other potential events to evaluate CARRM skill (some of which are closer to the 1-20 to 1-50 year flood events)  - https://water.ca.gov/-/media/DWR-Website/Web-Pages/Programs/Flood-Management/Flood-Planning-and-Studies/Central-Valley-Flood-Protection-Plan/Files/CVFPP-Updates/2022/Central_Valley_Flood_Protection_Plan_Update_2022_ADOPTED.pdf

ARkStorm is one of the potential applications of CP-RRMs that grew out of our interest. Due to its remarkable implication, we would like to have a conceptual exploration of ARkStorm in CARRM. Due to the insufficient sample size, we have added a lot of discussion of ARkStorm analysis in the literature. It is clearly insufficient to sample internal variability by this one realization alone. The CARRM simulations are also inadequate to answer 1-20 year events, as samples of up to 20 years do not yield

reliable GEV parameter estimates (please see also our response to the probabilistic gridded product). In response to CARRM's ability to capture extreme AR events, hindcasts are needed for an apple-to-apple comparison with observations. As in previous responses, this is explored in Bogenschutz et al. (2024). Here's a personal opinion: for the question of probabilistic changes in extreme events (independent of the PGW study), I think that low-resolution multi-ensembles are still more appropriate at this time to provide a range of probabilistic changes in the synoptic-scale background (sacrificing resolution to get enough samples), whereas a high-resolution model like CARRM is appropriate to provide the added value of local prediction given the chosen synoptic background derived from low-resolution models (to simulate only the case where the trend in temperature/precipitation lies at the ends of the spread). This is out of the scope of this work. Nevertheless, this figure illustrates that under the similar synoptic background, CARRM is significantly different from the low-resolution model for the change in probability over time for any given extreme event reference (here we used the ARkStorm level, but similarly for other levels).

Here we have added Bogenschutz et al. (2024) and added a discussion of return frequency.

Figure 18 – see earlier figure comments related to box and whisker plots, etc. Please add observation-based gridded products for comparison. I would also either add more flood event markers in addition to the ARkStorm Ref. I don't think the CARRM simulations are fit for purpose in comparing to ARkStorm (would require E3SMv1 LE and/or many more simulated years of SCREAMv0). See CA DWR report mentioned in earlier comment to track down other flood events/reference markers.

This figure has been updated according to your suggestions. We cannot add observation-based gridded products (PRISM) to this figure because PRISM does not seem to provide the daily products for the whole map (daily timeseries can be grabbed at individual stations), so we cannot get the 30-day running average for PRISM. I also thought about using unsplit Livneh gridded product because the data has daily statewide map and Fig. A1 shows that the data is extremely close to PRISM. However, that data is not suitable for a 30-day running average because the timing is not standardized. We have added an explanation here.

Thanks for the reference you recommend in the previous comment. I only found a definition of 100-year / 200-year flood in that document… This looks like a document on the administrative side, and I didn't find the statewide average/accumulative 30-day precipitation corresponding to these events, so I cannot reference to it directly.

Line 535-537 and Figure 19 – can the authors provide AR vs non AR contributed precipitation maps or stacked histograms or ... to show readers this point about AR contributed precipitation.  Figure 19 is unnecessarily difficult to read and interpret.  For example, could the authors reproduce Figure 2 from this study - https://agupubs.onlinelibrary.wiley.com/doi/full/10.1029/2020GL089096

This figure appears complex probably because we want to express the information in a single figure. This figure showed 4 variables x 4 segments x 2 simulations. Similar to Rhoades et al. (2020) Fig. 2, we initially divided the sample into {no AR, weak AR, AR category 1-5} based on the definition of Ralph et al. (2019). The sample size was dramatically smaller (one sample every 6 hours becomes an AR event every few days) for PDF plots. For the sake of uniformity in subsequent mapping (all four subfigures are PDF distributions, and continuous variables are more appropriately represented by PDFs than discrete quantities of length 5), we chose to show the IVT values directly (Fig. \ref{AR-PDF}b), rather than the category derived from the IVT values.

However, we have substantially modified this figure according to your suggestions. The subfigure AR-contributed precipitation (Fig. \ref{AR-PDF}a), shows seasonal averages (instead of PDFs) after updating the method according to your next suggestion. It's now more similar to Fig. 2 of Rhoades et al. (2020) now (but without the category distinction). The remaining three variables still show PDFs, but are as consistent as possible with Fig. \ref{AR-PDF}a, splitting the 4 segments into different positions on the x-axis from left to right, and showing only DJF for brevity. The updated graph is actually not as clear as the original in terms of expressing the change in PDF shape over time (since the x-axis of the 4 segments are stacked together for direct comparison in the original), but it may appear simpler.

Line 538-547 – do the authors only attribute AR-contributed precipitation based on the immediate grid cells where AR conditions coincide with precipitation (i.e., in grid cells where TempestExtremes binary masks are at a value of 1)?  If this is the methodology employed by the authors, this would lead to a considerable underrepresentation of AR-contributed precipitation, particularly at the leading edge/peripheries of the AR masks produced by TempestExtremes.  Why don't the authors instead use a shapefile/box region over California and associate any precipitation as AR-produced when an TempestExtremes AR mask exists over California?  This would avoid the issue in which an AR's IVT plume depletes rapidly below 250 kg/m/s as the AR progresses across California's complex terrain and would thus underrepresent AR-contributed precipitation during times/locations of heavy precipitation.

In the previous version, we assigned the overlapped precipitation over the CA shapefile mask and binary AR masks as AR-contributed precipitation. I agree that this will lead to

an underestimation of the AR contribution as discussed in the original main text. Thank you for the nice suggestion. We have updated the method that associates any precipitation as AR-produced when binary AR masks exist over California. This paragraph has been deleted.

Line 538-554 – This statement, "In summary, the greatly increased total precipitation in California by end of century is primarily due to greater amounts of precipitation falling from individual storms instead of a greater number of storms, which is dominated by larger precipitable water under the substantial warming scenario", is, in my opinion, not backed up by evidence in the study and inconsistent with other studies in the AR literature. If I read the Methods correctly, the authors didn't actually isolate distinct AR events in TempestExtremes using, for example, StitchBlobs (i.e., estimate when each individual AR lifecycle starts/ends). If that is correct, how could the authors make a statement about changes in single AR event precipitation totals vs sequential AR event precipitation totals (nor the number of AR events with warming)? I'm also guessing the conservative approach the authors took in estimating AR-contributed precipitation (only immediate grid cells where TempestExtremes says AR conditions exist) would bias this statement. For example, the core of the AR events (immediate grid cells where TempestExtremes says AR conditions exist) would cover larger and larger areas of precipitation as integrated water vapor increases with Clausius-Clapeyron. This would likely erode the AR periphery issue of AR-contributed precipitation (i.e., more and more grid cells over California would be under AR conditions as integrated water vapor increases with Clausius-Clapeyron). The alternate approach suggested in estimating AR-contributed precipitation (see previous comment) would fix the area over which AR-contributed precipitation is estimated and eliminate issues with changing areal extents of AR masks in warmer climates.

First, please note that all our results are based on the SSP585 realization that we performed. Given the large uncertainties in predictions of regional precipitation trends (radiative forcing, internal variability, resolution, model representation), I don't expect precipitation trends in California to be consistent across the relevant literature. The original sentence is misleading for the term "the number of storms". We have changed this sentence:

> "In our simulations, the large increase in total precipitation in California by the end of the century is primarily due to larger amounts of precipitation falling from stronger rather than more frequent moisture belts hitting California, which is dominated by larger precipitable water under the significant warming scenario."

Please see the previous response, we did not isolate individual AR events. We did not make any statement about "AR events". Therefore, we didn't intend to make a statement about changes in single AR event precipitation totals vs sequential AR event precipitation totals (nor the number of AR events with warming). The sentences related to the terminology "AR" have been clarified in Methods:

> "We did not isolate single AR events in TempestExtremes using StitchBlobs in order to compute the PDFs with as large a sample size as possible. Using StitchBlobs would make the sample size of variables corresponding to individual AR events in each 5-year winter very small. As a result, we did not divide ARs into a category-based definition such as in \cite{Ralph2019} and \cite{Rhoades2020b}. Therefore, the terminology ``AR'' in the context of this paper is strictly AR-related IVT 6-hourly samples."

Finally, we have modified the method for estimating AR-contributed precipitation and updated Fig. \ref{AR-PDF}a as per your last suggestion:

> "AR-contributed California precipitation was obtained by interpolating the 1° AR mask back to the model's native pg2 grid and then associating any precipitation as AR-produced when AR masks exist over California."

Line 555 – typically this subsection title is labeled "Discussion and Conclusions"  Also, if this is truly a Discussion, the authors needs to compare and contrast the findings made in this study with the broader literature more consistently throughout each paragraph of this section.  Please do so as this allows the reader to know if the author's findings are consistent/inconsistent with the broader literature.  This will also elevate the novelty/richness of the study.

Thanks for the suggestion. The subsection title has been changed. To my knowledge, there is little literature on comprehensive future climate projections for California at the convection-permitting scale, so I cannot make direct comparisons. But there are many recent literatures on downscaled future projections of the Sierra Nevada snow, and some AR hindcast studies using convection-permitting models. We have added them.

Line 556 and 578 – "SCREAM" again, inconsistency in naming conventions used throughout the study.

This phrase is meant to emphasize SCREAM as a base model, as different RRMs can be developed on top of SCREAM.

Line 561-562 – "Through the development history …" this sentence does not seem necessary.

We wish to keep it to emphasize the efforts of those who came before us and emphasize the usability of the tool. We have modified this sentence to:

"Thanks to the development efforts and documentation provided by the E3SM RRM community, the tool chains and workflows for generating new RRM grids are relatively mature."

Line 571 – "high internal variability" or extreme phasing of climate modes of variability (e.g., ENSO)?

Modified.

Line 578 – delete "with SCREAM" and put ", respectively,"

Modified.

Line 581 and 583 – "CA" to "California"

Modified.

Line 584 – "SCREAM-RRM" again, inconsistency in naming conventions used throughout the study.

Again, the subject in this sentence is the SCREAM-based (potential) RRMs, not CARRM only. We deleted "our" here.

Line 585 – "observations" to "observation-based gridded products"

Modified.

Line 592 – delete parentheses around "(positive)"

Deleted.

Line 597-599 – "This increase primarily stems from greater amounts of precipitation…" this statement is not backed up by the methodology. See earlier comments about how the authors used TempestExtremes.

Sorry for the misleading text. We have changed this sentence to:

"This increase is primarily due to larger amounts of precipitation falling from stronger rather than more frequent moisture belts hitting California. This aligns with the thermodynamic reaction to warming, resulting in an increased amount of precipitable water."

Line 607 – Change "meaning" to "context"

Modified.

Line 609 – These CARRM simulations are not "predictions". These CARRM simulations are "projections" based on a possible global socio-economic development pathway (SSP585).

Modified.

Line 610 – "warm bias". See earlier comment about DJF, especially in mountain regions. I believe CARRM has a "cold bias" in that context. I think its important to be nuanced when stating "warm bias".

Thanks for the good catch. We have updated this sentence.

Line 613-615 – this is why I suggested that the authors provide plots of total precipitation broken down into convective and stratiform and then further by % contributed as it will allow them to be more precise on where/when the CARRM convective parameterization matters. My hypothesis is that in California, convective precipitation is substantially smaller than stratiform precipitation (even with convective resolving simulations).

Please refer to the previous responses that the % contributed by convective precipitation is zero because CARRM doesn't have a deep convection scheme and it is not trivial to separate precipitation types. But I agree that in California, convective precipitation is substantially smaller than stratiform precipitation. We have changed "resolve convection" to "resolve precipitation systems".

Line 616 and 621 – "SCREAM-RRM" again, inconsistency in naming conventions used throughout the study.

The subject of this sentence is all SCREAM-based (potential) RRMs.

Figure A-C – see other map plot comments about redundant titles, delete variable acronyms, etc. Also, for Figure A-B, it is odd to not include Canadian and Mexican (etc.) state boundaries (Hawaii too?) Also, why not provide Figure A for all future climate

simulations in CARRM too.  Please also provide difference plots.  This will allow readers to visually see the amplifying effects of climate change on IVT through the Clausius-Clapeyron relationship and (potentially) diminishing effects of climate change on IVT through shifts in the storm track/jet that shape AR latitudinal variability.
Citation: https://doi.org/10.5194/egusphere-2023-1989-RC2

Thanks for the nice suggestions. This figure has been updated and all geophysical and US states boundaries have been included. We have added Fig. \ref{spa-IVTov850wind_DJF} to show the IVT and 850 horizontal wind vectors for all future climate simulations.

---

## Author Response (AR2)

I thank the authors for their concerted effort to address many of the points raised in my (admittedly) lengthy first review. I put in a considerable amount of time and energy into my review because I think this study is an important contribution to the literature and is a first glimpse of what the DOE's SCREAM model (combined with RRM) can provide in the climate impacts space. I look forward to reading more SCREAM-RRM-based climate impact studies in the coming years.

Thank you so much for such swift feedback, and thank you again for your careful reading, insightful suggestions, and thorough review! We have modified the manuscript according to your nice suggestions.

I have a few minor revision requests and would advise the editor to accept the manuscript pending these changes:

Line 470 - Change "older female adults" to "older populations that work in agriculture"

Changed.
Line 474 - Change "Tahoe City (one of the coldest city over High Sierra)" to "Tahoe City (a city representative of the High Sierra)"

Changed.
Line 505 - Change "an increase of precipitation more than 30% of annual total precipitation..." to "annual total precipitation is projected to increase by 30% in the northern, ..."

Changed.
Line 544 - Change "MCS-like" to "mesoscale convective system-like" (MCS was never defined up to this point)

Changed.
Line 581 - Change "agriculture and energy supplies for electricity" to "agriculture yields and energy supplies"

Changed.
Figure 15g - what is the random "orange blob" on the 0mm SWE line and left of the JJA x-axis label? Please check other plots throughout the manuscript for random artifacts.

Good catch! It's related to the processing of SNOTEL records— there are several missing values separated by two consecutive commas that were incorrectly skipped during processing. This bug has been fixed.

Besides, I found that the temperature records of "-60.3 F" seem to be invalid (please see the records from 1990-09-01 to 1990-09-06 at Tahoe City Cross pasted below, columns 4-6 are daily maximum/minimum/average temperatures). It makes no sense for such low temperatures and such large jumps. I didn't find any documentation for the -60.3 values, so I just set the corresponding records to missing values. After this change, the standard deviation of the temperatures is more consistent over the year.

> SNOTEL809_TahoeCityCross.6797ft.39.17162_-120.15362.1980-2024.csv
>
> ...
>
> 1990-09-01,0.0,25.2,37.8,-3.5,15.6,0.0
>
> 1990-09-02,0.0,25.2,39.4,-6.7,14.2,0.0
>
> 1990-09-03,0.0,25.2,32.2,-60.3,13.8,0.0
>
> 1990-09-04,0.0,25.2,,-60.3,-60.3,0.0
>
> 1990-09-05,0.0,25.2,,-60.3,-60.3,0.0
>
> 1990-09-06,0.0,25.2,,-60.3,-60.3,0.0
>
> ...

I also forgot to mention the quality control for GHCN in the last revision. Only values with an empty QFLAG field were kept, which means they passed all quality assurance checks. Figure \ref{OBS-SNOTEL-PRISMUA-SP-dayline} has been updated, and the description of the quality control for SNOTEL and GHCN has been added:

> "Only values with an empty QFLAG field are kept in GHCN records, meaning they pass all quality assurance checks. The temperatures of -60.3 F in SNOTEL records seem to be invalid and are set to missing."

Line 660 and Figure 17 - "Note here that we cannot add observation-based gridded

products...PRISM does not provide daily products for the full spatial distribution" A quick search on the PRISM homepage (https://prism.oregonstate.edu/downloads/) shows that this is not true as you can see that they provide PRISM daily fields in a publicly accessible FTP location - https://ftp.prism.oregonstate.edu/

Thank you for pointing to the FTP page! Added the statewide 30-day mean precipitation for PRISM from 1989 to 2020 to this figure. Text has been updated:

> "Considering the wet bias of CARRM as shown in Fig. \ref{boxRun30d-CA-PRECT} and Fig. \ref{compOBS1-PRECT-SNOWHLND}), the 14 mm/d statewide precipitation may underestimate the intensity of ARkStorm events. "

Line 683 - "...primarily due to larger amounts of precipitation falling from stronger rather than more frequent moisture belts hitting California..." Moisture belt or moisture surge? If you change to moisture surge, be sure to change everywhere else in the text too.

Thanks for the suggestion. All "moisture belt" has been changed to "moisture surge".

Line 747 - make sure that the citation formatting is correct throughout the manuscript. For example, in this sentence Huang et al. (2020) should not be in parentheses like this (Huang et al., 2020).

We have corrected this one and several other citations.